# A paired sequence language model for protein-protein interaction modeling

Jun Liu [1], Hungyu Chen [2] & Yang Zhang [1,2,3] ✉

Understanding protein–protein interactions (PPIs) is crucial for deciphering cellular processes and guiding therapeutic discovery. While recent protein language models have advanced sequence-based protein representation, most are designed for individual chains and fail to capture inherent PPI patterns. Here, we introduce a Protein Pair Language Model (PPLM) that jointly encodes paired sequences, enabling direct learning of interaction-aware representations beyond what single-chain models can provide. Building on this foundation, we develop PPLM-PPI, PPLM-Affinity, and PPLM-Contact for binary interaction, binding affinity, and interface contact prediction. Large-scale experiments show that PPLM-PPI achieves state-of-the-art performance across different species on binary interaction prediction, while PPLM-Affinity outperforms both ESM2 and structure-based methods on binding affinity modeling, particularly on challenging cases including antibody–antigen and TCR–pMHC complexes. PPLM-Contact further surpasses existing contact predictors on inter-protein contact prediction and interface residue recognition, including those deduced from cutting-edge complex structure predictions. Together, these results highlight the potential of co-represented language models to advance computational modeling of PPIs.

Protein–protein interactions (PPIs) are central to virtually all cellular processes, ranging from signal transduction and immune recognition to metabolic regulation and molecular assembly[1]. Understanding PPIs involves three fundamental and interrelated questions: Do two proteins interact? How strong is their binding? And which residues mediate the interaction? These questions map directly onto the key computational challenges of binary interaction prediction, binding affinity estimation, and interface residue identification[2–4].

Large-scale protein language models (PLMs), such as ESM[5–7] and ProtBERT[8], have demonstrated strong capabilities in learning protein sequence representations, driving progress in structure prediction[9,10], function annotation[11], protein design[12], and mutational effect estimation[13]. Although PLMs have been widely applied in PPI-related studies[14–21], these models are inherently trained on single-protein sequences and lack the ability to model inter-protein relationships. For protein interaction prediction, for example, methods such as

D-SCRIPT[15], Topsy-Turvy[22], and ESMDNN-PPI[23] leverage either LSTM-based encoders or ESM2-derived embeddings from a single sequence for binary classification, yet struggle to capture interface-level context. In the task of binding affinity prediction, most PLM-based approaches are designed for protein-ligand targets[24,25], with existing methods for protein–protein binding affinity prediction still depending heavily on complex structural models[26–30], highlighting a key opportunity for developing scalable, structure-free solutions based on sequence-derived representations. For inter-protein contact prediction, most methods are built on multiple sequence alignments (MSAs)[31–34]. While methods such as GLINTER[35], CDPred[19], and DeepInter[20] employ ESM-MSA-1b[36] to extract both intra- and inter-protein features, their performance is tightly coupled to the quality and availability of paired MSAs, as well as the fact that ESM-MSA-1b has been pretrained solely on monomeric sequences. These limitations, shared across all three major PPI-related tasks, underscore the need for a unified paired

[1]Cancer Science Institute of Singapore, National University of Singapore, Singapore, Singapore. [2]Department of Computer Science, School of Computing, National University of Singapore, Singapore, Singapore. [3]Department of Biochemistry, Yong Loo Lin School of Medicine, National University of Singapore, Singapore, Singapore. ✉e-mail: zhang@nus.edu.sg

protein language model that can directly learn inter-protein relationships from sequences.

In this work, we present PPLM, a protein pair language model specifically designed to learn contextual and relational features from paired protein sequences. To better capture inter-protein dependencies, we introduce a hybrid intra–/inter-protein attention mechanism that adapts transformer attention for co-represented protein pairs. Specifically, this mechanism integrates rotary embeddings-based positional encoding[37] for intra-protein attention with non-positional embedding for inter-protein residue pairs to avoid spurious spatial priors. In addition, learnable inter-protein attention weights and an explicit inter-protein attention mask are incorporated to enhance the modeling of cross-protein interactions while preserving a clear separation between intra- and inter-chain information flows. To support large-scale training, we further construct a comprehensive sequence-pair dataset comprising over 3.3 million high-quality protein pairs, enabling PPLM to learn robust interaction-aware representations. Building upon this foundation, we develop three task-specific methods for key PPI-related challenges: PPLM-PPI for binary interaction prediction, PPLM-Affinity for binding affinity estimation, and PPLM-Contact for inter-protein contact prediction and interface residue identification. Across all tasks, our model consistently outperforms existing methods. In inter-protein contact-map prediction, PPLM-Contact surpasses current state-of-the-art structure modelling-based methods, including AlphaFold2.3[38], AlphaFold3[39], and DMFold[40]. These results underscore PPLM's potential as a unified foundation for advancing computational PPI studies.

## Results

PPLM comprises four coordinated pipelines that support both core language modeling and its downstream applications. In the primary language modeling pipeline (Fig. 1A), each input sequence is independently tokenized with begin and end tokens and then concatenated to form the full paired input. The transformer blocks with a hybrid intra–/inter-protein attention mechanism are applied to jointly model intra- and inter-protein interactions. The resulting sequence embeddings and attention matrices provide versatile representations that can be flexibly leveraged across PPI-related tasks.

To demonstrate the versatility of PPLM, we developed three task-specific models that address major challenges in PPI studies. For PPI prediction (Fig. 1B), PPLM produces sequence embeddings together with intra- and inter-protein attention matrices for the sequence pair, which are aggregated through parallel max- and mean-pooling branches, each followed by a multilayer perceptron (MLP) for ensemble prediction. For binding affinity prediction (Fig. 1C), the final layer of PPLM is fine-tuned on a curated dataset of experimentally measured affinities. The full-sequence embeddings of the receptor and ligand proteins are aggregated using max pooling and then passed through linear layers to estimate their binding strength. For inter-protein contact-map prediction (Fig. 1D), the inter-protein attention matrix from PPLM is integrated with MSA-derived features and monomer distance maps to capture both evolutionary and structural information. A dedicated inter-protein transformer, composed of three specialized modules designed to update inter-protein representations, is used to predict contact maps and identify interface residues between the input proteins.

### PPLM improves the language modeling of protein pairs

Perplexity provides an objective, task-independent measure of a language model's confidence in predicting masked amino acids. Lower perplexity indicates higher predictive accuracy, with a value of 1 representing perfect prediction and ~20 reflecting near-random performance. We used perplexity to benchmark PPLM against ESM2[6], a state-of-the-art single-sequence language model that does not explicitly model inter-protein context. To more comprehensively evaluate model behavior on protein pairs, we implemented three complementary masking strategies (Supplementary Fig. S1).

Under a random masking strategy, perplexity was computed by iteratively masking non-overlapping 15% of residues in both sequences until all positions were covered. Identical positions in homomers were masked jointly to avoid trivial inference. As shown in the left panel of Fig. 2A, PPLM achieved average perplexities of 7.30, 5.08, and 4.50 on homomeric PDB pairs, heteromeric PDB pairs, and STRING-derived pairs, corresponding to reductions of 13.1%, 24.5%, and 21.0% relative to ESM2 (8.40, 6.73, and 5.70). Median perplexity showed consistent improvements of 15.1%, 28.5%, and 19.0%. These differences were highly statistically significant (Wilcoxon signed-rank test $p$-values $2.96 \times 10^{-139}$, $8.07 \times 10^{-177}$, and $< 10^{-300}$). A head-to-head comparison (Fig. 2B) further showed that PPLM outperformed ESM2 on 91.9% (4298/4678) of sequence pairs.

To evaluate PPLM's sensitivity to interface residues, we next masked only residues located at the experimentally defined interaction interface. For heteromeric pairs, we introduced Dual mode (masking all interface residues from both chains) and Single mode (masking all interface residues from one chain at a time); homomeric pairs used only the Dual mode to avoid trivial inference. As shown in the right panel of Fig. 2A, in the Dual mode, PPLM achieved an average perplexity of 6.79 (median 5.78), significantly lower than ESM2's 8.50 (7.75) ($p$-value = $9.90 \times 10^{-223}$). In the Single mode, PPLM again outperformed ESM2 (7.81 vs 10.36 in average perplexity; 5.62 vs 8.89 in median; $p$-value = $1.17 \times 10^{-208}$). Corresponding pairwise comparisons (Fig. 2C) showed that PPLM achieved lower perplexity in 80.7% (Dual) and 78.9% (Single) of cases.

To examine whether PPLM captures biologically meaningful dependencies, we analyzed the homodimer complex 1Y9B, a conserved transcription factor from Vibrio cholerae O1 biovar El Tor (Fig. 2D–F). The inter-protein attention matrix from the final PPLM layer showed high-scoring regions that closely matched the experimental contact map. Among the top 20 residue pairs ranked by attention, 90% corresponded to true heavy-atom contacts (80% to $C_\beta$–$C_\beta$ contacts). Of the 62 experimentally determined interface residues, 52 (83.9%) were recoverable from attention-derived residue rankings. The corresponding 3D visualization demonstrated that these predicted residues cluster around the physical interface, indicating that PPLM's attention mechanism naturally focuses on interaction-relevant regions during unsupervised learning. A more detailed representation-level comparison between PPLM and ESM2, including unsupervised and linear-probe analyses of inter-protein attention matrices, is provided in Supplementary Method S1.

As expected for paired-sequence modeling, smaller PPLM variants exhibited higher perplexity, whereas deeper models showed substantially improved performance (Fig. 2G). To assess potential pair-level redundancy and species-specific biases, we conducted a comprehensive bias audit by comparing all validation pairs to the training set. Most validation pairs had no detectable match, and PPLM's performance remained stable across identity intervals (Fig. 2H). PPLM consistently outperformed ESM2 across the entire identity spectrum, and this trend remained robust across alternative coverage thresholds (Supplementary Fig. S3).

Collectively, these results demonstrate that PPLM enhances language modeling of protein pairs, exhibits increased sensitivity to residue-level interactions at protein–protein interfaces, and effectively captures biologically meaningful inter-chain dependencies. These capabilities, enabled by paired-sequence training and a hybrid attention architecture, provide strong foundations for downstream tasks such as interface residue identification and structure-guided interface design, including antibody CDR optimization.

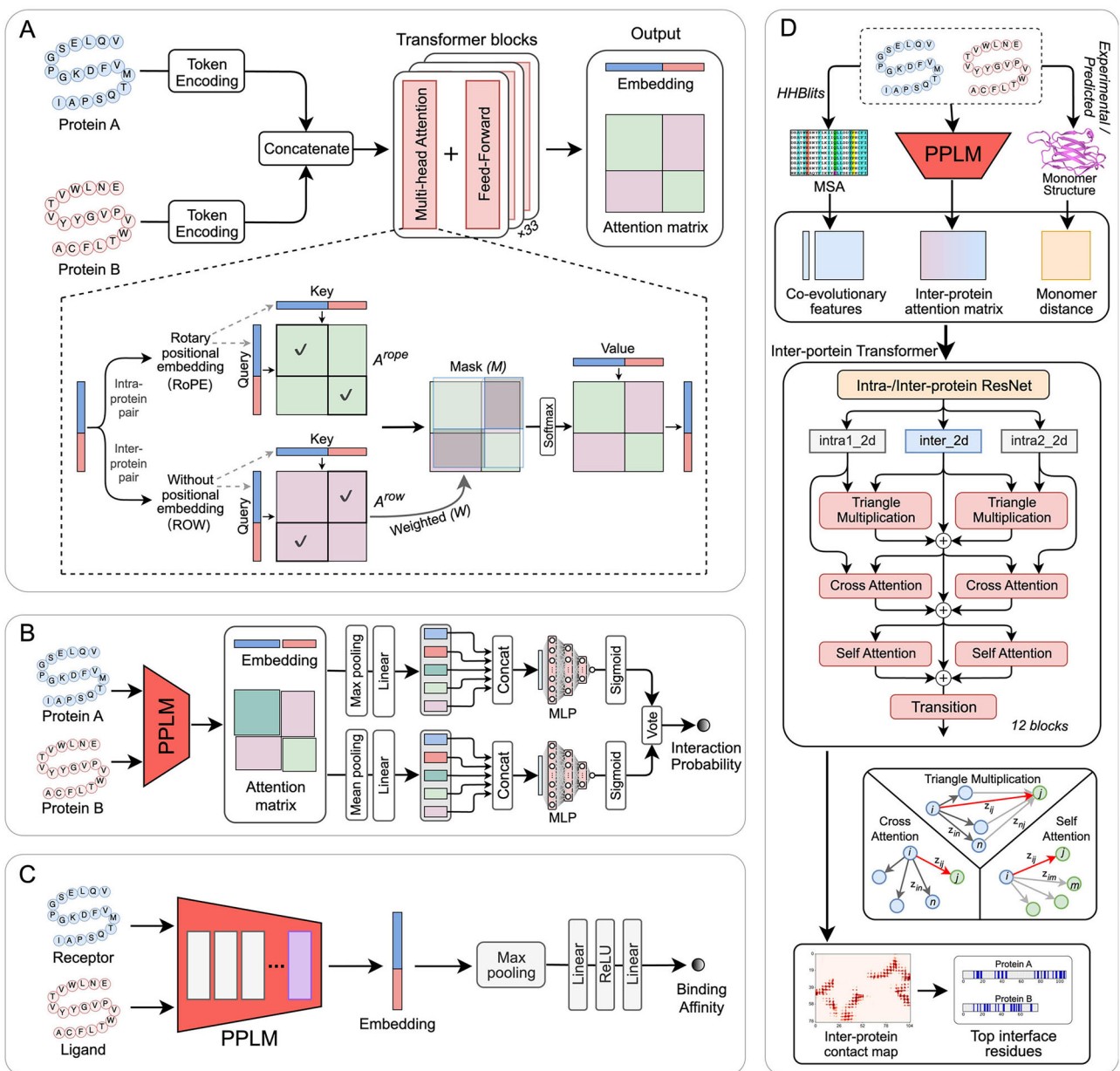

**Fig. 1 | Overview of the PPLM framework and its downstream applications.**
**A** PPLM architecture. The input protein sequences are independently tokenized and then processed through transformer blocks equipped with a hybrid intra–/inter-protein attention mechanism to capture intra- and inter-protein interactions. The resulting embeddings and attention matrices can be leveraged for diverse PPI tasks. **B** PPLM-PPI pipeline. For a given sequence pair, PPLM is used to generate the embeddings, intra- and inter-protein attention matrices. These representations are aggregated using max and mean pooling, each followed by a multilayer perceptron to estimate the interaction probability. The final prediction is obtained by averaging the probabilities from the two pooling branches. **C** PPLM-Affinity pipeline. The final transformer block of PPLM is finetuned on binding affinity data, where the resulting embeddings for paired sequences are aggregated via max pooling and passed through two fully connected layers to predict the binding affinity value. **D** PPLM-Contact pipeline. For a given sequence pair, the inter-protein attention matrix from PPLM is integrated with MSA-derived features and monomer distance maps to capture both evolutionary and structural information. These combined features are processed by a series of inter-protein transformer blocks, with each composed of three core modules to predict inter-protein contacts. Each module adopts a parallel architecture to update inter-protein representations. The top interface residues (top 50 displayed) were extracted from the top-ranked predicted contact pairs.

## PPLM-PPI outperforms existing methods in binary protein interaction prediction

To evaluate the effectiveness of PPLM-PPI on protein–protein interaction prediction, we benchmarked it across five species, including *M. musculus*, *D. melanogaster*, *C. elegans*, *S. cerevisiae*, and *E. coli*. PPLM-PPI was compared against four state-of-the-art LM-based PPI predictors, including TUnA[41], ESMDNN-PPI[23], D-SCRIPT[15], and Topsy-Turvy[22]. All methods were installed and executed locally for reproducibility, except ESMDNN-PPI, which we reimplemented following the original publication due to the lack of publicly available source code.

Comprehensive evaluation results, including precision, recall, accuracy, F1-score, area under the receiver operating characteristic curve (AUROC), and area under the precision–recall curve (AUPRC) for all species, are provided in Supplementary Table S1. PPLM-PPI achieved the highest performance on nearly all metrics across all datasets. Since AUPRC is especially informative for imbalanced PPI datasets, we use it as the primary metric. As summarized in Fig. 3A,

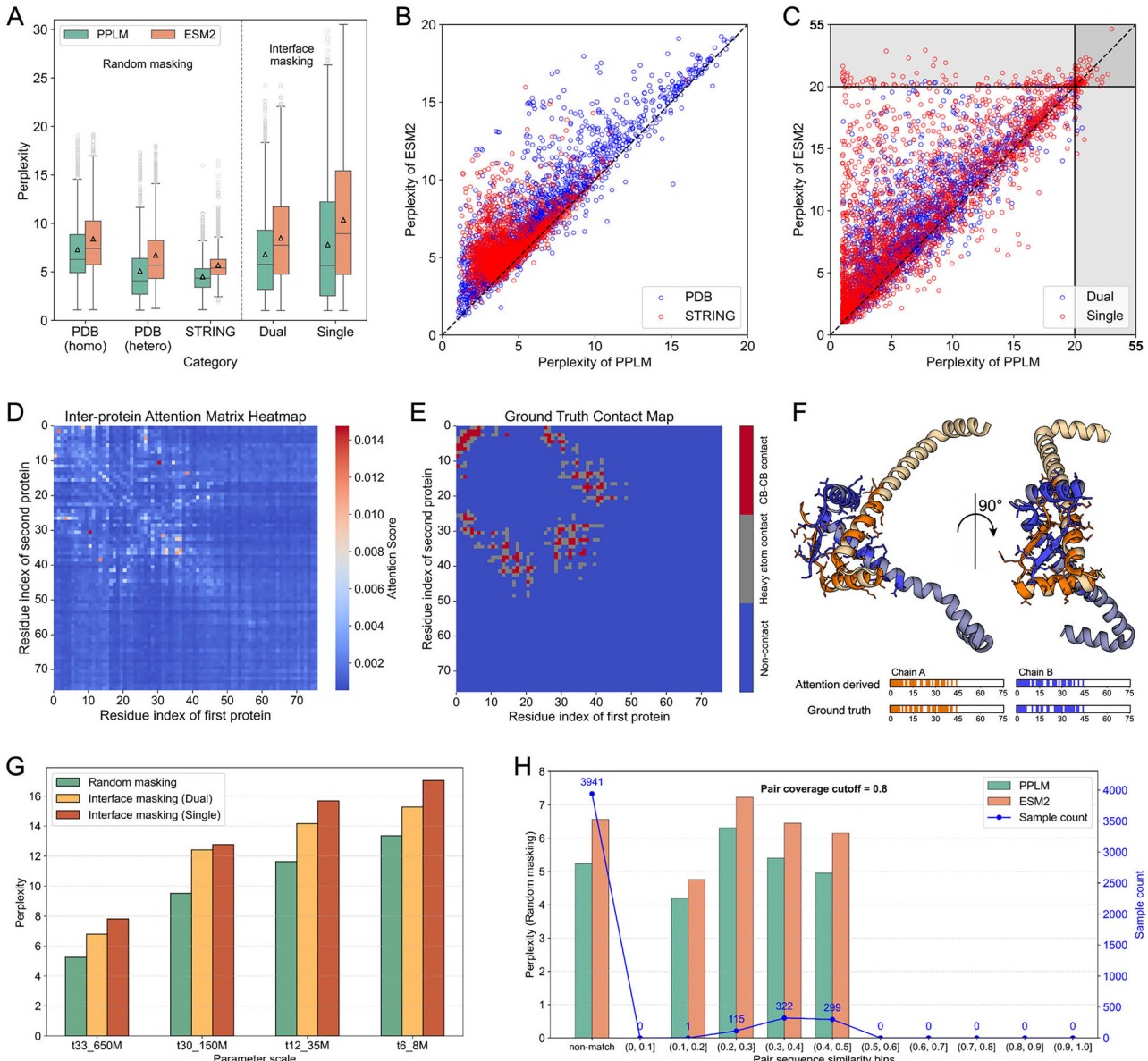

**Fig. 2 | Perplexity of PPLM and ESM2 on protein sequence pairs. A** Perplexity distribution for 1,204 heteromeric PDB pairs, 1003 homomeric PDB pairs, and 2471 STRING-derived pairs under random masking and interface masking strategies (unit of study: one protein sequence pair). Box plots show the mean (triangle), the median (line), the interquartile range (IQR; 25th–75th percentiles), whiskers (non-outlier minima and maxima within 1.5×IQR), and outliers (grey circles). Interface masking is applied only to sequence pairs from the PDB database, with single-chain interface masking used exclusively for heteromeric PDB pairs. **B** Head-to-head comparison of perplexity between PPLM and ESM2 under random masking. **C** Head-to-head comparison under overall interface masking, including both dual-chain and single-chain masking. **D–F** Example of a conserved putative transcription factor from Vibrio cholerae O1 biovar El Tor (PDB ID: 1Y9B). **D** Heatmap of the inter-protein attention matrix generated by PPLM. **E** Ground-truth contact map extracted from the experimental structure. **F** Three-dimensional visualization of the complex, with interface residues identified from the inter-protein attention matrix highlighted. **G** Perplexities of PPLM across different parameter scales. **H** Perplexities of PPLM and ESM2 under varying pairwise sequence similarity to the training set. The number of samples within each similarity bin is indicated. Source data are provided as a Source Data file.

PPLM-PPI achieved AUPRC scores of 0.920, 0.906, 0.883, 0.745, and 0.784 on the five species, corresponding to improvements of 4.0%, 6.2%, 7.2%, 17.6%, and 12.9% over the second-best method, TUnA. Similarly, PPLM-PPI achieved the highest F1-score for all species, surpassing TUnA by 4.8%, 8.6%, 4.8%, 16.9%, and 15.5% (Fig. 3B). Precision–recall curves across species further show that PPLM-PPI consistently dominates other methods across most recall thresholds (Fig. 3C). In addition, the curves of PPLM-PPI are markedly smoother than those of competing methods, indicating superior stability and predictive consistency, particularly in high-precision and high-recall regimes.

An overall analysis across the five species test sets showed that PPLM-PPI demonstrated superior stability, with a mean ± standard deviation of 0.848 ± 0.078, compared to 0.778 ± 0.109 for TUnA, 0.762 ± 0.106 for ESMDNN-PPI, 0.532 ± 0.072 for D-SCRIPT, and 0.548 ± 0.118 for Topsy-Turvy (Supplementary Table S2). Supplementary Table S3 details the statistical comparisons. Relative to TUnA and ESMDNN-PPI, PPLM-PPI improves F1-score by 10.1–10.5% and AUPRC by 9.6–11.9%, with large effect sizes (Cohen's d ≈ 1.742–6.586). Against D-SCRIPT and Topsy-Turvy, the improvements are even larger: 32.5–72.3% (F1-score) and 58.7–60.9% (AUPRC), also with consistently large effect sizes (Cohen's d ≈ 2.062–6.183).

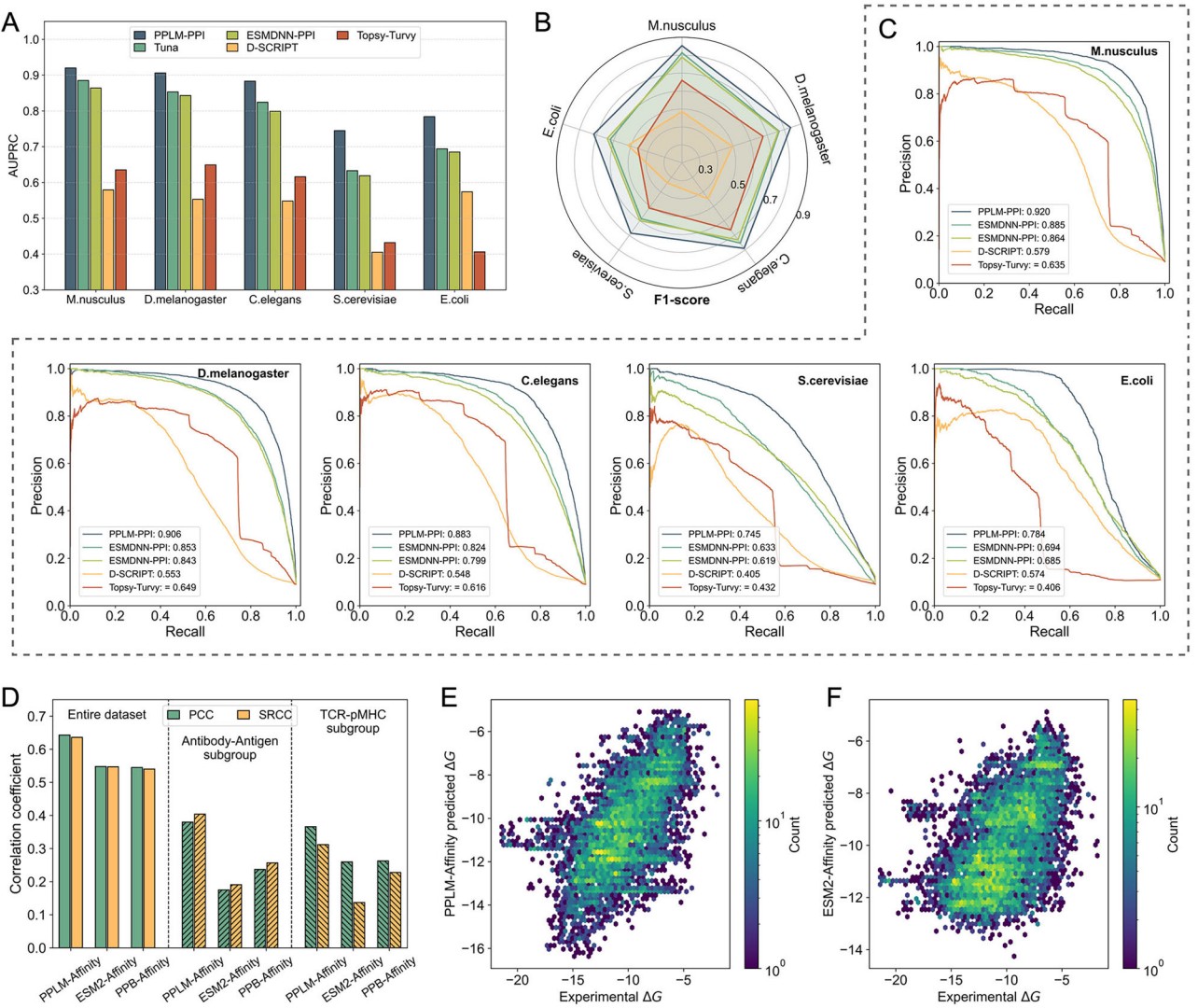

**Fig. 3 | Performance of PPLM-PPI and PPLM-Affinity on protein–protein interaction and binding affinity prediction. A** AUPRC of PPLM-PPI, TUnA, ESMDNN-PPI, D-SCRIPT, and Topsy-Turvy across five species test sets. **B** F1-score of the five PPI prediction methods on the five species test sets. **C** Precision-recall curves of the five PPI prediction methods on the five species test sets. **D** Pearson (PCC) and Spearman (SRCC) correlation coefficients of PPLM-Affinity, ESM2-Affinity, and PPB-

Affinity evaluated on the entire dataset, as well as the antibody–antigen and TCR–pMHC subgroups. **E, F** Head-to-head comparison between experimental and predicted binding affinity for PPLM-Affinity and ESM2-Affinity, respectively. Plots show hexagonal density maps to improve visibility at high point density. Color intensity reflects the local log-transformed point density. Source data are provided as a Source Data file.

To elucidate the contributions of different components, we conducted ablations by modifying pooling modules and selectively removing PPLM-derived features (Supplementary Method S2). Multi-seed results (mean ± 95% CI) are presented in Supplementary Table S4–5 and Fig. S4. Among the individual pooling strategies, mean pooling achieved the highest accuracy, while the combined mean–max pooling strategy produced the most robust and consistent performance across species. In the feature-level analysis, incorporating all three PPLM-derived feature types, including inter-protein attention, intra-protein attention, and embedding, yielded the best results, with embedding features contributing the strongest predictive signal. These findings highlight the importance of both effective pooling and the integration of complementary attention-based and embedding-based representations.

Finally, to assess inductive generalization and evaluate whether shortcut mechanisms might arise in our paired-input setting[42,43], we performed a sequence-similarity–stratified evaluation. Test interactions were grouped by the maximum single-sequence identity of either protein to any human protein in the training set, and all methods were

compared within each identity interval. As detailed in Supplementary Method S3, PPLM-PPI maintains strong performance across all homology ranges and delivers the largest gains over existing methods, demonstrating that its predictive accuracy is not driven by shortcut learning and that the model generalizes robustly to unseen proteins under a strict cross-species inductive setting.

### PPLM-Affinity achieves superior performance in protein binding affinity prediction

Although protein–protein interaction and binding affinity are related, they address distinct questions: PPLM-PPI estimates the probability that two proteins interact, whereas affinity prediction quantifies the continuous strength of binding. To explicitly target quantitative binding strength, we developed PPLM-Affinity by fine-tuning the last transformer block of PPLM to predict binding affinity values. We evaluated PPLM-Affinity on the PPB-Affinity dataset[28], a large and comprehensive benchmark for protein–protein binding affinity prediction. To avoid data leakage, we regrouped the five-fold splits according to structural similarity and performed five-fold cross-

validation on the revised partitions. To assess the effectiveness of language models for this task, we implemented ESM2-Affinity by concatenating the receptor and ligand sequences and fine-tuning the model using the same architecture and training procedure as PPLM-Affinity. In addition, we retrained the structure-based PPB-Affinity model using its released source code under the same cross-validation setting. Detailed fold-wise results, mean ± standard deviation values, and statistical comparisons are summarized in Supplementary Tables S7–S9.

The left panel of Fig. 3D reports the Pearson (PCC) and Spearman (SRCC) correlations between predicted and experimental ΔG values. These two metrics capture complementary aspects of predictive performance: PCC quantifies linear association, whereas SRCC measures rank consistency. PPLM-Affinity achieves mean ± standard deviation PCC and SRCC values of 0.643 ± 0.058 and 0.636 ± 0.082, representing improvements of 17.3% and 16.4% over ESM2-Affinity (0.548 ± 0.061 and 0.547 ± 0.053; Cohen's d = 1.274 and 0.958) and 18.0% and 17.8% over the structure-based PPB-Affinity model (0.545 ± 0.072 and 0.540 ± 0.088; Cohen's d = 1.326 and 1.064). Supplementary Fig. S6 depicts the distribution of absolute errors relative to experimental ΔG values. All three models show smooth, unimodal distributions without marked skewness. PPLM-Affinity exhibits the most concentrated error profile, with the lowest mean absolute error (1.68) and the smallest fitted dispersion (σ = 1.44), compared with higher values for ESM2-Affinity (1.85 and 1.52) and PPB-Affinity (1.85 and 1.63). These trends are consistent with the overall RMSE comparisons, where PPLM-Affinity again attains the lowest value (2.312 ± 0.297) relative to ESM2-Affinity (2.476 ± 0.376) and PPB-Affinity (2.463 ± 0.394). Figure 3E, F further illustrate the predicted versus experimental ΔG values for PPLM-Affinity and ESM2-Affinity, respectively. While both models show positive correlations with experimental measurements, the predictions from PPLM-Affinity cluster more tightly around the diagonal line, indicating improved accuracy, stronger correlations, and reduced systematic bias.

The middle and right panels of Fig. 3D further analyze two biologically important subsets—antibody–antigen and TCR–pMHC interactions—which play central roles in immune recognition and therapeutic development[44,45]. On the antibody–antigen subset, PPLM-Affinity achieves PCC and SRCC values of 0.380 and 0.404, corresponding to improvements of 117.1% and 111.5% over ESM2-Affinity, and 60.3% and 57.2% over PPB-Affinity. On the TCR–pMHC subset, PPLM-Affinity attains PCC and SRCC values of 0.366 and 0.312, yielding gains of 144.0% and 127.7% relative to ESM2-Affinity, and 39.2% and 36.8% relative to PPB-Affinity.

Because PPLM-Affinity and ESM2-Affinity share identical architectures and fine-tuning procedures, the substantial improvements delivered by PPLM-Affinity highlight the advantage of using representations explicitly tailored to protein–protein interactions. Its strong performance relative to the structure-based PPB-Affinity model further indicates that PPLM can capture binding energetics directly from sequence, offering a scalable and structure-independent alternative for affinity prediction. Together, these results suggest that PPLM-Affinity provides improved accuracy and robustness across both general protein complexes and immunologically challenging systems, supporting its potential utility in antibody engineering, TCR optimization, and other sequence-based therapeutic design applications.

### PPLM-Contact outperforms existing methods in inter-protein contact prediction

We evaluated PPLM-Contact on four benchmark datasets: Homodimer300 and Heterodimer99 from DeepInter20, and CASP_Homodimer43 and CASP_Heterodimer20. Comparisons were made against five state-of-the-art methods: PLMGraph-Inter[34], DeepInter[20], CDPred[19], GLINTER[35], and DeepHomo2.0[32]. All baseline methods were installed and executed locally to ensure reproducibility. Inter-protein contact

precision was assessed at multiple cutoffs (top 1, 10, 50, $L/10$, $L/5$, and $L$), where $L$ is the length of the shorter protein chain in the complex. Summary statistics and paired comparisons are provided in Supplementary Tables S10–13, with per-protein results in Supplementary Data 1, 2.

Figure 4A reports the top $L$ contact precision on homodimer test sets using either experimental or AlphaFold2-predicted monomer structures. On Homodimer300 with experimental structures, PPLM-Contact achieves 77.8% precision, substantially outperforming DeepInter (68.9%), PLMGraph-Inter (50.6%), CDPred (63.0%), DeepHomo2.0 (51.4%), and GLINTER (34.5%) by margins of 12.8%, 53.7%, 23.5%, 51.3%, and 125.3%. All comparisons are statistically significant (Wilcoxon signed-rank test $p = 3.63 \times 10^{-21}$ to $1.29 \times 10^{-47}$). Using AlphaFold2-predicted monomer structures, PPLM-Contact maintains strong performance (66.6%) and surpasses the second-best method, DeepInter, by 10.4% ($p = 3.96 \times 10^{-13}$). A similar trend is observed on CASP_Homodimer43. With experimental monomer structures, PPLM-Contact achieves 65.2% precision and improves over baseline methods by 6.0%–157.7%. When using AlphaFold2-predicted monomer structures, the corresponding improvements are 12.9%, 39.5%, 46.3%, 72.3%, and 140.0%, respectively.

Figure 4B presents results on heterodimer test sets. On Heterodimer99, PPLM-Contact achieves 48.9% precision using experimental monomers and 45.1% using AlphaFold2-predicted monomers. These correspond to improvements of 37.0–182.7% over competing methods, all statistically significant. On CASP_Heterodimer20, PPLM-Contact attains 45.6% precision with experimental monomer structures and 40.5% with AlphaFold2 monomers, again outperforming all baseline approaches. Comprehensive statistical analyses, including paired Cohen's d effect sizes and Benjamini–Hochberg false-discovery-rate (FDR) correction across the four test sets, are provided in Supplementary Method S4.

Figure 4C provides head-to-head comparisons between PPLM-Contact and each baseline method across all 434 homodimer and 119 heterodimer proteins using experimental monomer structures. Relative to DeepInter, PPLM-Contact achieves higher/lower top $L$ contact precision on 214/48 homodimer proteins (62.4%/14.0%) and 79/22 heterodimer proteins (66.4%/18.5%). The corresponding higher/lower proportions are 82.5%/12.0% and 63.0%/24.4% for PLMGraph-Inter, 80.8%/12.2% and 65.5%/20.2% for CDPred, 87.2%/8.2% for DeepHomo2.0, and 91.0%/6.7% and 78.2%/13.4% for GLINTER. These results indicate that PPLM-Contact achieves higher performance on a clear majority of targets, while also making it explicit that a non-trivial fraction of predictions is tied rather than strictly lower. Supplementary Fig. S7 presents the corresponding comparisons using AlphaFold2-predicted monomer structures, showing a similar trend in which PPLM-Contact outperforms baseline methods for most targets.

Figure 4D, E summarize the mean and standard deviation of top $L$ precision across three random seeds for PPLM-Contact and all ablation variants. These results show that removing any major network component or feature of the model reduces precision, with the largest performance drops observed when excluding the triangle-multiplication module, PPLM inter-protein attention, MSA features, or monomer distance maps. Detailed ablation analyses and statistical comparisons are provided in Supplementary Method S5. The effect of MSA depth is further examined in Supplementary Method S6.

Figure 5A, B illustrates an example on the EF-hand domain of human RASEF (PDB ID: 2PMY), consisting of two 91-residue identical chains (72 resolved structurally). When relying solely on MSA-derived features and monomer structures, PPLM-Contact without (w/o) PPLM features yields heavily incorrect top $N$ predictions, achieving only 16.7% precision (20.8% for top $L$). In contrast, replacing MSA-derived features with PPLM-generated features allows PPLM-Contact w/o MSA to correctly identify 201 of 257 ground-truth contacts (78.2%) with 100% top $L$ precision. Combining both PPLM and MSA features further

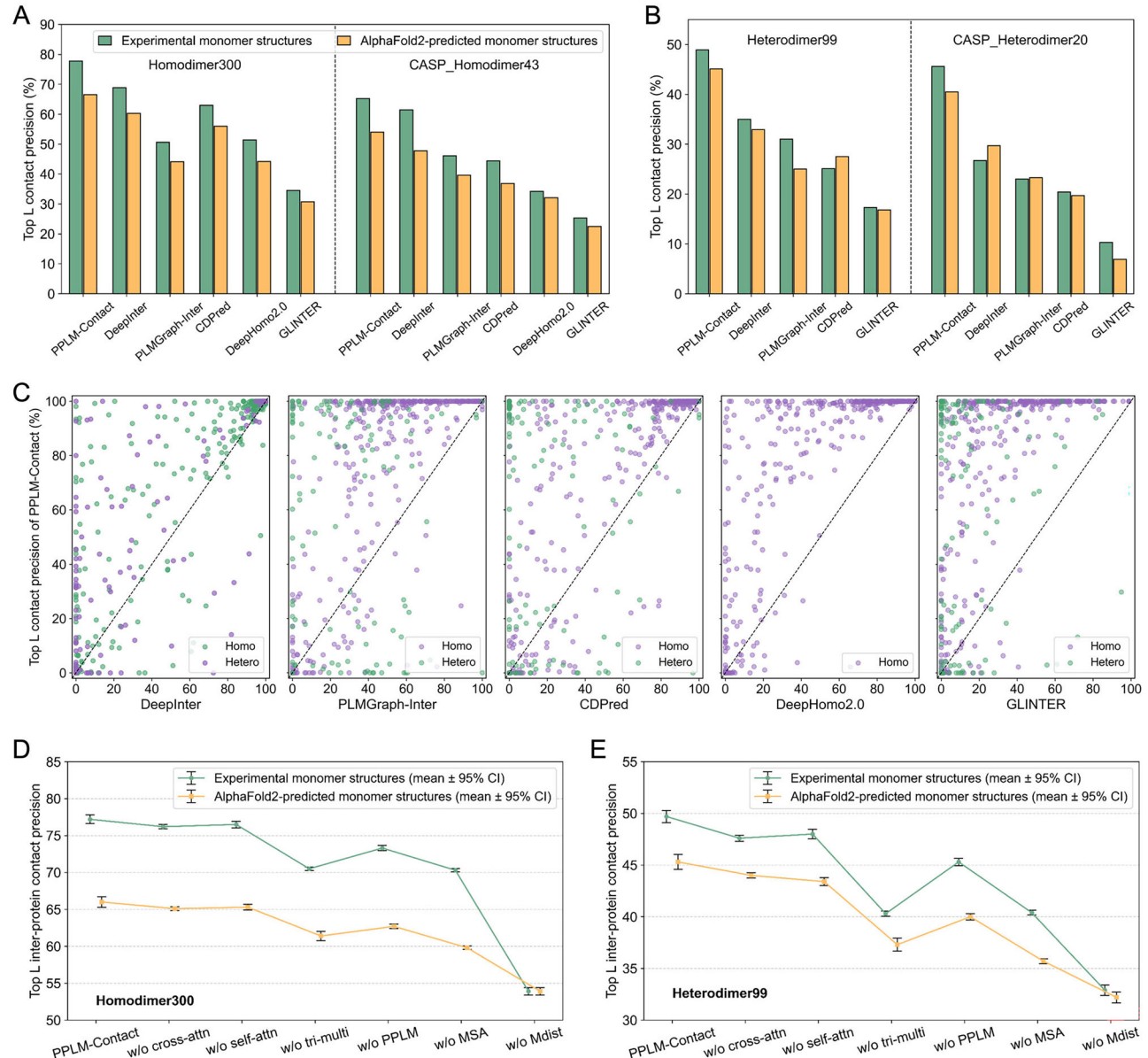

**Fig. 4 | Performance of PPLM-Contact and the comparison methods in inter-protein contact prediction. A** Top *L* contact precision of PPLM-Contact, DeepInter, CDPred, DeepHomo2.0, and GLINTER on the Homodimer300 and CASP_Homodimer43 test sets using both experimental monomer structures and AlphaFold2-predicted monomer structures. **B** Top *L* contact precision of PPLM-Contact, DeepInter, CDPred, and GLINTER on the Heterodimer99 and CASP_Heterodimer20 test sets using both experimental monomer structures and

AlphaFold2-predicted monomer structures. **C** Head-to-head comparison of top *L* contact precision between PPLM-Contact and other methods on the 343 homodimer and 119 heterodimer proteins. **D**, **E** Mean top L contact precision and 95% confidence intervals (CI) of PPLM-Contact and its ablation variants across three independent runs with different random seeds on the Homodimer300 (300 complexes) and Heterodimer99 (99 complexes) test sets, respectively (unit of study: one complex). Source data are provided as a Source Data file.

improves performance to 81.3% top *N* precision while maintaining 100% top *L* precision. This example illustrates the strong ability of PPLM-derived features to capture geometric and interface-level spatial patterns critical for accurate inter-protein contact prediction.

### Enhanced inter-protein contact prediction using predicted complex structures

Building on the performance of PPLM-Contact, we next examined whether incorporating structural information from modern complex-prediction models could further enhance inter-protein contact prediction. Recent advances in multimer structure prediction now provide highly accurate models of protein complexes, capturing detailed geometric organization at interfaces that is difficult to infer from sequence alone. These predicted structures therefore offer an opportunity to

supply language-model pipelines with precise spatial context. To leverage this, we developed PPLM-Contact2, an enhanced variant of PPLM-Contact that integrates inter-protein distance maps extracted from predicted complex structures. For evaluation, PPLM-Contact2 was applied separately to inter-protein distance maps derived from AlphaFold2.3, AlphaFold3, and DMFold predicted complex structures, and the corresponding predictions were ensembled as the final output. We benchmarked PPLM-Contact2 against AlphaFold2.3[38], AlphaFold3[39], and DMFold[40] across all 434 homodimer and 119 heterodimer proteins (Supplementary Tables S14, S15; Supplementary Data 3, 4).

Figure 5C, D summarize the top *L* precision of all general contact-prediction methods (using AlphaFold2-predicted monomer structures) and the complex structure-based methods (AlphaFold2.3, AlphaFold3, DMFold, and PPLM-Contact2) on the 434 homodimer and

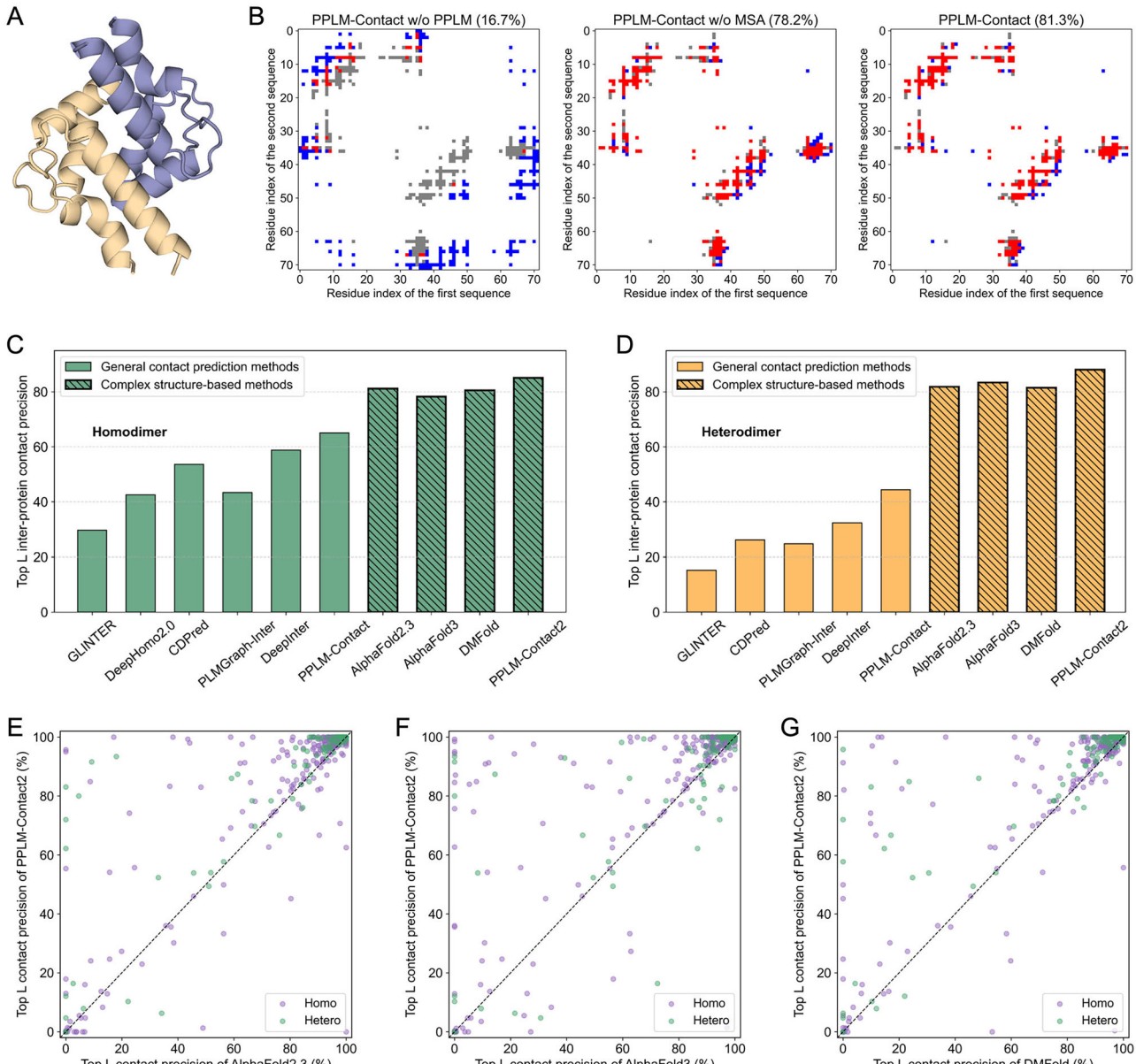

**Fig. 5 | Inter-protein contact prediction by PPLM-Contact and PPLM-Contact2.**
A Example of inter-protein contact prediction for the EF-hand domain of human RASEF (PDB ID: 2PMY). **B** Contact maps for 2PMY predicted by PPLM-Contact w/o PPLM, PPLM-Contact w/o MSA, and the full PPLM-Contact models, respectively. Each contact map shows the alignment between ground-truth contacts and top $N$ predicted contacts, where $N$ equals the number of ground-truth contacts. Grey dots representing ground-truth contacts, red dots indicate true positive predictions, and blue dots denote false positive predictions. The number in the parentheses denotes the top $N$ contact precision. **C**, **D** Top $L$ contact precision of general contact-prediction methods (GLINTER, DeepHomo2.0, CDPred, PLMGraph-Inter, DeepInter, and PPLM-Contact) and complex structure–based approaches (Alpha-Fold2.3, AlphaFold3, DMFold, and PPLM-Contact2) on 343 homodimers and 119 heterodimers, respectively. **E**–**G** Pairwise comparisons of top $L$ contact precision between PPLM-Contact2 and AlphaFold2.3, AlphaFold3, and DMFold. Source data are provided as a Source Data file.

119 heterodimer proteins. As expected, the integration of complex-structure features allows PPLM-Contact2 to markedly improve upon PPLM-Contact, increasing top $L$ precision from 65.0% to 85.1% on homodimers and from 44.3% to 88.0% on heterodimers. Overall, all complex structure-based methods outperform general contact-prediction methods, consistent with the fact that AlphaFold leverages deep 3D structure-prediction pipelines, extensive geometric reasoning, and large-scale template and MSA signals, thereby providing much richer structural context than purely sequence- or MSA-based contact predictors. Compared with structure-modeling approaches, PPLM-Contact2 achieves improvements of 4.8% and 7.6% over Alpha-Fold2.3, 8.7% and 5.6% over AlphaFold3, and 5.6% and 8.1% over

DMFold on homodimers and heterodimers. All improvements are statistically significant (Supplementary Table S15).

Head-to-head comparisons across the full benchmark (Fig. 5E–G) further highlight the advantage of PPLM-Contact2. Relative to Alpha-Fold2.3, PPLM-Contact2 achieves higher/lower top $L$ precision on 207/44 homodimer proteins (60.3%/12.8%) and 75/10 heterodimer proteins (63.0%/8.4%). The corresponding higher/lower proportions are 62.1%/11.7% and 56.3%/18.5% for AlphaFold3, and 62.7%/11.1% and 64.7%/10.1% for DMFold. Supplementary Fig. S9 presents the distributions of precision, recall, and F1-score across all test proteins. Compared with the structure-based modeling methods, PPLM-Contact2 shows a clear shift toward higher ranges for all three metrics. It achieves an F1-score of

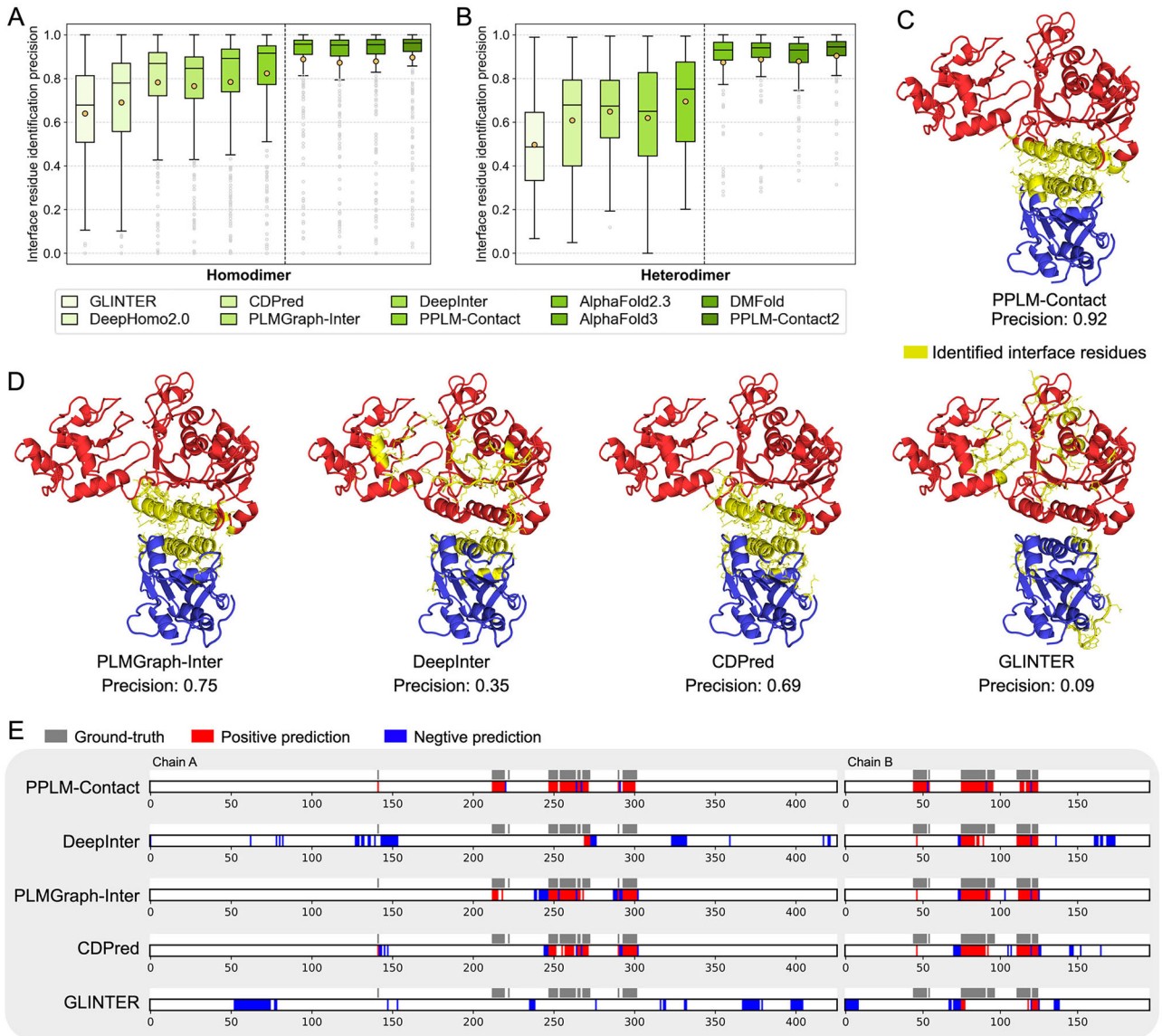

**Fig. 6 | Comparative analysis of protein–protein interface residue identification. A, B** Precision of interface-residue identification by general contact-prediction methods (GLINTER, DeepHomo2.0, PLMGraph-Inter, CDPred, DeepInter, and PPLM-Contact) and complex structure–based approaches (AlphaFold2.3, AlphaFold3, DMFold, and PPLM-Contact2) on 343 homodimers and 119 heterodimers, respectively (unit of study: one complex). Box plots show the mean (black circle with yellow fill), the median (line), the interquartile range (IQR; 25th–75th percentiles), whiskers (non-outlier minima and maxima within 1.5×IQR), and outliers (grey circles). **C, D** Crystal structure of the DNMT3A–DNMT3L complex (PDB ID: 4U7P), with identified interface residues from inter-protein contact-prediction methods highlighted in yellow and shown in stick representation. **E** Residue-level annotation maps for 4U7P, where grey bars indicate ground-truth interface residues, red bars denote true positives, and blue bars denote false positives. Source data are provided as a Source Data file.

0.780, exceeding AlphaFold2.3 (0.748), AlphaFold3 (0.739), and DMFold (0.743) by 4.3%, 5.5%, and 5.0%, respectively, highlighting its robustness and consistency in inter-protein contact prediction.

Together, these results demonstrate that even with the remarkable advances in end-to-end complex structure prediction, additional gains can be achieved by integrating predicted structural models with interaction-aware language-model features. The resulting hybrid framework not only improves contact-map prediction but also provides more reliable localization of inter-chain binding sites. This integrated information may serve as spatial constraints for downstream quaternary structure assembly and offers a principled strategy for further improving protein complex modeling.

### PPLM-Contact enhances interface residue recognition

Beyond predicting inter-protein contacts, accurately identifying interface residues is essential for understanding complex formation,

signaling, and functional regulation of protein assemblies[46]. We therefore evaluated the interface-recognition performance of all general contact-prediction methods (using AlphaFold2-predicted monomer structures; corresponding results using experimental monomer structures are provided in Supplementary Table S16 and Fig. S10) as well as complex structure-based approaches on the 434 homodimer and 119 heterodimer proteins. Interface residues were defined as those with any heavy atom within 8 Å of a heavy atom from the partner chain in the experimental complex structure. For each target, we assessed the top $N$ predicted residues, where $N$ equals the number of ground-truth interface residues, independent of their specific contact partners. Figure 6A, B report the interface identification precision for all methods (Supplementary Data 5, 6).

Among general contact-prediction methods, PPLM-Contact achieves the highest average precision, reaching 0.824 for homodimers and 0.695 for heterodimers. These values represent

improvements of 5.0% and 12.1% over DeepInter, 7.6% and 7.1% over PLMGraph-Inter, 5.3% and 14.2% over CDPred, 19.3% over DeepHomo2.0 (homomers), and 28.7% and 39.7% over GLINTER. PPLM-Contact also attains the highest median precision (0.916 and 0.752), indicating consistently strong predictions across targets. All improvements are statistically significant (Supplementary Table S17). Integrating complex-structure features further boosts performance: PPLM-Contact2 mark-edly improves upon PPLM-Contact, increasing average and median precision to 0.897 and 0.962 for homodimers and 0.904 and 0.945 for heterodimers. Compared with structure-based methods, PPLM-Contact2 achieves the highest average and median precision across both homodimer and heterodimer test sets. Comprehensive statistical analyses confirm that these improvements are significant (Supplementary Table S18). The head-to-head comparisons between PPLM-Contact and general contact-prediction methods, as well as between PPLM-Contact2 and complex structure modelling approaches, further show that both PPLM-Contact and PPLM-Contact2 outperform competing methods on a clear majority of targets, underscoring their robustness and broad applicability (Supplementary Figs. S11, 12).

Figure 6C, D present an illustrative example using the crystal structure of the DNMT3A–DNMT3L complex (PDB ID: 4U7P), consisting of 426 and 197 experimentally resolved residues in two distinct chains. Interface residues identified by each method are highlighted on the 3D structure, and Fig. 6E compares ground-truth and predicted residues along sequence positions. PPLM-Contact accurately identifies 40 of 43 (93.0%) interface residues on chain A and 40 of 44 (90.9%) on chain B, substantially outperforming baseline approaches. In contrast, Dee-pInter identifies only 4 (9.3%) and 26 (59.1%) residues; PLMGraph-Inter identifies 34 (79.1%) and 31 (70.5%), CDPred identifies 29 (67.4%) and 31 (70.5%), and GLINTER misidentifies all interface residues on chain A and recovers only 8 (18.2%) on chain B. Detailed inspection reveals that baseline methods tend to overpredict residues near the N-terminus of chain A and the C-terminus of chain B, whereas PPLM-Contact pro-duces a more balanced residue distribution along both chains, yielding an overall precision of 0.92, markedly higher than DeepInter (0.35), PLMGraph-Inter (0.75), CDPred (0.69), and GLINTER (0.09).

These results demonstrate that PPLM-Contact achieves superior performance in interface residue recognition. This improvement can be primarily attributed to two key innovations. First, PPLM effectively captures the PPI signals that complement co-evolutionary information from MSA, providing reliable features for the deep learning model. Second, the inter-protein specific transformer network, designed with a parallel architecture, enables efficient learning of geometric inter-actions between residues across protein interfaces.

## Discussion

Computational modeling of protein–protein interactions (PPIs) is essential for elucidating cellular mechanisms and accelerating ther-apeutic discovery. Although protein language models have recently emerged as scalable tools for PPI analysis, they often fail to capture inter-protein context explicitly. To address this limitation, we intro-duced PPLM, a paired-sequence language model designed to learn inter-protein co-representation through a dedicated attention mechanism. Together with task-specific downstream architectures, PPLM supports interaction prediction, binding affinity estimation, and inter-protein contact prediction. Trained on a large composite dataset, PPLM achieves substantially lower perplexity than ESM2 at protein interfaces and consistently improves performance across the three downstream tasks. When combined with predicted structural features, the extended PPLM-Contact2 model also outperforms state-of-the-art structure-based approaches, including AlphaFold2.3, AlphaFold3, and DMFold.

Despite its strong performance, PPLM has several limitations that warrant further investigation. Its training relies on a composite sequence-pair dataset that, while diverse, may not fully capture the breadth of protein interactions across organisms and cellular states. In

particular, transient, weak, or condition-dependent interactions remain underrepresented. A representative failure case is shown in Supplementary Fig. S13, involving a nanobody–antigen complex with a very small, loop-mediated interface and weak co-evolutionary signal. Although PPLM-derived features partially compensate for the lack of MSA or structural information, they remain insufficient to recover these small, flexible, and weakly co-evolving interfaces, highlighting an intrinsic limitation of sequence-based contact prediction.

While this study focuses on three core PPI-related tasks, PPLM offers broader opportunities for integration with structural and func-tional modeling frameworks. Combining its interface-aware repre-sentations with AI-driven complex prediction tools such as AlphaFold3 may further improve the modeling of multimeric assemblies. Likewise, the strong performance of PPLM-Affinity on antibody–antigen and TCR–pMHC interactions suggests potential applications in CDR modeling and therapeutic design. By capturing fine-grained interface features, PPLM may provide a foundation for more accurate modeling of binding interactions and for guiding the development of next-generation biologics.

## Methods
### Datasets
To train PPLM, we constructed a composite dataset of protein interac-tion sequence pairs by integrating protein structure complexes released before January 1, 2024 from the Protein Data Bank (PDB)[47] and interac-tion sequences from the STRING[48] database. After filtering for physical interfaces and removing redundancy through pair-level clustering using MMseqs2[49], together with a custom clustering procedure, the dataset comprised 25,245 heteromeric clusters and 23,082 homomeric clusters from PDB, as well as 629,045 clusters from STRING, covering over 3.3 million protein sequence pairs. In total, 672,372 clusters were used to train the language model, and 4678 single-pair clusters that were non-redundant to any training pairs were held out for validation (Supple-mentary Method S7). During the training of PPLM, sequence pairs were sampled from the PDB and STRING clusters at a 1:2 ratio.

To train and evaluate PPLM-PPI for PPI prediction, we adopted benchmark datasets from D-SCRIPT[15] spanning six species: *H. sapiens*, *M. musculus*, *D. melanogaster*, *C. elegans*, *S. cerevisiae*, and *E. coli*. Given the relative rarity of true PPIs, each dataset maintains a 10:1 ratio of negative to positive pairs. To ensure data integrity, we identified and removed duplicate, erroneous, and invalid samples arising from the random negative sampling process (Supplementary Method S8). The PPLM-PPI model was trained and validated on the *H. sapiens* dataset and tested on the remaining five species datasets. Detailed dataset statistics are listed in Table S19.

For protein–protein binding affinity prediction, we adopted the PPB-Affinity dataset[28], a large curated resource compiled from multiple public affinity databases. The dataset provides experimentally deter-mined binding affinities, crystal structures, mutation profiles, and annotations of receptor and ligand chains, comprising 12,052 inter-action samples from 3,027 distinct PDB entries, including subgroups such as antibody–antigen and TCR–pMHC complexes. To prevent potential homology-based leakage, all PDB entries were clustered with US-align[50] using a complex-level TM-score cutoff of 0.8. Complexes with different numbers of chains were clustered separately to maintain meaningful structural comparability. All complexes within each structural cluster were assigned to the same fold in the five-fold cross-validation scheme. The resulting five folds contained 606, 606, 605, 605, and 605 PDB entries, respectively, and the list of PDB IDs for each fold is provided in Supplementary Data 7.

For inter-protein contact prediction, we adopted the same data-sets used in DeepInter[20] to ensure fair comparison with existing methods. These datasets consist of non-redundant homodimers and heterodimers curated from the PDB. The training set includes 3,504 homodimers and 1,881 heterodimers, while the validation set

comprises 296 homodimers and 96 heterodimers. The test set contains 300 homodimers (Homodimer300) and 99 heterodimers (Heterodimer99). We also constructed two independent test sets comprising 43 homodimer targets (CASP_Homodimer43) and 20 heterodimer targets (CASP_Heterodimer20) collected from CASP13 to CASP16 to further assess model robustness and real-world applicability (Supplementary Data 1 and 2). Two sets of PPLM-Contact models were trained separately for homodimers and heterodimers: the homodimer model was trained exclusively on homodimers, whereas the heterodimer model was trained on the full combined training set.

## Protein pair language model

PPLM is a pretrained protein pair language model that can capture potential protein interaction information by providing informative representations of protein pairs (Fig. 1A). Given a protein sequence pair as input, the sequences are first tokenized independently and then concatenated with special beginning-of-sequence (BOS) and end-of-sequence (EOS) tokens to explicitly mark the boundaries of each chain. The concatenated initial representations will be used as the input to a stack of 33 serially connected transformer blocks, with each block consisting of a tailored multi-head attention module and a feed-forward network. The preprocessing workflows for paired sequences in PPLM and in ESM2's single-chain approach are illustrated in Supplementary Fig. S15.

In the multi-head attention module, we introduce a hybrid intra–/inter-protein attention mechanism to capture both intra- and inter-protein residue interactions. To distinguish these two interaction types, two separate attention matrices are computed: one with rotary positional embeddings (RoPE)[37] for intra-chain residue pairs, and another without positional encoding (ROW) for inter-protein residue pairs. These two matrices are then integrated into a full-size attention matrix. To enhance PPLM's ability to capture diverse inter-protein interaction patterns, we introduce a set of learnable weights ($W_{lh}$) across transformer layers ($l$) and attention heads ($h$), applied specifically to the inter-protein attention components. Additionally, an inter-protein binary mask ($M_{ij}$) is employed to distinguish inter-protein from intra-chain residue pairs ($i, j$) and to better guide the model's focus on inter-protein interactions. Thus, the full attention matrix is defined as:

$$A_{lhij} = W_{lh} \cdot M_{ij} \cdot A_{lhij}^{row} + \left(1 - M_{ij}\right) \cdot A_{lhij}^{rope} + M_{ij} \tag{1}$$

where $A_{lhij}^{row}$ and $A_{lhij}^{rope}$ represent the attention matrices computed without and with RoPE, respectively, and $M_{ij}$ is a binary mask matrix with value 1 for inter-protein residue pairs and 0 for intra-protein pairs.

The masking strategy plays a pivotal role in masked language model training. In PPLM, we adopt distinct masking strategies for sequence pairs derived from different data resources. For high-quality sequence pairs from the PDB database, greater emphasis is placed on interface residues: 30% of interface residues are randomly masked, compared to 15% of non-interface residues. In contrast, for sequence pairs from the STRING database, a uniform masking strategy is applied, where 15% of residues in each sequence are randomly selected for masking. To accommodate GPU memory limitations, sequence pairs with full lengths exceeding 1024 residues are cropped. For sequence pairs from the PDB database, cropping is biased toward retaining fragments that include most interface residues. For sequence pairs from the STRING database, continuous fragments are cropped from each sequence, with fragment lengths proportional to the original sequence lengths.

The final objective function is a weighted average of cross-entropy losses computed separately over masked residues in each sequence:

$$\mathcal{L}_{MLM} = -\frac{1}{|M_1|}\sum_{i \in M_1} \log p(x_i^{(1)}|x \backslash M) - \frac{1}{|M_2|}\sum_{i \in M_2} \log p(x_i^{(2)}|x \backslash M) \tag{2}$$

where $x^{(1)}$ and $x^{(2)}$ represent the amino acid sequences of the two proteins, and $x = (x^{(1)}, x^{(2)})$ represents all residues from both sequences. $M_1$ and $M_2$ denote the sets of masked residue positions in each sequence, and $M = M_1 \bigcup M_2$ denotes all masked positions across both sequences. The notation $x \backslash M$ refers to the unmasked residues from both sequences.

PPLM was initialized from the 650M-parameter ESM2 model[6] and trained using four NVIDIA A100 GPUs with a gradient accumulation step of 32, for a total of 50,000 steps. Optimization was performed using the AdamW optimizer with exponential decay rates $\beta_1 = 0.9$ and $\beta_2 = 0.98$. The learning rate was linearly warmed up to $1 \times 10^{-6}$ over the first 2000 steps, followed by a linear decay to $5 \times 10^{-7}$ over the remainder of the training.

## Interaction prediction using PPLM

Building upon PPLM, we developed PPLM-PPI for protein–protein interaction prediction by leveraging the protein pair representations generated by PPLM (Fig. 1B). Given a pair of input protein sequences, PPLM is first used to extract several features: the embeddings of each sequence, the intra-protein attention matrices for both sequences, and the inter-protein attention matrix between sequences. For each feature, max pooling and mean pooling are applied independently along the sequence dimension, forming two parallel representation branches. Each pooled representation is projected into a unified dimensional space through a dedicated linear layer and subsequently passed into its own multilayer perceptron (MLP) followed by a sigmoid activation to produce an interaction probability. Each MLP consists of five linear layers, where the first four layers are followed by layer normalization and a ReLU activation. The final interaction score is obtained by averaging the outputs of the two branches. The dimensionality parameters used at each transformation step are summarized in Tables S20–21.

PPLM-PPI was trained using 10-fold cross-validation on the *H. sapiens* dataset. The data was randomly divided into ten subsets, ensuring a balanced ratio of positive and negative samples in each. For each fold, one subset was held out for validation, while the remaining nine were used for training. Models were trained for 12 epochs per fold, and the checkpoint with the highest AUPRC on the validation set was selected. The five best-performing models across all folds were then ensembled to generate final predictions during inference on the test sets. Training was performed on a single NVIDIA A100 GPU with a batch size of 32. The model was optimized using the AdamW optimizer with a learning rate of $5 \times 10^{-5}$. The loss function was binary cross-entropy between predicted probabilities and ground-truth interaction labels, defined as:

$$\mathcal{L}_{PPI} = -\frac{1}{N}\sum_{i=1}^{N}\left[y_i \log p_i + (1 - y_i) \log(1 - p_i)\right] \tag{3}$$

where $y = 0$ *or* 1 denotes the ground-truth interaction label, with 1 for positive and 0 for negative, $p \in (0, 1)$ is the predicted probability of interaction, and $N$ is the batch size.

## Binding affinity prediction using PPLM

We further developed a sequence-based method, PPLM-Affinity, for protein–protein binding affinity prediction by fine-tuning the final layer (i.e., the last transformer block) of PPLM on a curated affinity dataset (Fig. 1C). Given the limited size of the dataset—many samples involve mutations and include multi-chain receptor or ligand proteins, especially within antibody–antigen and TCR–pMHC subgroups—we fine-tuned the final layer to better adapt PPLM's contextual representations for capturing sequence-level determinants of molecular interaction strength. The full-length embeddings output by the fine-tuned PPLM are aggregated along the sequence dimension using max pooling, and the resulting vector is passed through two fully

connected layers with an intermediate ReLU activation to predict the binding affinity value. In cases where the receptor or ligand consists of multiple chains, their sequences are concatenated into a single sequence prior to input into PPLM.

PPLM-Affinity was trained and evaluated using five-fold cross-validation. To prevent data leakage, samples sharing the same PDB ID were grouped into the same fold. Each fold was trained for 15 epochs on a single NVIDIA A100 GPU. Optimization was performed using the AdamW optimizer with a learning rate of $1 \times 10^{-4}$, and the loss function was the mean squared error (MSE) between predicted and experimentally determined binding affinities:

$$\mathcal{L}_{\text{Affinity}} = \frac{1}{N} \sum_{i=1}^{N} (\hat{y}_i - y_i)^2 \tag{3}$$

where $y_i$ and $\hat{y}_i$ represent the ground-truth and predicted binding affinity, respectively; $N$ is the batch size, which equals 1 in our setting.

### Inter-protein contact prediction using PPLM

In addition to predicting interaction and binding strength, we further developed PPLM-Contact for inter-protein contact prediction and identified interaction interfaces. As illustrated in Fig. 1D, PPLM-Contact consists of two main components: (1) feature extraction from PPLM, MSA, and monomer structures; and (2) an inter-protein transformer architecture incorporating parallel triangle multiplication, cross-attention and self-attention mechanisms.

For each input protein pair, features are extracted from three sources. First, PPLM is used to create the inter-protein attention matrix, which captures cross-chain dependencies and serves as a key inter-protein feature. Second, MSA for each monomer sequence is collected by searching against the UniRef30_2021_03[51] database using HHblits[52] with hyperparameters set to '-e 1e-3 -cov 0.4 -id 99'. For heterodimers, paired MSA are constructed based on species annotation[40], while for homodimers, the monomer MSA is directly reused as the paired MSA. A variety of features are then extracted from both monomer and paired MSA, including position-specific scoring matrices (PSSMs)[53], row scores, and average product corrections from direct coupling analysis (DCA)[54], as well as MSA embeddings and attention matrices derived from ESM-MSA-1b[5]. Lastly, the monomer distance maps are extracted from either experimentally determined (when available) or predicted monomer structures (by AlphaFold2[55]). The dimensions of all features are detailed in Table S22. For the enhanced version, PPLM-Contact2, inter-protein distance map extracted from the complex structure predicted by AlphaFold2.3 is incorporated as an additional feature during training.

In the inter-protein transformer architecture, two ResNet modules are used to independently encode intra-protein and inter-protein features before passing into the inter-protein transformer blocks (Fig. S16). The intra-protein features of both sequences are processed by a shared intra-protein ResNet module with tied parameters. This is followed by 12 inter-protein transformer blocks, each containing two parallel triangle multiplication modules, two parallel cross-attention modules, two parallel self-attention modules, and a transition layer (see middle box of Fig. 1D).

The triangle multiplication module is designed to update the inter-protein pair representation by integrating information from both intra- and inter-protein pairs (Fig. S17). Specifically, for a given inter-protein residue pair $z_{ij}$, the module aggregates signals from an intermediate residue $n$, utilizing intra-chain interactions $r_{in}$ and cross-chain interactions $z_{nj}$, while ensuring geometric triangle constraints are maintained. A parallel operation is simultaneously applied to both proteins, allowing inter-protein features to be updated using intra-protein information from each chain. The detailed update equations are:

$$z'_{ij} = \mathcal{L}(z_{ij}) \odot \sigma(\mathcal{L}(z_{ij})); r'_{in} = \mathcal{L}(r_{in}) \odot \sigma(\mathcal{L}(r_{in})); g_{ij} = \sigma(\mathcal{L}(z_{ij})) \tag{4}$$

$$\tilde{z}_{ij} = g_{ij} \odot \varphi\left(\sum_{n=1}^{L} r'_{in} \cdot z'_{ij}\right) \tag{5}$$

where $z_{ij}$ and $r_{ik}$ are the inter- and intra-protein pair representations, respectively, $\tilde{z}_{ij}$ is the updated $z_{ij}$ based on one protein; $\mathcal{L}$ presents a linear transformation, $\sigma$ the sigmoid activation, and $\varphi$ a LayerNorm followed by a linear transformation. $L$ is the monomer sequence length. Two triangle multiplication operations, with one per protein, are performed in parallel, and the resulting outputs are combined to update the original $z_{ij}$.

The cross-attention module is designed to update the inter-protein pair representation $z_{ij}$ by attending to all intra-protein pair $r_{in}$ that share residue $i$ (or $r_{jn}$ that share residue $j$), thereby capturing rich contextual information from the corresponding monomer protein (Fig. S18). In this mechanism, the inter-protein pair representation acts as the query, while the intra-protein pairs serve as the key and value in the attention operation, i.e.,

$$q_{ij}^h = \mathcal{L}(z_{ij}); k_{in}^h = \mathcal{L}(r_{in}); v_{in}^h = \mathcal{L}(r_{in}); b_{ij}^h = \mathcal{L}(z_{ij}); g_{ij}^h = \sigma(\mathcal{L}(z_{ij})) \tag{6}$$

$$a_{ijk}^h = \text{softmax}\left(\frac{1}{\sqrt{c}} q_{ij}^h \cdot k_{in}^h + b_{ij}^h\right) \tag{7}$$

$$\tilde{z}_{ij} = \mathcal{L}\left(g_{ij}^h \odot \sum_{n=1}^{L} a_{ijk}^h \cdot v_{in}^h\right) \tag{8}$$

Two parallel cross-attention operations are performed across two proteins according to the equations above, and the resulting updates are then combined to update the inter-protein pair representation.

A self-attention module is applied to further update the inter-protein representation from its own context (Fig. S19). Specifically, it attends over all inter-protein pairs $z_{im}$ that share residue $i$ (or $z_{jm}$ that share residue $j$) to update $z_{ij}$, thereby enabling long-range information flow across the protein interface. During this process, intra-protein distances extracted from monomer structure are transformed using a Gaussian kernel and used to modulate the attention weights[20]. The detailed formulation is:

$$q_{ij}^h = \mathcal{L}(z_{ij}); k_{im}^h = \mathcal{L}(z_{im}); v_{im}^h = \mathcal{L}(z_{im}); g_{ij}^h = \sigma(\mathcal{L}(z_{ij})) \tag{9}$$

$$a_{ijk}^h = \text{softmax}\left(\frac{1}{\sqrt{c}} q_{ij}^h \cdot k_{im}^h \cdot d_{im}\right) \tag{10}$$

$$\tilde{z}_{ij} = \mathcal{L}\left(g_{ij}^h \odot \sum_{m-1}^{L} a_{ijk}^h \cdot v_{im}^h\right) \tag{11}$$

where $d_{im}$ denotes the intra-protein distance transformed by the Gaussian kernel function. Parallel self-attention operations are performed independently along two monomer protein dimensions and then aggregated to update the inter-protein pair representation $z_{ij}$. Finally, a transition layer consisting of two linear transformations is applied at the end of each transformer block.

PPLM-Contact was trained on a single NVIDIA A100 GPU for a total of 100 epoch using the AdamW optimizer. The initial learning rate was set to $1 \times 10^{-3}$ and decayed by a factor of 0.98 after each epoch. To handle long sequences during training, monomer proteins longer than

256 residues were cropped by randomly selecting a contiguous 256-residue fragment that contained the highest number of interface residues. Inter-protein residue pairs were defined as contacts if the minimum distance between their heavy atoms was less than 8Å[56]. The top five models with the highest top $L$ inter-protein contact precision on the validation set were then used for ensemble prediction during inference. Given the severe class imbalance between contact and non-contact residue pairs, we employed the Focal Loss function[57] to mitigate the influence of dominant non-contact pairs:

$$\mathcal{L}_{\text{Contact}} = -\frac{1}{L_1 L_2} \sum_{i=1}^{L_1} \sum_{j=1}^{L_2} \alpha \left(1 - p_{ij}\right)^{\gamma} \log(p_{ij}) \qquad (12)$$

where $p_{ij}$ is the predicted contact probability; $\alpha = 0.25$ is a balance factor, and $\gamma = 1.5$ is a focusing factor; $L_1$ and $L_2$ are the protein lengths of the two monomer proteins, respectively.

## Reporting summary

Further information on research design is available in the Nature Portfolio Reporting Summary linked to this article.

## Data availability

The dataset used in this study is deposited in our GitHub repository: https://github.com/junliu621/PPLM/tree/main/data/. Protein pairs used to train PPLM are collected from the Protein Data Bank (PDB, https://www.rcsb.org/) and the STRING database (https://string-db.org/). The interaction dataset for PPLM-PPI is sourced from D-SCRIPT (https://github.com/samsledje/D-SCRIPT/), while the binding affinity dataset for PPLM-Affinity is taken from PPB-Affinity (https://github.com/ChenPy00/PPB-Affinity). The Uniref30_2021_03 database is available at https://www.user.gwdguser.de/~compbiol/uniclust/. Source data are provided with this paper.

## Code availability

The PPLM webserver and source codes, including the language model, interaction prediction, binding affinity prediction, and inter-protein contact prediction, are freely accessible at https://zhanggroup.org/PPLM/. The source codes are also available on GitHub at https://github.com/junliu621/PPLM under the MIT License. The publication release is deposited on Zenodo at https://zenodo.org/records/18256392[58].

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

## Acknowledgements

We thank Dr. Yang Li for the discussions. This work was supported in part by the Ministry of Education, Singapore (T1251RES2309 and T2EP20125-0039 to Y.Z.), the Agency for Science, Technology and Research (A*STAR), Singapore (IAF-PP H25J6a0034 to Y.Z.), the National Research Foundation, Singapore (NRF-CRP33-2025-0048 to Y.Z.) and the National Medical Research Council Open Fund – Young Individual Research Grant, Singapore (MOH-OFYIRG25jan-0011 to J.L.). The funders had no role in study design, data collection and analysis, decision to publish, or preparation of the manuscript.

## Author contributions

Y.Z. conceived and designed the project and supervised the work. J.L. developed the methods, trained the models, and analyzed the results. J.L and H.C. prepared the initial data and performed the benchmarks. J.L and Y.Z wrote the manuscript. All authors reviewed and approved the final version.

## Competing interests

The authors declare no competing interests.
