## [Transparent Peer Review file · Nature Communications]

A Paired Sequence Language Model for Protein-Protein Interaction Modeling

Corresponding Author: Professor Yang Zhang

Version 0:

Reviewer comments:

Reviewer #1

(Remarks to the Author)

The authors develop three specialized predictors—PPLM-PPI, PPLM-Affinity, and PPLM-Contact—targeting binary interaction, binding affinity, and interface contact predictions, respectively. The results demonstrate that the proposed method outperform existing methods.

1. Clarify Model Novelty: The manuscript claims novelty for PPLM's hybrid attention mechanism but the current claim of novelty appears overstated.
2. Address Dataset Redundancy and Bias: While MMseqs2 clustering (50% sequence identity) is mentioned, pair-level sequence redundancy (e.g., species-specific biases) is not analyzed. Conduct a bias audit, including sequence identity distributions and performance on underrepresented subsets, to ensure generalizability and mitigate redundancy effects.
3. Strengthen Ablation Studies: The ablation experiments (e.g., Figures S1, 4D) lack depth and statistical rigor. Supplement with additional experiments removing individual components (e.g. triangle multiplication) across multiple seeds, and report confidence intervals to validate the significance of each module's contribution.
4. Enhance Model Interpretability: The biological significance of PPLM's attention matrices is underexplored. Incorporate attention visualizations or SHAP analyses in the results section, with case studies on interface residues, to improve interpretability and relevance to PPI studies.
5. Justify Model Architecture and Computational Resources: The choice of a 33-layer architecture, inherited from ESM2, is not justified. Provide benchmarks comparing shallower models and detail computational resources (e.g., 6.4M training steps, 4x A100 GPUs) to clarify design rationale and efficiency.
6. Revise Figures for Clarity and Informativeness:
 - Enhance Figure 1D: Add detailed annotations (e.g., equations, arrows) for cross-attention, self-attention, and triangle multiplication modules, and illustrate the aggregation process from contact maps to residue-level predictions.
 - Refine Figure 1 and 2: Restructure Figure 1 to better reflect the PPLM framework and downstream tasks; include a schematic in Figure 2 to depict PPLM's masking strategy and workflow.
 - Correct and Improve Figure 3, 4, and 5: Fix the incorrect captions for Figures 3E/F (should read "PPLM-Affinity vs. ESM2-Affinity"). Replace head-to-head comparisons in Figures 4C and 5B with bar charts or heatmaps for clearer model performance visualization. Enhance Figure 4E with 3D structural overlays to improve interpretability.
7. Detail ESM2 Comparison for Perplexity: The comparison of perplexity with single-input ESM2 is insufficiently described. Include pseudocode or a flowchart in the methods section to clarify how pair-sequence inputs are processed in PPLM versus ESM2's single-chain approach.
8. Consolidate PPLM-Contact and PPLM-Contact2: As both models address the same task, discuss them in a unified results or methods subsection. Provide a table summarizing differences (e.g., PPLM-Contact2's use of predicted structure distance maps) and performance gains for clarity.
9. Improve Statistical Rigor: Excessive reliance on Wilcoxon tests without multiple-testing corrections is problematic. Incorporate effect sizes (e.g., Cohen's d) and false discovery rate adjustments to ensure robust statistical comparisons across all metrics.
10. Analyze Failure Cases and Model Limitations: The manuscript lacks analysis of failure modes (e.g., poor performance on specific heterodimers) and model limitations (e.g., underrepresentation of transient interactions). Include case studies with structural visualizations and discuss potential biases in the discussion section.
11. Update Benchmark Models: The chosen baseline models (e.g., D-SCRIPT, Topsy-Turvy) are not sufficiently representative of recent advances. Include comparisons with cutting-edge 2024–2025 methods (e.g., AlphaFold3 variants or GLINTER extensions) to strengthen the evaluation.

12. Analyze Impact of MSA Depth: The influence of multiple sequence alignment (MSA) depth on PPLM-Contact's performance is not evaluated. Assess how varying MSA depths (e.g., shallow vs. deep alignments from UniRef30) affect contact prediction accuracy, particularly for heterodimers, and report results in a supplementary table or figure.

13. From my experience, there exists a big issue, short-cut learning, for pair-input models. As shown in previous studies (PMID: 37031187, 38030641), the pair-input models do not work well for unseen proteins in the training set, where none of the proteins in the test set overlap with the training set. The authors should evaluate the inductive setting and compare with existing methods.

Reviewer #2

(Remarks to the Author)

The authors present a language model, PPLM, that embeds inter- and intraprotein interactions based on sequence information, and later incorporating structure information as well. They demonstrate that this model is supportive of applications for predicting binary protein interactions, predicting PPI affinity, and predicting amino acids that form intermolecular contacts. In their results, the authors demonstrate that their own techniques, which employ PPLM, can outperform single-purpose methods designed for one of these applications.

Overall, the presentation of the method is good and fairly clear. The experimental methodology, with regard to machine learning techniques is reasonably well designed, and the datasets are very comprehensive. Data and code have been shared publicly, though this reviewer is uncertain if everything needed for reproduction has been shared.

Major comments:

To this reviewer, the major contribution of the paper is that PPLM and its successor methods may offer a one-size-fits-all model that can support a class of techniques that better perform the comparison methods tested in the paper, and possibly others as well. It suggests, as the authors say, that something is being learned by PPLM in the embedding of inter- and intra-protein interactions that enables this outperformance and makes it accessible to downstream applications. However, the authors only offer speculation as to what that is. This is not an explainable AI method, so the authors cannot really introspect on the nature of the system, but a comparison of what is being represented by existing methods, and how it differs from PPLM, should at least be offered. It is not clear that inter-protein interactions are not represented at all in existing methods, but perhaps in a less explicit way than in PPLM. The ablation study was a good start, but a point by point comparison should be made in supplementary data.

For each of the three applications, the authors demonstrate that their method outperforms existing methods, but they offer no evidence as to how significant that outperformance is. The authors should provide the standard deviation of method performance across different folds of their cross validation experiments. Since they re-ran other methods in-house, the same should be done for other methods. Likewise, some statement about how other methods were trained, especially if they were trained at variance from the way their creators designed them.

To train PPLM, sequence pairs from PDB were clustered, and clusters redundant with those from the PDB were removed, yielding 629,045 clusters. But for the binding affinity prediction task, the method used five-fold cross-validation and grouped samples sharing the same PDB ID into the same fold to prevent data leakage (as mentioned in line 583). While this is a reasonable precaution to avoid exact duplicate entries between training and test sets, it does not adequately prevent homology-based leakage. Protein complexes with high sequence or structural similarity may be associated with different PDB IDs and end up in separate folds. This can lead to overestimated model performance, as the model may still encounter similar complexes in both training and testing. This issue must be examined, and the PDB IDs in each fold should be provided in supplementary materials.

Minor points:

- several misspellings in the text (line 58) and figure 1d.

Reviewer #3

(Remarks to the Author)

Overview

This paper introduces a new protein language model trained specifically on protein-protein interaction data. The authors cleverly modify existing architectures and show impressive performance gains in a variety of PPI-related tasks, from binary PPI prediction, to protein-ligand binding affinity prediction, to contact map prediction.

My main concern is that I don't exactly understand how data leakage is guarded against in a variety of tasks. In the PPLM-Affinity task, the authors say they avoid data leakage by grouping samples that share the same PDB ID into the same fold of cross-validation. But this is insufficient, since partial crystal structures of the same protein will be given different PDB IDs.

The authors need to repeat their experiments instead with the requirement of some structure similarity cutoff to avoid including the same structure across different cross-validation folds. As another example, in the PPI-prediction task, they follow D-SCRIPT's paradigm of training on human PPI, and then testing on PPI in five other species. However, it seems their MSA is species agnostic and constructed from UniRef30 using HHblits. It is hard to tell if this results in a serious data leakage problem or not. The performance they are getting is similar to deep learning methods for this problem that are trained and then tested on the same species, which makes me worried. In order to investigate further, the authors should also test a simple non-deep predictor on this dataset that relies only on the MSA: looking at the closest human homolog to any protein included in the MSA, they should predict "interact" if the human proteins participating in the same MSA interact, and don't interact otherwise. How does this simple predictor relying on the MSA do on the PPI prediction task compared to their method? The authors could also benchmark their PPI-predictor on a dataset carefully designed to avoid this problem, namely the dataset from Bernett, Judith, David B. Blumenthal, and Markus List. "Cracking the black box of deep sequence-based protein-protein interaction prediction." Briefings in Bioinformatics 25, no. 2 (2024), but they should still test the same suggested non-deep predictor suggested above on this dataset.

Major comments

- Very impressive performance gains in every task, and clever modifications to existing network architectures, PPI vs Affinity model variants
- The PPI task model and the Affinity task model are quite different, but when restricted to the Affinity dataset, how well do PPI model predicted probabilities correlate with binding affinity?
- Initial Contact model should be compared to AlphaFold contact predictions. AlphaFold results are reported later when comparing to Contact2, but are relevant in assessing the quality of the original Contact model as well. By eye comparing across figures, AlphaFold outperforms the original Contact model - this should be discussed
- Ablation study shows that the PPLM embeddings give the smallest boost in performance out of all the input features, might be worth noting/discussing that
- Not clear how the pairwise nature of the data was handled when clustering the dataset with MMseqs2. Is the 80% coverage minimum applied to each member of the pair separately? Are two pairs clustered together only if both members of one pair are highly homologous to a member of the other pair or just one? Which clustering method (greedy set cover or connected component) was used in MMseqs clustering?

Minor Comments

- Line 95 talks about independent benchmark datasets in such a way that it sounds like all these datasets were created by the authors: in fact, based on the more detailed description further on in the paper, they choose several benchmarks constructed in previous work for their tests: it only strengthens the paper to use an existing benchmark. But they should make clear here that the PPI dataset they train and test on is nearly the same as the D-SCRIPT dataset (with some small housekeeping adjustments), and the binding affinity dataset they use is the PPB-Affinity dataset.
- Some scatterplots have too many points (figures 3E,F S2B-E)- should convert to 2D histograms or color each point based on the density of other points around it

Software installation and testing notes

- References to config.py should always use pplm_contact/config.py to avoid confusion
- Should specify that Uniclust30 needs to be unzipped
- Parameter downloads from their lab server seem to be throttled to ~250KB/s - this is impractically slow
- Would be nice to have a script that downloads all model parameters instead of having to do each separately
- Had to uninstall and re-install MKL (version 2023.1.0-h213fc3f_46344) to get the conda environment to work
- Image at top of github page is cropped
- Issue with libperl.so file not found despite it's presence in the conda environment folder - prevented PPLM-Contact and PPLM-Contact2 from running properly
- PPLM-PPI and PPLM-Affinity run properly on test cases

Reviewer #4

(Remarks to the Author)

Version 1:

Reviewer comments:

Reviewer #1

(Remarks to the Author)

The authors improved the manuscript and addressed all my comments

Reviewer #2

(Remarks to the Author)

The authors present a language model, PPLM, that embeds inter- and intraprotein interactions based on sequence information, and later incorporating structure information as well. They demonstrate that this model is supportive of applications for predicting binary protein interactions, predicting PPI affinity, and predicting amino acids that form intermolecular contacts. In their results, the authors demonstrate that their own techniques, which employ PPLM, can outperform single-purpose methods designed for one of these applications.

The revisions developed by the authors effectively address my earlier concerns, and have not advanced new concerns. This reviewer supports the acceptance of this paper with its complete supplemental materials as described by the authors.

Reviewer #3

(Remarks to the Author)

The authors have substantially strengthened the manuscript by carefully responding, often with additional experiments, to reviewer comments. The benchmarking is much better. I now think the manuscript is ready to be published.

Reviewer #4

(Remarks to the Author)

Response to Reviewer #1's comments

We sincerely thank the Reviewer for the insightful comments, which have substantially improved the quality of our manuscript. The primary concerns involved dataset redundancy, species-specific biases, and potential shortcut learning in paired-input models. In response, we implemented stricter pair-level redundancy filtering and added similarity-stratified evaluations across five non-human species, demonstrating strong inductive generalization and consistent improvements over all baselines. The Reviewer also highlighted the need for enhanced interpretability, deeper ablation analyses, more rigorous statistical evaluations, and inclusion of newer baselines. Accordingly, we incorporated attention-based case studies, multi-seed ablations with confidence intervals, effect-size analyses with FDR correction, and updated comparisons including TUnA and PLMGraph-Inter. We also revised key figures, added workflow schematics, examined the effects of MSA depth, and included representative failure cases to address the remaining concerns. Point-by-point responses are detailed below, and all revisions in the manuscript are highlighted in yellow.

1. The Reviewer commented:

The authors develop three specialized predictors—PPLM-PPI, PPLM-Affinity, and PPLM-Contact—targeting binary interaction, binding affinity, and interface contact predictions, respectively. The results demonstrate that the proposed method outperform existing methods.

Here are the comments:

1. Clarify Model Novelty: The manuscript claims novelty for PPLM's hybrid attention mechanism but the current claim of novelty appears overstated.

Response: We thank the Reviewer for this helpful comment. In the revised manuscript, we have updated the novelty description to clarify that our contribution does not lie in proposing a fundamentally new attention primitive. Instead, the novelty comes from adapting transformer attention specifically for paired protein sequences through a hybrid intra-/inter-protein attention design.

More specifically, PPLM integrates (i) rotary-embedding-based intra-protein positional encoding, (ii) non-positional inter-protein embeddings to avoid spurious distance priors, (iii) learnable inter-protein attention weights, and (iv) an explicit inter-protein attention mask. Although each component builds upon standard transformer attention, their coordinated integration for co-represented protein pairs enables the model to separate and balance intra-chain and inter-chain contextual signals—capabilities that single-chain language models inherently lack.

To avoid overstating novelty and improve clarity, we have added the following paragraph in the Introduction (Page 2):

To address these limitations, we present PPLM, a protein-protein language model specifically designed to learn contextual and relational features from paired protein sequences. To better capture inter-protein dependencies, we introduce a hybrid intra-/inter-protein attention mechanism that adapts transformer attention for co-represented protein pairs. Specifically, this mechanism integrates rotary embeddings-based positional encoding³⁷ for intra-protein attention with non-positional embedding for inter-protein residue pairs to avoid spurious spatial priors. In addition, learnable inter-protein attention weights and an explicit inter-protein attention mask are incorporated to enhance the modeling of cross-protein interactions while preserving a clear separation between intra- and inter-chain information flows. To support large-scale training, we further construct a comprehensive sequence-pair dataset comprising over 3.3 million high-quality protein pairs, enabling PPLM to learn robust interaction-aware representations.

2. The Reviewer commented:

2. Address Dataset Redundancy and Bias: While MMseqs2 clustering (50% sequence identity) is mentioned, pair-level sequence redundancy (e.g., species-specific biases) is not analyzed. Conduct a bias audit, including sequence identity distributions and performance on underrepresented subsets, to ensure generalizability and mitigate redundancy effects.

Response: We thank the Reviewer for this insightful comment. In the revised manuscript, we have performed a stricter, species-agnostic redundancy audit by removing all validation pairs that shared $\geq 50\%$ identity and $\geq 80\%$ coverage with any training pair, regardless of pair type, and added the detailed dataset construction procedures in **Supplementary Text S7**. We also updated the random masking procedure to ensure comprehensive coverage of both chains when computing perplexity, with synchronized masking for homomers to avoid trivial inference.

We also conducted a detailed bias analysis by computing the sequence-identity distribution between training and validation pairs and evaluating performance across identity intervals under a coverage cutoff of 0.8 (**Figure 2H**), as well as cutoffs of 0.7, 0.6, 0.5, 0.4, 0.3, and 0.2 (**Figure S3**). Across all coverage thresholds, most validation pairs had no match in the training set (“non-match”), and PPLM consistently maintained stable performance and a clear advantage over ESM2 across all identity bins. These results demonstrate that PPLM performs strongly on underrepresented subsets and retains generalizability even under stringent redundancy controls, thereby mitigating potential bias from residual pair-level similarity.

We have added the following paragraphs and figures to summarize the results (Pages 3 and 9):

Page 3:

PPLM improves language modeling of protein pairs

Perplexity provides an objective, task-independent measure of a language model’s confidence in predicting masked amino acids. Lower perplexity indicates higher predictive accuracy, with a value of 1 representing perfect prediction and ~20 reflecting near-random performance. We used perplexity to benchmark PPLM against ESM2⁹, a state-of-the-art single-sequence language model that does not explicitly model inter-protein context. To more comprehensively evaluate model behavior on protein pairs, we implemented three complementary masking strategies (**Supplementary Fig. S1**).

Under a random masking strategy, perplexity was computed by iteratively masking non-overlapping 15% of residues in both sequences until all positions were covered. Identical positions in homomers were masked jointly to avoid trivial inference.

To assess potential pair-level redundancy and species-specific biases, we conducted a comprehensive bias audit by comparing all validation pairs to the training set. Most validation pairs had no detectable match, and PPLM’s performance remained stable across identity intervals (**Figure 2H**). PPLM consistently outperformed ESM2 across the entire identity spectrum, and this trend remained robust across alternative coverage thresholds (**Supplementary Figure S3**).

Page 9:

Datasets

To train PPLM, we constructed a composite protein dataset of protein interaction sequence pair by integrating protein complexes released before January 1, 2024 from the Protein Data Bank (PDB)⁴⁷ and interaction sequences from the STRING⁴⁸ database. After filtering for physical interfaces and removing redundancy through pair-level clustering using MMseqs2⁴⁹ together with a custom clustering procedure, the dataset comprised 25,245 heteromeric clusters and 23,082 homomeric clusters from PDB, as well as 629,045 clusters from STRING, covering over 3.3 million protein sequence pairs. For model validation, we randomly selected 5,000 singleton clusters and used the remaining clusters for training. After removing validation pairs that were redundant to any pairs in the training clusters, the final validation set consisted of 1,204 heteromeric, 1,003

homomeric, and 2,471 STRING-derived pairs (**Supplementary Text S7**). During the training of PPLM, sequence pairs were sampled from the PDB and STRING clusters at a 1:2 ratio.

Figure 2H:

Figure 2. Perplexity of PPLM and ESM2 on protein sequence pairs. (H) Perplexities of PPLM and ESM2 under varying pairwise sequence similarity to the training set. The number of samples within each similarity bin is indicated.

Figure S3:

Figure S3. Sequence pair identity distribution and model performance across coverage cutoffs. Distribution of pairwise sequence identities between validation and training pairs, and corresponding model performance (perplexity) under alignment coverage cutoffs of 0.7 (A), 0.6 (B), 0.5 (C), 0.4 (D), 0.3 (E), and 0.2 (F).

3. The Reviewer commented:

3. Strengthen Ablation Studies: The ablation experiments (e.g., Figures S1, 4D) lack depth and statistical rigor. Supplement with additional experiments removing individual components (e.g. triangle multiplication) across multiple seeds, and report confidence intervals to validate the significance of each module's contribution.

Response: We thank the Reviewer for this constructive suggestion. In the revised manuscript, we have extended the ablation analyses of both PPLM-PPI and PPLM-Contact to include multi-seed evaluations and statistical reporting. For each variant, we report the mean performance \pm 95 % confidence intervals (CI) across three random seeds. The detailed analysis has been added to **Supplementary Texts S2 and S5**.

We have added the following paragraphs to report the results (Pages 4 and 6):

Page 4:

To elucidate the contributions of different components, we conducted ablations by modifying pooling modules and selectively removing PPLM-derived features (**Supplementary Text S2**). Multi-seed results (mean \pm 95% CI) are presented in **Supplementary Table S4-5 and Figure S4**. Among the individual pooling strategies, mean pooling achieved the highest accuracy, while the combined mean-max pooling strategy produced the most robust and consistent performance across species. In the feature-level analysis, incorporating all three PPLM-derived feature types, including inter-protein attention, intra-protein attention, and embedding, yielded the best results, with embedding features contributing the strongest predictive signal. These findings highlight the importance of both effective pooling and the integration of complementary attention-based and embedding-based representations.

Page 6:

Figure 4D–F summarize the mean and standard deviation of top L precision across three random seeds for PPLM-Contact and all ablation variants. These results show that removing any major network component or feature of the model reduces precision, with the largest performance drops observed when excluding the triangle-multiplication module, PPLM inter-protein attention, MSA features, or monomer distance maps. Detailed ablation analyses and statistical comparisons are provided in **Supplementary Text S5**. The effect of MSA depth is further examined in **Supplementary Text S6**.

Supplementary Text S2. Ablation analysis of PPLM-PPI.

To assess the influence of network-level pooling operations and input feature compositions on cross-species PPI prediction, we performed two groups of ablation experiments for PPLM-PPI. Each variant was evaluated on the *M. musculus*, *D. melanogaster*, *C. elegans*, *S. cerevisiae*, and *E. coli* test sets, and we report the mean accuracy \pm 95 % confidence intervals (CI) across three random seeds.

Seven pooling configurations were compared, combining mean, max, and min operations in different ways. As shown in **Figure S4A**, the mean-max pooling consistently achieved the best or near-best AUPRC across all species, reaching 0.921 ± 0.002 in *M. musculus*, 0.906 ± 0.002 in *D. melanogaster*, and 0.884 ± 0.002 in *C. elegans*. Removing either the mean or max component led to noticeable declines (e.g., mean only: 0.914 ± 0.002 ; max only: 0.903 ± 0.002 in *M. musculus*). Among single pooling operations, mean pooling achieved the highest accuracy, while min pooling performed the worst across species, likely due to its sensitivity to weak or noisy signals. These trends are consistent across all five species, confirming that combining mean and max pooling provides a balanced representation of global and salient pairwise features. The narrow confidence intervals (< 0.01) demonstrate stable performance across random seeds.

Under the optimal single mean-pooling configuration, we further examined the contributions of different feature types, including PPLM-derived inter-attention, intra-attention, and embedding features. As shown in **Figure S4B**, the full model integrating all three components achieved the highest AUPRC across all species (e.g., 0.914 ± 0.002 in *M. musculus*, 0.886 ± 0.005 in *D. melanogaster*, and 0.853 ± 0.003 in *C. elegans*). Removing any attention stream markedly reduced

performance—for instance, the inter-attention + intra-attention variant dropped to 0.560 ± 0.001 in *M. musculus* and 0.586 ± 0.007 in *D. melanogaster*—indicating that attention signals alone without embeddings are insufficient. Models retaining embeddings (e.g., *intra-attention + embedding* or *embedding only*) maintained substantially higher accuracy than those without, demonstrating that embeddings contribute most strongly to predictive performance. Models using a single attention type (either intra- or inter-attention alone) produced the lowest accuracies (~ 0.52 – 0.69).

Across both analyses, all confidence intervals remained below 1%, supporting the statistical robustness of the observed trends. Collectively, these experiments demonstrate that the mean–max pooling strategy and the combined inter-attention + intra-attention + embedding feature set are critical for achieving robust and generalizable cross-species PPI prediction. Detailed numerical results for all metrics are listed in **Table S2-3**.

Supplementary Text S5. Ablation analysis of PPLM-Contact.

To evaluate the contribution of individual architectural modules and feature types, we trained six ablated variants of PPLM-Contact, each excluding one key component while keeping all other settings unchanged: (i) w/o cross-attn, which removes the cross-attention module from the architecture; (ii) w/o self-attn, which removes the self-attention module; (iii) w/o tri-multi, which excludes the triangle-multiplication module; (iv) w/o MSA, which excludes MSA-derived features including PSSM, DCA, and ESM-MSA-1b outputs; (v) w/o PPLM, which removes inter-protein attention features generated by PPLM; and (vi) w/o Mdist, which omits monomer distance maps extracted from either experimental or AlphaFold-predicted monomer structures.

Figure 4D-F illustrates the overall performance changes of these variants on the Homodimer300 and Heterodimer99 test sets. For each variant, we report the mean top L inter-protein contact precision and 95% confidence intervals (CI) across three random seeds. Detailed top 1 to top L results are provided in **Supplementary Table S13**. Compared with the full model ($77.2 \pm 0.58\%$ for homodimers and $49.7 \pm 1.02\%$ for heterodimers), all ablations resulted in consistent and statistically meaningful declines, confirming the robustness of each component's contribution.

At the network-architecture level, removing either cross-attention or self-attention modules led to modest yet reproducible reductions—by approximately 0.8–1.0 percentage points (abbreviated as pp) for homodimers and 1.7–2.0 pp for heterodimers—indicating their supportive roles in enhancing intra- and inter-chain contextualization. In contrast, excluding the triangle-multiplication module caused a much greater decline, reducing precision by 6.7 pp and 9.4 pp on homodimers and heterodimers, respectively, highlighting its central role in capturing residue-pair geometric relationships.

At the feature level, removing the PPLM inter-protein attention features—the paired-sequence embeddings generated by PPLM—lowered precision from 77.2% to 73.3% for homodimers and from 49.7% to 45.3% for heterodimers, underscoring PPLM's key role in modeling cross-chain interactions. Removing MSA-derived features reduced precision by 7.0 pp and 9.3 pp, reflecting the contribution of co-evolutionary information, while omitting M-distance features yielded the largest degradation (23.3 pp and 16.8 pp for homodimers and heterodimers, respectively), emphasizing the indispensable role of intra-chain geometric priors.

Together, these analyses confirm that every component of PPLM-Contact contributes meaningfully to predictive accuracy, with particularly strong and statistically supported effects from the triangle-multiplication, PPLM inter-protein attention, MSA, and monomer distance modules.

Figure 4D-E:

Figure 4. Performance of PPLM-Contact and the comparison methods on the inter-protein contact prediction. (D–E) Mean top L contact precision and 95% confidence intervals (CI) of

PPLM-Contact and its ablation variants across three seeds on the Homodimer300 and Heterodimer99 test sets, respectively.

Figure S4:

Figure S4. Performance of PPLM-PPI and its Ablated variants. (A) AUPRC of different pooling strategies **(B)** AUPRC of different feature combinations. Values represent the mean \pm 95 % confidence intervals (CI) computed across three random seeds.

4. The Reviewer commented:

4. Enhance Model Interpretability: The biological significance of PPLM's attention matrices is underexplored. Incorporate attention visualizations or SHAP analyses in the results section, with case studies on interface residues, to improve interpretability and relevance to PPI studies.

Response: We thank the Reviewer for this valuable suggestion. To strengthen model interpretability and explore the biological relevance of PPLM's attention matrices, we have added a case study for a homodimer complex 1Y9B, a conserved transcription factor from *Vibrio cholerae* O1 biovar El Tor, in the revised manuscript (**Figure 2D–F**). These analyses demonstrate that PPLM's inter-protein attention captures biologically meaningful residue–residue interaction patterns despite the absence of explicit structural supervision.

We have added the following paragraph to clarify the insights (Page 3):

To examine whether PPLM captures biologically meaningful dependencies, we analyzed the homodimer complex 1Y9B, a conserved transcription factor from *Vibrio cholerae* O1 biovar El Tor

(Figure 2D–F). The inter-protein attention matrix from the final PPLM layer showed high-scoring regions that closely matched the experimental contact map. Among the top 20 residue pairs ranked by attention, 90% corresponded to true heavy-atom contacts (80% to C_β–C_β contacts). Of the 62 experimentally determined interface residues, 52 (83.9%) were recoverable from attention-derived residue rankings. The corresponding 3D visualization demonstrated that these predicted residues cluster around the physical interface, indicating that PPLM’s attention mechanism naturally focuses on interaction-relevant regions during unsupervised learning. A more detailed representation-level comparison between PPLM and ESM2, including unsupervised and linear-probe analyses of inter-protein attention matrices, is provided in **Supplementary Text S1**.

Figure 2D-F:

Figure 2. Perplexity of PPLM and ESM2 on protein sequence pairs. (D-F) Example of a conserved putative transcription factor from *Vibrio cholerae* O1 biovar El Tor (PDB ID: 1Y9B). (D) Heatmap of the inter-protein attention matrix generated by PPLM. (E) Ground-truth contact map extracted from the experimental structure. (F) Three-dimensional virtualization of the complex, with interface residues identified from the inter-protein attention matrix highlighted.

5. The Reviewer commented:

5. Justify Model Architecture and Computational Resources: The choice of a 33-layer architecture, inherited from ESM2, is not justified. Provide benchmarks comparing shallower models and detail computational resources (e.g., 6.4M training steps, 4x A100 GPUs) to clarify design rationale and efficiency.

Response: We thank the Reviewer for this helpful suggestion. In the revised manuscript, we have trained PPLM variants inherited from ESM2 models with different depths (t6_8M, t12_35M, t30_150M, and t33_650M) using the same training protocol. As shown in **Figure 2G**, perplexity decreases monotonically with increasing model size under all masking strategies, and the 33-layer model achieves the best overall performance. Relative to t30_150M, the t33_650M model reduces perplexity by 44.8% under random masking, 45.3% under “Dual” interface masking, and 38.8% under “Single” interface masking, supporting our choice of the larger architecture.

In addition, we corrected an earlier reporting error in training steps. The correct training schedule involved 50,000 steps with 128 sequence pairs per step, totaling 6.4 million pairs involved.

Regarding computational resources, our cluster nodes are equipped with four NVIDIA A100 GPUs each. For balance between efficiency and resource availability, we trained PPLM on 4 GPUs within a single node. Training PPLM with t33_650M parameters required approximately 93 hours. We have updated these details in the revised manuscript.

We have added the following paragraphs to clarify the insights (Pages 3 and 11):

Page 3:

As expected for paired-sequence modeling, smaller PPLM variants exhibited higher perplexity, whereas deeper models showed substantially improved performance (Figure 2G). To assess potential pair-level redundancy and species-specific biases, we conducted a comprehensive bias audit by comparing all validation pairs to the training set. Most validation pairs had no detectable match, and PPLM’s performance remained stable across identity intervals (Figure 2H).

Page 11:

PPLM was initialized from the 650M-parameter ESM2 model⁶ and trained using four NVIDIA A100 GPUs with a gradient accumulation step of 32, for a total of 50,000 steps. Optimization was performed using the AdamW optimizer with exponential decay rates $\beta_1=0.9$ and $\beta_2=0.98$. The learning rate was linearly warmed up to 1×10^{-6} over the first 2,000 steps, followed by a linear decay to 5×10^{-7} over the remainder of the training.

Figure 2G:

Figure 2. Perplexity of PPLM and ESM2 on protein sequence pairs. (G) Perplexities of PPLM across different parameter scales. (H) Perplexities of PPLM and ESM2 under varying pairwise sequence similarity to the training set. The number of samples within each similarity bin is indicated.

6. The Reviewer commented:

6. Revise Figures for Clarity and Informativeness:

- Enhance Figure 1D: Add detailed annotations (e.g., equations, arrows) for cross-attention, self-attention, and triangle multiplication modules, and illustrate the aggregation process from contact maps to residue-level predictions.
- Refine Figure 1 and 2: Restructure Figure 1 to better reflect the PPLM framework and downstream tasks; include a schematic in Figure 2 to depict PPLM’s masking strategy and workflow.
- Correct and Improve Figure 3, 4, and 5: Fix the incorrect captions for Figures 3E/F (should read “PPLM-Affinity vs. ESM2-Affinity”). Replace head-to-head comparisons in Figures 4C and 5B with bar charts or heatmaps for clearer model performance visualization. Enhance Figure 4E with 3D structural overlays to improve interpretability.

Response: We thank the Reviewer for these valuable suggestions regarding figure clarity and informativeness. In the revised manuscript, we have refined Figures 1, 3E–F, 4 and 5, and added a schematic (Supplementary Figure S1) illustrating PPLM’s masking strategy and end-to-end processing pipeline. For Figures 4C and the original Figure 5B (now Supplementary Figures S10–S11), we now present head-to-head comparisons where each panel shows PPLM against a single baseline, improving visual separation and interpretability. For Figure 4E, overlaying predicted contacts on this small homodimer produced heavily overlapping and cluttered views, so we retain it as a simple structural reference and instead present the detailed interpretability analysis in Figure 4F, where native contacts, true positives, and false positives from multiple variants are visualized on the contact maps.

We have updated these figures in the manuscript:

Figure 2G:

Figure 1. Overview of the PPLM framework and its downstream applications. (A) PPLM architecture. The input protein sequence pair are independently tokenized and then processed through transformer blocks equipped with a cross-protein attention mechanism to capture intra- and inter-protein interactions. The resulting embeddings and attention matrices can be leveraged for diverse PPI tasks. **(B)** PPLM-PPI pipeline. For a given sequence pair, PPLM is used to generate the embeddings, intra- and inter-protein attention matrix. These representations are aggregated using max and mean pooling, each followed by a multilayer perceptron to estimate the interaction probability. The final prediction is obtained by averaging the probabilities from the two pooling branches. **(C)** PPLM-Affinity pipeline. The final transformer block of PPLM is finetuned on binding affinity data, where the resulting embeddings for paired sequences are aggregated via max pooling and passed through two fully connected layers to predict binding affinity value. **(D)** PPLM-Contact pipeline. For a given sequence pair, inter-protein attention matrix from PPLM is integrated with MSA-derived features and monomer distance maps to capture both evolutionary and structural information. These combined features are processed by a series of inter-protein transformer blocks, with each composed of three core modules to predict the inter-protein contacts. Each module adopts a parallel architecture to update inter-protein representations. The top interface residues (top 50 displayed) were extracted from the top-ranked predicted contact pairs.

Figure S1:

Random masking (masking 15% of residues in each sequence without replacement; 10% for the final round)

```

G P L G S M Q R I N N A I D S L I G H L V P A A A G D D D L A R T R A E K Q A A A Q Q A V D I L H E I A T S
G P L G S M Q R I N N A I D S L I G H L V P A A A G D D D L A R T R A E K Q A A A Q Q A V D I L H E I A T S
G P L G S M Q R I N N A I D S L I G H L V P A A A G D D D L A R T R A E K Q A A A Q Q A V D I L H E I A T S
G P L G S M Q R I N N A I D S L I G H L V P A A A G D D D L A R T R A E K Q A A A Q Q A V D I L H E I A T S
G P L G S M Q R I N N A I D S L I G H L V P A A A G D D D L A R T R A E K Q A A A Q Q A V D I L H E I A T S
G P L G S M Q R I N N A I D S L I G H L V P A A A G D D D L A R T R A E K Q A A A Q Q A V D I L H E I A T S
G P L G S M Q R I N N A I D S L I G H L V P A A A G D D D L A R T R A E K Q A A A Q Q A V D I L H E I A T S
G P L G S M Q R I N N A I D S L I G H L V P A A A G D D D L A R T R A E K Q A A A Q Q A V D I L H E I A T S

```

Single-chain interface masking (masking all interface residues from one sequence)

```

G P L G S M Q R I N N A I D S L I G H L V P A A A G D D D L A R T R A E K Q A A A Q Q A V D I L H E I A T S
G P L G S M Q R I N N A I D S L I G H L V P A A A G D D D L A R T R A E K Q A A A Q Q A V D I L H E I A T S

```

Dual-chain interface masking (masking all interface residues from both sequences)

```

G P L G S M Q R I N N A I D S L I G H L V P A A A G D D D L A R T R A E K Q A A A Q Q A V D I L H E I A T S
G P L G S M Q R I N N A I D S L I G H L V P A A A G D D D L A R T R A E K Q A A A Q Q A V D I L H E I A T S

```

Figure S1. Schematic of masking strategies for perplexity calculation. We designed three masking strategies. (i) Random masking: randomly masking 15% of residues in each sequence without replacement (10% for the final round), with the average perplexity across rounds used as the perplexity for the sequence pair. (ii) Single-chain interface masking: masking all interface residues from one sequence at a time. (iii) Dual-chain interface masking: masking all interface residues from both sequences simultaneously. Interface residues are highlighted using red text and colored backgrounds.

Figure 3D-F:

Figure 3. Performance of PPLM-PPI and PPLM-Affinity on protein-protein interaction and binding affinity prediction. (D) Pearson (PCC) and Spearman (SRCC) correlation coefficients of PPLM-Affinity, ESM2-Affinity, and PPB-Affinity evaluated on the entire dataset, as well as the antibody-antigen and TCR-pMHC subgroups. (E-F) Head-to-head comparison between experimental and predicted binding affinity for PPLM-Affinity and ESM2-Affinity, respectively. Plots show hexagonal density maps to improve visibility at high point density. Color intensity reflects the local log-transformed point density.

Figure 4C:

Figure 4. Performance of PPLM-Contact and the comparison methods on the inter-protein contact prediction. (C) Head-to-head comparison of top L contact precision between PPLM-Contact and other methods on the 343 homodimer and 119 heterodimer proteins.

Figure S10:

Figure S10. Comparative analysis of protein–protein interface residue identification using experimental monomer structures. (A) Precision of interface-residue identification by general contact-prediction methods (GLINTER, DeepHomo2.0, PLMGraph-Inter, CDPred, DeepInter, and PPLM-Contact). **(B–F)** Head-to-head comparison of identification precision between PPLM-Contact and each control method on the homodimer and heterodimer proteins, respectively.

7. The Reviewer commented:

7. Detail ESM2 Comparison for Perplexity: The comparison of perplexity with single-input ESM2 is insufficiently described. Include pseudocode or a flowchart in the methods section to clarify how pair-sequence inputs are processed in PPLM versus ESM2’s single-chain approach.

Response: We thank the Reviewer for the insightful suggestion. In the revised manuscript, we have added a schematic to clarify how paired sequences are processed in PPLM versus ESM2’s single-chain architecture (**Supplementary Figure S15**).

We have added the following paragraph and figure to report the results (Page 11):

Page 11:

Protein–protein language model

PPLM is a pretrained protein–protein language model that can capture potential protein interaction information by providing informative representations of protein pairs (**Figure 1A**). Given a protein sequence pair as input, the sequences are first tokenized independently and then concatenated with special beginning-of-sequence (BOS) and end-of-sequence (EOS) tokens to explicitly mark the boundaries of each chain. The concatenated initial representations will be used as the input of a stack of 33 serially connected transformer blocks, with each block consisting of a tailored multi-head attention module and a feed-forward network. **The preprocessing workflows for paired sequences in PPLM and in ESM2’s single-chain approach are illustrated in Supplementary Figure S15.**

Figure S15:

Figure S15. Overview of PPLM and ESM2 processing for protein sequence pair. (A) In PPLM, the two input sequences are first encoded separately and then concatenated before being passed through the model’s pairwise transformer blocks. (B) In ESM2, which is designed for single-chain modeling, the two sequences are directly concatenated into a single input and processed by its single-chain transformer blocks. For both PPLM and ESM2, the outputs include logits, embedding and attention matrix corresponding to the input sequence pair.

8. The Reviewer commented:

8. Consolidate PPLM-Contact and PPLM-Contact2: As both models address the same task, discuss them in a unified results or methods subsection. Provide a table summarizing differences (e.g., PPLM-Contact2’s use of predicted structure distance maps) and performance gains for clarity.

Response: We thank the Reviewer for this helpful suggestion. In the revised manuscript, we have added a unified performance overview (**Figure 5C–D**) to enable direct comparison across all methods. To maintain a clear separation of scope between general inter-protein contact prediction and methods that use predicted complex structures, and to avoid an overly long combined subsection, we have retained the two subsections but now present them consecutively and explicitly emphasize their progressive relationship.

We have revised the following paragraphs to clarify the overall narrative and delineate the methodological boundaries between the two versions (Page 7):

Enhanced inter-protein contact prediction using predicted complex structures

Building on the performance of PPLM-Contact, we next examined whether incorporating structural information from modern complex-prediction models could further enhance inter-protein contact prediction. Recent advances in multimer structure prediction now provide highly accurate models of protein complexes, capturing detailed geometric organization at interfaces that is difficult to infer from sequence alone. These predicted structures therefore offer an opportunity to supply language-model pipelines with precise spatial context. To leverage this, we developed PPLM-Contact2, an enhanced variant of PPLM-Contact that integrates inter-protein distance maps extracted from predicted complex structures. We benchmarked PPLM-Contact2 against AlphaFold2.3³⁸, AlphaFold3³⁹, and DMFold⁴⁰ across all 434 homodimer and 119 heterodimer proteins (**Supplementary Tables S14–S15; Supplementary Data 3–4**).

Figure 5C–D summarize the top L precision of all general contact-prediction methods (using AlphaFold2-predicted monomer structures) and the complex structure-based methods (AlphaFold2.3, AlphaFold3, DMFold, and PPLM-Contact2) on the 434 homodimer and 119 heterodimer proteins. As expected, the integration of complex-structure features allows PPLM-Contact2 to markedly improve upon PPLM-Contact, increasing top L precision from 65.0% to 85.1% on homodimers and from 44.3% to 88.0% on heterodimers. Compared with structure-modeling approaches, PPLM-Contact2 achieves improvements of 4.8% and 7.6% over AlphaFold2.3, 8.7% and 5.6% over AlphaFold3, and 5.6% and 8.1% over DMFold on homodimers and heterodimers. All improvements are statistically significant (**Supplementary Table S15**).

Figure 5C-D:

Figure 2. Perplexity of PPLM and ESM2 on protein sequence pairs. (C–D) top L contact precision of general contact-prediction methods (GLINTER, DeepHomo2.0, CDPred, PLMGraph-Inter, DeepInter, and PPLM-Contact) and complex structure-based approaches (AlphaFold2.3, AlphaFold3, DMFold, and PPLM-Contact2) on 343 homodimers and 119 heterodimers, respectively.

9. The Reviewer commented:

9. Improve Statistical Rigor: Excessive reliance on Wilcoxon tests without multiple-testing corrections is problematic. Incorporate effect sizes (e.g., Cohen’s d) and false discovery rate adjustments to ensure robust statistical comparisons across all metrics.

Response: We thank the Reviewer for this valuable suggestion. In the revised manuscript, we have incorporated three metrics for strict statistical analysis: (i) paired Wilcoxon signed-rank tests, (ii) paired Cohen’s d effect sizes, and (iii) Benjamini–Hochberg false-discovery-rate (FDR) correction across all comparisons. The detailed results are listed in **Supplementary Tables S3, S9, S11, S12, S15, S16, and S17**. Among the three downstream tasks, only PPLM-Contact involves per-target paired comparisons, where Wilcoxon signed-rank and FDR correction are applicable. For PPLM-PPI and PPLM-Affinity, we have reported the relative percentage improvement and paired Cohen’s d effect sizes.

We have added the following paragraph and tables to report the analysis results (Pages 4–8):

Page 4:

Supplementary Table S3 details the statistical comparisons. Relative to TUNA and ESMDNN-PPI, PPLM-PPI improves F1-score by 10.1–10.5% and AUPRC by 9.6–11.9%, with large effect sizes (Cohen’s $d \approx 1.742$ – 6.586). Against D-SCRIPT and Topsy-Turvy, the improvements are even larger: 32.5–72.3% (F1) and 58.7–60.9% (AUPRC), also with consistently large effect sizes (Cohen’s $d \approx 2.062$ – 6.183).

Table S3. Statistical comparison between PPLM-PPI and baseline models (TUnA, ESM2-DNN-PPI, D-SCRIPT, and Topsy-Turvy) across the five species test sets. For each evaluation metric (F1-score, AUROC, and AUPRC), the table reports the fold-wise mean difference (Δ), relative percentage improvement, and paired effect size (Cohen’s d).

	Metric	Mean Δ	Percent gain (%)	Cohen’s d
vs TUnA	F1-score	0.068	10.1	2.252
	AUROC	0.021	2.3	1.742
	AUPRC	0.070	9.6	2.268
vs ESM2-DNN-PPI	F1-score	0.072	10.5	6.586
	AUROC	0.014	1.5	3.084
	AUPRC	0.086	11.9	3.026
vs D-SCRIPT	F1-score	0.319	72.3	4.765
	AUROC	0.134	16.4	2.824
	AUPRC	0.316	60.9	5.306
vs Topsy-Turvy	F1-score	0.183	32.5	3.728
	AUROC	0.15	20.0	2.062
	AUPRC	0.3	58.7	6.183

Page 5:

The left panel of **Figure 3D** reports the Pearson (PCC) and Spearman (SRCC) correlations between predicted and experimental ΔG values. These two metrics capture complementary aspects of predictive performance: PCC quantifies linear association, whereas SRCC measures rank consistency. PPLM-Affinity achieves mean \pm standard deviation PCC and SRCC values of 0.643 ± 0.058 and 0.636 ± 0.082 , representing improvements of 17.3% and 16.4% over ESM2-Affinity (0.548 ± 0.061 and 0.547 ± 0.053 ; Cohen’s $d = 1.274$ and 0.958) and 18.0% and 17.8% over the structure-based PPB-Affinity model (0.545 ± 0.072 and 0.540 ± 0.088 ; Cohen’s $d = 1.326$ and 1.064).

Table S9. Statistical comparison between PPLM-Affinity and baseline models (ESM2-Affinity and PPB-Affinity) across five-fold cross-validation. For each dataset and metric, the table reports the fold-wise mean difference (Δ), relative percentage improvement, and paired effect size (Cohen’s d).

Dataset	PCC			SRCC			RMSE		
	Mean Δ	Percent gain (%)	Cohen’s d	Mean Δ	Percent gain (%)	Cohen’s d	Mean Δ	Percent gain (%)	Cohen’s d
vs ESM2-Affinity									
Entire dataset	0.095	17.3	1.274	0.09	16.4	0.958	0.164	6.6	0.719
Antibody-antigen subgroup	0.205	117.1	1.45	0.213	111.5	1.645	0.137	6.7	0.96
TCR-pMHC subgroup	0.216	144.0	0.706	0.175	127.7	1.421	0.465	22.2	0.893
vs PPB-Affinity									
Entire dataset	0.097	18.0	1.326	0.096	17.8	1.064	0.152	6.2	0.62
Antibody-antigen subgroup	0.143	60.3	1.163	0.147	57.2	1.14	0.09	4.5	0.705
TCR-pMHC subgroup	0.103	39.2	0.394	0.084	36.8	0.25	0.209	11.4	0.522

Page 6:

Figure 4B presents results on heterodimer test sets. On Heterodimer99, PPLM-Contact achieves 48.9% precision using experimental monomers and 45.1% using AlphaFold2-predicted monomers. These correspond to improvements of 37.0%–182.7% over competing methods, all statistically significant. On CASP Heterodimer20, PPLM-Contact attains 45.6% precision with experimental monomer structures and 40.5% with AlphaFold2 monomers, again outperforming all

baseline approaches. Comprehensive statistical analyses, including paired Cohen's d effect sizes and Benjamini–Hochberg false-discovery-rate (FDR) correction across the four test sets, are provided in **Supplementary Text S4**.

Table S12. Comprehensive statistical comparison between PPLM-Contact and baseline models (PLMGraph-Inter, DeepInter, CDPred, DeepHomo2.0, and GLINTER) across four test datasets (Homodimer300, Heterodimer99, CASP_Homo43, and CASP_Hetero20) using distance maps derived from AlphaFold2-predicted monomer structures. For each dataset and baseline, improvements in per-target top L contact precision were evaluated using mean difference (Δ), relative percent gain, paired effect size (Cohen's d), Wilcoxon signed-rank p -values, and Benjamini–Hochberg FDR-adjusted q -values. Statistically significant results ($q < 0.05$) are indicated.

Dataset	Mean Δ	Percent gain (%)	Effect size (Cohen's d)	Wilcoxon p-value	FDR q-value	Significant ($q < 0.05$)
vs PLMGraph-Inter						
Homodimer300	22.52	51.1	0.88	3.92E-33	1.57E-32	✓
Heterodimer99	20.07	80.2	0.67	2.39E-08	4.77E-08	✓
CASP_Homodimer43	15.29	39.5	0.51	1.53E-03	2.04E-03	✓
CASP_Heterodimer20	17.20	73.9	0.53	4.95E-02	4.95E-02	✓
vs DeepInter						
Homodimer300	6.27	10.4	0.35	3.96E-13	1.59E-12	✓
Heterodimer99	12.18	37.0	0.43	6.10E-05	1.22E-04	✓
CASP_Homodimer43	6.17	12.9	0.27	1.30E-02	1.30E-02	✓
CASP_Heterodimer20	10.78	36.3	0.68	4.29E-03	5.71E-03	✓
vs CDPred						
Homodimer300	10.54	18.8	0.48	3.43E-21	1.37E-20	✓
Heterodimer99	17.64	64.2	0.55	1.69E-06	3.38E-06	✓
CASP_Homodimer43	17.09	46.4	0.66	1.51E-04	2.02E-04	✓
CASP_Heterodimer20	20.81	105.9	0.65	2.00E-02	2.00E-02	✓
vs DeepHomo2.0						
Homodimer300	22.43	50.8	0.89	1.68E-37	3.37E-37	✓
CASP_Homodimer43	22.63	72.3	0.84	1.02E-05	1.02E-05	✓
vs GLINTER						
Homodimer300	35.85	116.7	1.25	1.30E-42	5.19E-42	✓
Heterodimer99	28.28	167.9	0.88	1.29E-11	2.58E-11	✓
CASP_Homodimer43	31.48	140.0	1.09	5.33E-07	7.11E-07	✓
CASP_Heterodimer20	33.52	483.3	0.94	2.26E-03	2.26E-03	✓

Page 7:

Figure 5C–D summarize the top L precision of all general contact-prediction methods (using AlphaFold2-predicted monomer structures) and the complex structure-based methods (AlphaFold2.3, AlphaFold3, DMFold, and PPLM-Contact2) on the 434 homodimer and 119 heterodimer proteins. As expected, the integration of complex-structure features allows PPLM-Contact2 to markedly improve upon PPLM-Contact, increasing top L precision from 65.0% to 85.1% on homodimers and from 44.3% to 88.0% on heterodimers. Compared with structure-modeling approaches, PPLM-Contact2 achieves improvements of 4.8% and 7.6% over AlphaFold2.3, 8.7% and 5.6% over AlphaFold3, and 5.6% and 8.1% over DMFold on homodimers and heterodimers. All improvements are statistically significant (**Supplementary Table S15**).

Table S15. Comprehensive statistical comparison between PPLM-Contact2 and complex structure-based models (AlphaFold2.3, AlphaFold3, and DMFold). For each dataset and baseline, improvements in per-target top L contact precision were evaluated using mean difference (Δ), relative percent gain, paired effect size (Cohen’s d), Wilcoxon signed-rank p -values, and Benjamini–Hochberg FDR-adjusted q -values. Statistically significant results ($q < 0.05$) are indicated.

Dataset	Mean Δ	Percent gain (%)	Effect size (Cohen’s d)	Wilcoxon p -value	FDR q -value	Significant ($q < 0.05$)
vs AlphaFold2.3						
Homodimers	3.93	0.05	0.26	2.42E-18	4.84E-18	✓
Heterodimers	6.20	0.08	0.36	2.01E-10	2.01E-10	✓
vs AlphaFold3						
Homodimers	6.77	0.09	0.34	1.62E-23	3.24E-23	✓
Heterodimers	4.65	0.06	0.25	4.60E-05	4.60E-05	✓
vs DMFold						
Homodimers	4.53	0.06	0.28	1.34E-23	2.68E-23	✓
Heterodimers	6.57	0.08	0.40	5.51E-11	5.51E-11	✓

Page 8:

Among general contact-prediction methods, PPLM-Contact achieves the highest average precision, reaching 0.824 for homodimers and 0.695 for heterodimers. These values represent improvements of 5.0% and 12.1% over DeepInter, 7.6% and 7.1% over PLMGraph-Inter, 5.3% and 14.2% over CDPred, 19.3% over DeepHomo2.0 (homomers), and 28.7% and 39.7% over GLINTER. PPLM-Contact also attains the highest median precision (0.916 and 0.752), indicating consistently strong predictions across targets. All improvements are statistically significant (**Supplementary Table S16**). Integrating complex-structure features further boosts performance: PPLM-Contact2 markedly improves upon PPLM-Contact, increasing average and median precision to 0.897 and 0.962 for homodimers and 0.904 and 0.945 for heterodimers. Compared with structure-based methods, PPLM-Contact2 achieves the highest average and median precision across both homodimer and heterodimer test sets. Comprehensive statistical analyses confirm that these improvements are significant (**Supplementary Table S18**).

Table S16. Precision of interface residue identification by PPLM-Contact, DeepInter, CDPred, DeepHomo2.0 and GLINTER on the 343 homodimer and 119 heterodimer test proteins, using AlphaFold2-predicted monomer structures. The improvements of PPLM-Contact over other baseline methods were evaluated using mean difference (Δ), relative percent gain, paired effect size (Cohen’s d), Wilcoxon signed-rank p -values, and Benjamini–Hochberg FDR-adjusted q -values. Statistically significant results ($q < 0.05$) are indicated.

Dataset	Average	Median	Mean Δ	Percent gain (%)	Effect size (Cohen’s d)	Wilcoxon p -value	FDR q -value	Significant ($q < 0.05$)
Homodimer								
PPLM-Contact	0.824	0.916	-	-	-	-	-	-
DeepInter	0.784	0.892	0.039	5.0	0.253	7.82E-16	1.56E-15	✓
PLMGraph-Inter	0.765	0.847	0.058	7.6	0.408	1.75E-21	3.50E-21	✓
CDPred	0.782	0.869	0.041	5.3	0.173	3.29E-07	3.29E-07	✓
DeepHomo2.0	0.691	0.779	0.133	19.3	0.622	3.51E-34	3.51E-34	✓
GLINTER	0.640	0.679	0.184	28.7	0.990	8.31E-43	1.66E-42	✓
Heterodimer								
PPLM-Contact	0.695	0.752	-	-	-	-	-	-
DeepInter	0.620	0.649	0.075	12.1	0.370	2.04E-04	2.04E-04	✓
PLMGraph-Inter	0.649	0.674	0.046	7.1	0.288	2.12E-04	2.12E-04	✓
CDPred	0.608	0.679	0.087	14.2	0.409	1.63E-05	1.63E-05	✓
GLINTER	0.497	0.486	0.198	39.8	1.080	7.45E-18	7.45E-18	✓

10. The Reviewer commented:

10. Analyze Failure Cases and Model Limitations: The manuscript lacks analysis of failure modes (e.g., poor performance on specific heterodimers) and model limitations (e.g., underrepresentation of transient interactions). Include case studies with structural visualizations and discuss potential biases in the discussion section.

Response: We thank the Reviewer for this valuable suggestion. In the revised manuscript, we have added a failure case to characterize the limitations of PPLM-Contact in the Discussion section. **Supplementary Figure 13** presents this failure case involving the nanobody–antigen complex formed between the CD9 EC2 domain and nanobody 4C8 (PDB ID: 6Z20). This interaction features a small, loop-mediated binding interface with weak co-evolutionary signal, a class of interactions that is intrinsically challenging for sequence-based contact predictors.

We have added the following paragraph and figure to clarify the limitations (Page 9):

Page 9:

Despite its strong performance, PPLM has **several** limitations that warrant further investigation. Its training relies on a composite **sequence-pair** dataset that, while diverse, may not fully capture the breadth of protein interactions across organisms and cellular states. In particular, transient, weak, or condition-dependent interactions remain underrepresented. A representative failure case is shown in **Supplementary Figure S13**, involving a nanobody–antigen complex with a very small, loop-mediated interface and weak co-evolutionary signal. Although PPLM-derived features partially compensate for the lack of MSA or structural information, they remain insufficient to recover these small, flexible, and weakly co-evolving interfaces, highlighting an intrinsic limitation of sequence-based contact prediction.

Figure S13:

Figure S13. A representative challenging heterodimer for inter-protein contact prediction: the EC2 domain of CD9 in complex with nanobody 4C8 (PDB ID: 6Z20). (A) Structural visualization of the complex with interface residues highlighted. (B–C) Inter-protein contact maps predicted by PPLM-Contact w/o PPLM, PPLM-Contact w/o MSA, and the full PPLM-Contact model, respectively. In each panel, gray dots indicate ground-truth contacts, red dots denote correctly predicted contacts (true positives), and blue dots represent false positives. For each method, the number in parentheses indicates the top N contact precision, where N equals the number of ground-truth contacts.

11. The Reviewer commented:

11. Update Benchmark Models: The chosen baseline models (e.g., D-SCRIPT, Topsy-Turvy) are not sufficiently representative of recent advances. Include comparisons with cutting-edge 2024–2025 methods (e.g., AlphaFold3 variants or GLINTER extensions) to strengthen the evaluation.

Response: We thank the Reviewer for this helpful suggestion. In the revised manuscript, we have incorporated two recent state-of-the-art baselines with publicly available implementations: TUnA (Briefings in Bioinformatics, 2024) for PPI prediction and PLMGraph-Inter (eLife, 2024) for inter-protein contact prediction.

Across the five species PPI test sets, PPLM-PPI consistently outperforms TUnA and all other comparison methods in precision, recall, accuracy, F1-score, and AUPRC. Compared with TUnA, PPLM-PPI improves F1-score by 10.1% and AUPRC by 9.6%, with large effect sizes (Cohen's $d \approx 1.742$ – 2.268) (**Figure 3A–C** and **Supplementary Tables S1–S3**).

Across four inter-protein contact benchmark test sets, PPLM-Contact achieves significantly higher top L precision than PLMGraph-Inter, with average gains of 6–12 percentage points (10–37% relative improvement) and moderate effect sizes (Cohen's $d \approx 0.27$ – 0.68). All comparisons remain statistically significant after Benjamini–Hochberg correction (FDR $q < 0.05$) (**Figure 4A–C** and **Supplementary Tables S10–S12**).

We have revised the relevant paragraphs to incorporate these new baseline comparisons (Pages 4 and 6):

Page 4:

PPLM-PPI outperforms existing methods in binary protein interaction prediction

To evaluate the effectiveness of PPLM-PPI on protein–protein interaction predictions, we benchmarked it across five species, including *M. musculus*, *D. melanogaster*, *C. elegans*, *S. cerevisiae*, and *E. coli*. PPLM-PPI was compared against four state-of-the-art LM-based PPI predictors, including TUnA⁴¹, ESMDNN-PPI²³, D-SCRIPT¹⁵, and Topsy-Turvy²². All methods were installed and executed locally for reproducibility, except ESMDNN-PPI, which we reimplemented following the original publication due to the lack of publicly available source code.

Comprehensive evaluation results, including precision, recall, accuracy, F1-score, area under the receiver operating characteristic curve (AUROC), and area under the precision–recall curve (AUPRC) for all species, are provided in **Supplementary Table S1**. PPLM-PPI achieved the highest performance on nearly all metrics across all datasets. Since AUPRC is especially informative for imbalanced PPI datasets, we use it as the primary metric. As summarized in **Figure 3A**, PPLM-PPI achieved AUPRC scores of 0.920, 0.906, 0.882, 0.5745, and 0.784 on the five species, corresponding to improvements of 4.0%, 6.2%, 7.2%, 17.6%, and 12.9% over the second-best method, TUnA. Similarly, PPLM-PPI achieved the highest F1-score for all species, surpassing TUnA by 4.8%, 8.6%, 4.8%, 16.9%, and 15.5% (**Figure 3B**). Precision–recall curves across species further show that PPLM-PPI consistently dominates other methods across most recall thresholds (**Figure 3C**). In addition, the curves of PPLM-PPI are markedly smoother than those of competing methods, indicating superior stability and predictive consistency, particularly in high-precision and high-recall regimes.

An overall analysis across the five species test sets showed that PPLM-PPI demonstrated superior stability, with a mean \pm standard deviation of 0.848 ± 0.078 , compared to 0.778 ± 0.109 for TUnA, 0.762 ± 0.106 for ESMDNN-PPI, 0.532 ± 0.072 for D-SCRIPT, and 0.548 ± 0.118 for Topsy-Turvy (**Supplementary Table S2**). **Supplementary Table S3** details the statistical comparisons. Relative to TUnA and ESMDNN-PPI, PPLM-PPI improves F1-score by 10.1–10.5% and AUPRC by 9.6–11.9%, with large effect sizes (Cohen's $d \approx 1.742$ – 6.586). Against D-SCRIPT and Topsy-Turvy, the improvements are even larger: 32.5–72.3% (F1) and 58.7–60.9% (AUPRC), also with consistently large effect sizes (Cohen's $d \approx 2.062$ – 6.183).

Page 6:

PPLM-Contact outperforms existing methods in inter-protein contact prediction

We evaluated PPLM-Contact on four benchmark datasets: Homodimer300 and Heterodimer99 from DeepInter20, and CASP_Homodimer43 and CASP_Heterodimer20. Comparisons were made against five state-of-the-art methods: PLMGraph-Inter³⁴, DeepInter²⁰, CDPred¹⁹, GLINTER³⁵, and DeepHomo2.0³². All baseline methods were installed and executed locally to ensure reproducibility. Inter-protein contact precision was assessed at multiple cutoffs (top 1, 10, 50, $L/10$, $L/5$, and L), where L is the length of the shorter protein chain in the complex. Summary statistics and paired

comparisons are provided in **Supplementary Table S10–13**, with per-protein results in **Supplementary Data 1–2**.

Figure 4A reports the top L contact precision on homodimer test sets using either experimental or AlphaFold2-predicted monomer structures. On Homodimer300 with experimental structures, PPLM-Contact achieves 77.8% precision, substantially outperforming DeepInter (68.9%), PLMGraph-Inter (50.6%), CDPred (63.0%), DeepHomo2.0 (51.4%), and GLINTER (34.5%) by margins of 12.8%, 53.7%, 23.5%, 51.3%, and 125.3%. All comparisons are statistically significant (Wilcoxon signed-rank test $p = 3.63 \times 10^{-21}$ to 1.29×10^{-47}). Using AlphaFold2-predicted monomer structures, PPLM-Contact maintains strong performance (66.6%) and surpasses the second-best method, DeepInter, by 10.4% ($p = 3.96 \times 10^{-13}$). A similar trend is observed on CASP_Homodimer43. With experimental monomer structures, PPLM-Contact achieves 65.2% precision and improves over baseline methods by 6.0%–157.7%. When using AlphaFold2-predicted monomer structures, the corresponding improvements are 12.9%, 39.5%, 46.3%, 72.3%, and 140.0%, respectively.

Figure 4B presents results on heterodimer test sets. On Heterodimer99, PPLM-Contact achieves 48.9% precision using experimental monomers and 45.1% using AlphaFold2-predicted monomers. These correspond to improvements of 37.0%–182.7% over competing methods, all statistically significant. On CASP_Heterodimer20, PPLM-Contact attains 45.6% precision with experimental monomer structures and 40.5% with AlphaFold2 monomers, again outperforming all baseline approaches. Comprehensive statistical analyses, including paired Cohen’s d effect sizes and Benjamini–Hochberg false-discovery-rate (FDR) correction across the four test sets, are provided in **Supplementary Text S4**.

Figure 4C provides head-to-head comparisons between PPLM-Contact and each baseline method across all 434 homodimer and 119 heterodimer proteins using experimental monomer structures. Relative to DeepInter, PPLM-Contact achieves higher / lower top L contact precision on 214 / 48 homodimer proteins (62.4% / 14.0%) and 79 / 22 heterodimer proteins (66.4% / 18.5%). The corresponding higher / lower proportions are 82.5% / 12.0% and 63.0% / 24.4% for PLMGraph-Inter, 80.8% / 12.2% and 65.5% / 20.2% for CDPred, 87.2% / 8.2% for DeepHomo2.0, and 91.0% / 6.7% and 78.2% / 13.4% for GLINTER. These results indicate that PPLM-Contact achieves higher performance on a clear majority of targets, while also making it explicit that a non-trivial fraction of predictions is tied rather than strictly lower. **Supplementary Figure S7** presents the corresponding comparisons using AlphaFold2-predicted monomer structures, showing a similar trend in which PPLM-Contact outperforms baseline methods for most targets.

Figure 3A-C:

Figure 3. Performance of PPLM-PPI and PPLM-Affinity on protein-protein interaction and binding affinity prediction. (A) AUPRC of PPLM-PPI, TUNA, ESM-DNN-PPI, D-SCRIPT, and Topsy-Turvy across five species test sets. **(B)** F1-score of the five PPI prediction methods on the five species test sets. **(C)** Precision-recall curves of the five PPI prediction methods on the five species test sets.

Figure 4A-C:

Figure 4. Performance of PPLM-Contact and the comparison methods on the inter-protein contact prediction. (A) Top L contact precision of PPLM-Contact, DeepInter, CDPred, DeepHomo2.0, and GLINTER on the Homodimer300 and CASP_Homodimer43 test sets using both experimental monomer structures and AlphaFold2-predicted monomer structures. (B) Top L contact precision of PPLM-Cxcontact, DeepInter, CDPred, and GLINTER on the Heterodimer99 and CASP_Heterodimer20 test sets using both experimental monomer structures and AlphaFold2-predicted monomer structures. (C) Head-to-head comparison of top L contact precision between PPLM-Contact and other methods on the 343 homodimer and 119 heterodimer proteins.

12. The Reviewer commented:

12. Analyze Impact of MSA Depth: The influence of multiple sequence alignment (MSA) depth on PPLM-Contact's performance is not evaluated. Assess how varying MSA depths (e.g., shallow vs. deep alignments from UniRef30) affect contact prediction accuracy, particularly for heterodimers, and report results in a supplementary table or figure.

Response: We thank the Reviewer for this valuable suggestion. In the revised manuscript, we have conducted a detailed analysis of the relationship between MSA depth (N_{eff}) and top L inter-protein contact precision for PPLM-Contact, using UniRef30-derived paired MSAs from 443 homodimers and 119 heterodimers. The results are shown in **Supplementary Figure S8**, which separately report the trends for heterodimers, homodimers, and the combined dataset. The detailed analysis has been provided in **Supplementary Text S6**.

We have added the following paragraphs to clarify the insights (Page 6 and SI):

Page 6:

Figure 4D–F summarize the mean and standard deviation of top L precision across three random seeds for PPLM-Contact and all ablation variants. These results show that removing any major network component or feature of the model reduces precision, with the largest performance drops observed when excluding the triangle-multiplication module, PPLM inter-protein attention, MSA features, or monomer distance maps. Detailed ablation analyses and statistical comparisons are provided in **Supplementary Text S5**. **The effect of MSA depth is further examined in Supplementary Text S6.**

Figure S8:

Figure S8. Relationship between MSA depth and inter-protein contact-prediction precision for PPLM-Contact. (A–B) Heterodimers, (C–D) homodimers, and (E–F) all test complexes. Bars show mean top L precision across MSA-depth bins with 95% confidence intervals, and scatter plots display precision versus $\log_{10}(N_{eff})$.

Supplementary Text S6. Analysis of the impact of MSA depth on PPLM-Contact performance:

To evaluate the effect of MSA depth on model performance, we analyzed the relationship between the effective MSA depth (N_{eff}) and the top L inter-protein contact prediction precision of PPLM-Contact. Results for heterodimers (Heterodimer99 + CASP_Heterodimer20), homodimers (Homodimer300 + CASP_Homodimer43), and the combined dataset are shown in **Supplementary Figure S8**.

For heterodimers, which generally have shallower paired MSAs, precision shows a moderate positive correlation with N_{eff} (Pearson's $r = 0.23$, $p = 1.1 \times 10^{-2}$). As shown in Fig. SxA–B, the average precision increases from ~30% at $N_{eff} < 1$ to ~80% at N_{eff} 30–60, indicating that PPLM-Contact can extract useful co-evolutionary information even from moderately deep alignments. Although a positive trend is observed, the explanatory power remains limited ($R^2 \approx 0.05$), suggesting that MSA depth alone does not fully account for performance variation. For homodimers, which typically have richer MSAs, no significant correlation was observed between N_{eff} and precision ($r = 0.05$, $p = 0.37$; Fig. SxC–D). Precision remains consistently high across all depth bins, averaging 70–90% and reaching nearly 100% for $N_{eff} \geq 200$, indicating that further MSA enrichment yields limited additional gains. When all test complexes are analyzed together (Fig. SxE–F), PPLM-Contact achieves a moderate positive correlation between N_{eff} and precision ($r = 0.27$, $p = 2.7 \times 10^{-9}$), with precision rising from ~48% at $N_{eff} < 1$ to ~76–91% when $N_{eff} > 100$, albeit with large target-to-target variability ($R^2 \approx 0.07$).

The difference between heterodimers and homodimers may partly reflect how paired MSAs are constructed for these two oligomer types. For homodimers, inter-protein features are typically

derived by duplicating the same single-chain MSA for both subunits, so increasing alignment depth may reinforce intra-chain rather than cross-chain coevolutionary signals, leading to performance saturation. By contrast, heterodimers rely on paired alignments of distinct proteins, where deeper MSAs can introduce additional cross-chain evolutionary constraints.

Overall, MSA depth shows a modest positive association with inter-protein contact prediction accuracy. This relationship may vary between oligomer types, with heterodimers benefiting moderately from deeper paired MSAs, whereas homodimers maintain consistently high accuracy across all depth ranges.

13. The Reviewer commented:

13. From my experience, there exists a big issue, short-cut learning, for pair-input models, As shown in previous studies (PMID: 37031187, 38030641), the pair-input models do not work well for unseen proteins in the training set, where none of the proteins in the test set overlap with the training set. The authors should evaluate the inductive setting and compare with existing methods.

Response: We thank the Reviewer for highlighting the important issue of shortcut learning in pair-input models and for directing us to the related studies (PMID: 37031187 and 38030641). These works focus on protein–ligand or drug–target interaction prediction, where the same proteins or ligands appear in many training pairs and the annotations are highly imbalanced. As shown in AI-Bind (PMID: 37031187), this can lead to node-degree shortcuts, where models rely on how frequently a protein or ligand has positive annotations rather than on its molecular features. ZeroBind (PMID: 38030641) further demonstrates that end-to-end concatenation-based models may overfit protein identities and molecular scaffolds, resulting in poor generalization to unseen targets.

In contrast, our PPLM-PPI setting is inherently different: the model is trained only on human PPIs and evaluated on five non-human species (mouse, fly, worm, yeast, and *E. coli*), so that none of the test proteins ever appear in the supervised training set. Thus, the specific shortcut mechanisms based on reusing the same protein nodes across training and test pairs, as discussed in AI-Bind and ZeroBind, cannot arise in our cross-species formulation. Nevertheless, we fully agree that it is important to directly examine inductive performance as a function of similarity to the training set.

In the revised manuscript, we have added an additional similarity-stratified analysis. For every protein pair in each of the five test species, we computed its maximum single-sequence identity to any human training protein (with alignment coverage ≥ 0.8). These protein pairs were then grouped test interaction pairs into identity bins: (0, 0.5], (0.5, 0.6], (0.6, 0.7], (0.7, 0.8], (0.8, 0.9], (0.9, 1.0), and [1.0], according to the higher identity of its two constituent proteins, which reflects the maximal similarity between any protein in the pair and the training set (**Supplementary Text S3**).

We have added the following paragraphs and figures to clarify these insights and have incorporated citations to the two studies (PMID: 37031187 and 38030641) in the revised manuscript to acknowledge prior discussions of shortcut learning in pair-input models (Page 5 and SI):

Page 5:

Finally, to assess inductive generalization and evaluate whether shortcut mechanisms might arise in our paired-input setting^{42,43}, we performed a sequence-similarity–stratified evaluation. Test interactions were grouped by the maximum single-sequence identity of either protein to any human protein in the training set, and all methods were compared within each identity interval. As detailed in **Supplementary Text S3**, PPLM-PPI maintains strong performance across all homology ranges and delivers the largest gains over existing methods, demonstrating that its predictive accuracy is

not driven by shortcut learning and that the model generalizes robustly to unseen proteins under a strict cross-species inductive setting.

Figure S5:

Figure S5. AUPRC of PPLM-PPI and baseline methods across sequence-identity bins for five species test sets. (A) *M. musculus*. (B) *D. melanogaster*. (C) *C. elegans*. (D) *S. cerevisiae*. (E) *E. coli*.

Supplementary Text S3. Inductive generalization analysis under sequence-similarity stratification for PPI prediction:

To evaluate whether the performance of PPLM-PPI could be affected by shortcut learning and to directly assess its inductive generalization ability, we conducted a sequence-similarity-stratified analysis across all five non-human test species (mouse, fly, worm, yeast, and *E. coli*). For every protein in each test organism, we computed its maximum sequence identity to any human protein in the training set (alignment coverage ≥ 0.8). Test interaction pairs were then grouped into seven identity bins: (0, 0.5], (0.5, 0.6], (0.6, 0.7], (0.7, 0.8], (0.8, 0.9], (0.9, 1.0), and [1.0], according to the higher identity of its two constituent proteins, which reflects the maximal similarity between any protein in the pair and the training set. For each bin, we evaluated the AUPRC of PPLM-PPI and all baseline models (TUN-A, ESM-based PPI, D-SCRIPT, and Topsy-Turvy), and recorded the negative-to-positive sample ratio to quantify class imbalance (**Supplementary TableS6 and Supplementary Figure S5**).

Across species, the lowest-identity (0, 0.5] bin typically contains the strongest class imbalance, with negative–positive ratios ranging from 8.5:1 (*E. coli*) to 86.4:1 (mouse). Higher-identity bins are progressively less imbalanced, except for the small [1.0] bin in *D. melanogaster*, which is highly skewed due to limited sample size. This pattern indicates that raw AUPRC in extremely low-identity bins reflects the combined effect of homology and class imbalance rather than shortcut learning.

Despite large differences in class imbalance and sequence identity, PPLM-PPI maintains robust performance across all bins. In all five species, PPLM-PPI achieves high AUPRC for identity > 0.5 and does not exhibit degradation when identity decreases. In *D. melanogaster* and *C. elegans*, PPLM-PPI even attains higher AUPRC values in the (0.6, 0.9] bins than in the (0.9, 1.0) bin, showing that performance does not strictly track sequence similarity to the human training set. In

the *E. coli* dataset, where all test proteins exhibit sequence identities below 0.8 relative to the human training set, PPLM-PPI also continues to deliver strong predictive accuracy.

Crucially, PPLM-PPI consistently outperforms all baseline methods in nearly every identity bin, including the most remote (0, 0.5] range. In this bin, PPLM-PPI improves AUPRC over TUN-A by 26.0%, 12.7%, 13.9%, 33.4%, and 13.3% on the mouse, fly, worm, yeast, and *E. coli* datasets, respectively. Similar or larger improvements are observed relative to ESM-based PPI, D-SCRIPT, and Topsy-Turvy. These results demonstrate that the advantage of PPLM-PPI is greatest in the most challenging remote-homology regime, where shortcut learning—if present—would be least effective.

Taken together, this similarity-stratified evaluation shows that (i) the dependence of AUPRC on identity is largely driven by underlying class imbalance and biological conservation and is shared by all models; (ii) PPLM-PPI maintains strong and stable predictive accuracy across sequence-identity ranges; and (iii) the method does not rely on protein-identity shortcuts, but instead generalizes robustly to unseen proteins in a strict cross-species inductive setting.

Response to Reviewer #2's comments

We sincerely thank the Reviewer for the insightful comments, which have substantially improved the quality of our manuscript. A major concern involved the interpretability and distinctiveness of PPLM's representations. In response, we added direct comparisons with ESM2 using unsupervised and linear-probe analyses, together with a structural case study demonstrating biological relevance. The Reviewer also noted issues in statistical rigor and potential data leakage; we therefore reconstructed the affinity cross-validation using structure-based clustering, retrained all methods, and reported fold-wise mean \pm s.d., effect sizes, and FDR-corrected significance. Point-by-point responses are detailed below, and all revisions in the manuscript are highlighted in yellow.

1. The Reviewer commented:

The authors present a language model, PPLM, that embeds inter- and intraprotein interactions based on sequence information, and later incorporating structure information as well. They demonstrate that this model is supportive of applications for predicting binary protein interactions, predicting PPI affinity, and predicting amino acids that form intermolecular contacts. In their results, the authors demonstrate that their own techniques, which employ PPLM, can outperform single-purpose methods designed for one of these applications.

1. Overall, the presentation of the method is good and fairly clear. The experimental methodology, with regard to machine learning techniques is reasonably well designed, and the datasets are very comprehensive. Data and code have been shared publicly, though this Reviewer is uncertain if everything needed for reproduction has been shared.

Response: We thank the Reviewer for the positive assessment of our work. We appreciate the comment regarding reproducibility. In the revised manuscript, we have carefully ensured that all resources required to reproduce our results are made publicly available and clearly referenced in the **Data availability** and **Code availability** sections.

2. The Reviewer commented:

Major comments:

2. To this Reviewer, the major contribution of the paper is that PPLM and its successor methods may offer a one-size-fits-all model that can support a class of techniques that better perform the comparison methods tested in the paper, and possibly others as well. It suggests, as the authors say, that something is being learned by PPLM in the embedding of inter- and intra-protein interactions that enables this outperformance and makes it accessible to downstream applications. However, the authors only offer speculation as to what that is. This is not an explainable AI method, so the authors cannot really introspect on the nature of the system, but a comparison of what is being represented by existing methods, and how it differs from PPLM, should at least be offered. It is not clear that inter-protein interactions are not represented at all in existing methods, but perhaps in a less explicit way than in PPLM. The ablation study was a good start, but a point by point comparison should be made in supplementary data.

Response: We thank the Reviewer for this insightful suggestion. In the revised manuscript, we have added a new analyses that directly probe how inter-protein attentions from PPLM versus ESM2 relate to true inter-chain contacts under two complementary settings: (i) **unsupervised**, by averaging the last-layer inter-protein attention to predict contacts; and (ii) **linear-probe**, by training a single frozen linear layer on all-layer inter-protein attentions on the training set of PPLM-Contact (5 epochs, learning rate = 0.001) (**Supplementary Text S1**). We report top k (ranging from top 1 to top L , where L is the length of the shorter protein in the dimer) inter-

protein contact precision of PPLM and ESM2 on the Homodimer300 and Heterodimer99 test sets, together with paired Wilcoxon signed-rank tests (one-sided, PPLM > ESM2), as shown in **Supplementary Figure S2**.

To further illustrate the biological relevance of PPLM's attention representations, we added a case study on a conserved transcription factor from *Vibrio cholerae* O1 biovar El Tor (PDB ID: 1Y9B). The inter-protein attention matrix, obtained by averaging last-layer attention across all heads, aligns closely with the experimentally determined contact map and correctly identify 83.9% of interface residues (**Figure 2D–E**). This case highlights that PPLM's inter-protein attentions capture spatially meaningful interaction patterns, supporting the interpretability of the model.

We have added the following paragraphs and figures to clarify these insights (Page 3 and SI):

Page 3:

To examine whether PPLM captures biologically meaningful dependencies, we analyzed the homodimer complex 1Y9B, a conserved transcription factor from *Vibrio cholerae* O1 biovar El Tor (**Figure 2D–F**). The inter-protein attention matrix from the final PPLM layer showed high-scoring regions that closely matched the experimental contact map. Among the top 20 residue pairs ranked by attention, 90% corresponded to true heavy-atom contacts (80% to C_β–C_β contacts). Of the 62 experimentally determined interface residues, 52 (83.9%) were recoverable from attention-derived residue rankings. The corresponding 3D visualization demonstrated that these predicted residues cluster around the physical interface, indicating that PPLM's attention mechanism naturally focuses on interaction-relevant regions during unsupervised learning. A more detailed representation-level comparison between PPLM and ESM2, including unsupervised and linear-probe analyses of inter-protein attention matrices, is provided in **Supplementary Text S1**.

Supplementary Text S1. Inter-protein attention matrix analysis of PPLM and ESM2.

To further investigate what is captured by PPLM beyond existing single-protein language models, we conducted a representation-level comparison between PPLM and ESM2. Specifically, we examined how their inter-protein attention matrices relate to true inter-chain contacts under two complementary settings: (i) **Unsupervised**, where the last-layer inter-protein attention was averaged across heads and layers to directly predict contact likelihoods; and (ii) **Linear-probe**, where a single frozen linear layer was trained on all-layer inter-protein attentions using the training set of PPLM-Contact (5 epochs, learning rate = 0.001).

Performance was evaluated on the Homodimer300 and Heterodimer99 test sets in terms of top k inter-protein contact precision ($k \in \{1, 5, 10, 25, 50, 100, L/10, L/5, L/2, L\}$, where L is the length of the shorter protein in each complex). Paired Wilcoxon signed-rank tests (one-sided, PPLM > ESM2) were used to assess statistical significance. Error bars in **Figure S2** represent the standard error (SE) across complexes, and significance levels are annotated as $p < 0.05$ (*), $p < 0.01$ (**), and $p < 0.001$ (***)

On the Homodimer300 test set, PPLM shows consistent and statistically significant gains over ESM2 across all top k thresholds. In the unsupervised last-attention setting, the mean improvements (PPLM–ESM2, percentage points) are +7.33 (top 1), +6.87 (top 5), +5.23 (top 10), +4.23 (top 25), +3.18 (top 50), +2.03 (top 100), +4.53 (top $L/10$), +3.59 (top $L/5$), +2.52 (top $L/2$), and +1.78 (top L); all are significant according to paired Wilcoxon tests (***, $p < 0.001$). With the single linear probe, gains remain robust—+6.67, +6.40, +5.57, +4.47, +3.31, +2.20, +4.43, +4.14, +2.37, +1.49 for top 1 to top L —with statistical significance at $p < 0.01$ (**) or $p < 0.001$ (***). Specifically, PPLM achieves 6.45% and 11.68% top L precision in the unsupervised and linear-probe settings, representing 38.4% and 14.7% relative improvements over ESM2 (4.66% and 10.18%). At the per-complex level (top L evaluation), PPLM outperformed ESM2 on 195 homodimers, tied on 24, and underperformed on 81 in the last-layer-attention setting; the corresponding numbers for the linear-probe setting are 161, 20, and 119.

On the Heterodimer99 test set, in the unsupervised last-attention setting, PPLM performs lower than ESM2 at top 1 (–1.52 points) but surpasses it from top 10 onward (e.g., +1.77 at top 10, +1.03 at top $L/5$, +0.41 at top L). Correspondingly, Wilcoxon tests are not significant for top 1, top 5, and

top $L/10$, but become significant from top 25 onward (except top $L/10$), indicating a gradual emergence of representational advantage as k increases. With the single linear probe, PPLM improves by +4.04 (top 1), +2.22 (top 5), +2.22 (top 10), +2.42 (top 25), +1.76 (top 50), +1.67 (top 100), +2.32 (top $L/10$), +1.95 (top $L/5$), +1.89 (top $L/2$), and +1.74 (top L). Statistical significance is observed at top 25, top 50, top 100, top $L/10$, top $L/2$, and top L ($p < 0.05 / 0.01 / 0.001$, indicated as */**/**), while top $L/5/10$ and top $L/5$ show no formal significance but still exhibit notable average precision gains. Specifically, PPLM achieves 2.76% and 6.44% top L precision in the unsupervised and linear-probe settings, representing 17.5% and 27.0% relative improvements over ESM2 (2.35% and 4.70%). At the per-complex level, PPLM outperformed ESM2 on 54 heterodimers, tied on 23, and underperformed on 22 in the last-layer-attention setting; for the linear-probe setting, the corresponding numbers are 51, 28, and 20.

These results demonstrate that the improvement provided by PPLM is broadly distributed across both homodimer and heterodimer proteins, with consistent advantages in the majority of cases. Although the gains are smaller and less significant at top 1 for heterodimers in the unsupervised setting, PPLM consistently achieves higher precision as k increases. These findings indicate that PPLM's inter-protein attention encodes more explicit and linearly decodable patterns of inter-protein interactions than ESM2, confirming that paired-protein language modeling captures deeper relational representations beyond single-chain embeddings.

Figure S2:

Figure S2. Comparative analysis of inter-protein attentions from PPLM and ESM2. (A) Results on the Homodimer300 test set. (B) Results on the Heterodimer99 test set. Bar plots show top k inter-protein contact precision (%) for PPLM and ESM2 under two settings: unsupervised (last-layer mean attention) and linear-probe (all-layer attention + frozen linear layer). Error bars indicate standard errors (SE) across proteins. Asterisks denote significance of paired Wilcoxon tests (one-sided, PPLM > ESM2): $p < 0.05$ (*), $p < 0.01$ (**), and $p < 0.001$ (***)

Figure 2D-F:

Figure 2. Perplexity of PPLM and ESM2 on protein sequence pairs. (D-F) Example of a conserved putative transcription factor from *Vibrio cholerae* O1 biovar El Tor (PDB ID: 1Y9B). (D) Heatmap of the inter-protein attention matrix generated by PPLM. (E) Ground-truth contact map extracted from the experimental structure. (F) Three-dimensional virtualization of the complex, with interface residues identified from the inter-protein attention matrix highlighted.

3. The Reviewer commented:

3. For each of the three applications, the authors demonstrate that their method outperforms existing methods, but they offer no evidence as to how significant that outperformance is. The authors should provide the standard deviation of method performance across different folds of their cross validation experiments. Since they re-ran other methods in-house, the same should be done for other methods. Likewise, some statement about how other methods were trained, especially if they were trained at variance from the way their creators designed them.

Response: We thank the Reviewer for this valuable suggestion. In the revised manuscript, we have reported the standard deviation of model performance across cross-validation folds or test sets for all three downstream tasks, and further provide fold- or set-wise performance differences, relative percentage gains, and effect sizes (Cohen's d) comparing our methods against existing approaches. We have also added the statements to clarify the training of baseline methods.

For the binding affinity prediction task, we have reconstructed the five-fold cross-validation scheme by considering the structure similarities between proteins, and retrained PPLM-Affinity, ESM2-Affinity, and PPB-Affinity from scratch on the revised folds. We have reported the performance of all methods on each individual fold, the fold-averaged metrics expressed as mean \pm s.d. and quantify the magnitude of improvement using fold-wise mean differences (Δ), relative percentage gains, and effect sizes (Cohen's d) (**Supplementary Table S7-9**).

For the PPI prediction task, we report species-level variability by providing the mean performance and standard deviation across the five species test sets. The full set of metrics for PPLM-PPI and baseline models is presented in **Supplementary Table S2**, and the statistical comparisons are summarized in **Supplementary Table S3**.

For the inter-protein contact prediction task, we expanded the statistical evaluation of PPLM-Contact vs baseline methods by provide a complete set of paired Wilcoxon signed-rank tests, paired Cohen's d effect sizes, and Benjamini-Hochberg FDR-adjusted q -values, enabling rigorous assessment of both the magnitude and the statistical significance of performance differences (**Supplementary Tables S10-S13**).

We have updated the following paragraphs and tables to summarize the analysis results (Pages 4-8):

Page 4:

PPLM-PPI outperforms existing methods in binary protein interaction prediction

To evaluate the effectiveness of PPLM-PPI on protein–protein interaction predictions, we benchmarked it across five species, including *M. musculus*, *D. melanogaster*, *C. elegans*, *S. cerevisiae*, and *E. coli*. PPLM-PPI was compared against four state-of-the-art LM-based PPI predictors, including TUnA⁴¹, ESMDNN-PPI²³, D-SCRIPT¹⁵, and Topsy-Turvy²². All methods were installed and executed locally for reproducibility, except ESMDNN-PPI, which we reimplemented following the original publication due to the lack of publicly available source code.

An overall analysis across the five species test sets showed that PPLM-PPI demonstrated superior stability, with a mean \pm standard deviation of 0.848 ± 0.078 , compared to 0.778 ± 0.109 for TUnA, 0.762 ± 0.106 for ESMDNN-PPI, 0.532 ± 0.072 for D-SCRIPT, and 0.548 ± 0.118 for Topsy-Turvy (**Supplementary Table S2**). **Supplementary Table S3** details the statistical comparisons. Relative to TUnA and ESMDNN-PPI, PPLM-PPI improves F1-score by 10.1–10.5% and AUPRC by 9.6–11.9%, with large effect sizes (Cohen’s $d \approx 1.742$ – 6.586). Against D-SCRIPT and Topsy-Turvy, the improvements are even larger: 32.5–72.3% (F1) and 58.7–60.9% (AUPRC), also with consistently large effect sizes (Cohen’s $d \approx 2.062$ – 6.183).

Table S3. Statistical comparison between PPLM-PPI and baseline models (TUnA, ESMDNN-PPI, D-SCRIPT, and Topsy-Turvy) across the five species test sets. For each evaluation metric (F1-score, AUROC, and AUPRC), the table reports the fold-wise mean difference (Δ), relative percentage improvement, and paired effect size (Cohen’s d).

	Metric	Mean Δ	Percent gain (%)	Cohen’s d
vs TUnA	F1-score	0.068	10.1	2.252
	AUROC	0.021	2.3	1.742
	AUPRC	0.070	9.6	2.268
vs ESMDNN-PPI	F1-score	0.072	10.5	6.586
	AUROC	0.014	1.5	3.084
	AUPRC	0.086	11.9	3.026
vs D-SCRIPT	F1-score	0.319	72.3	4.765
	AUROC	0.134	16.4	2.824
	AUPRC	0.316	60.9	5.306
vs Topsy-Turvy	F1-score	0.183	32.5	3.728
	AUROC	0.15	20.0	2.062
	AUPRC	0.3	58.7	6.183

Page 5:

PPLM-Affinity achieves superior performance in protein binding affinity prediction

We evaluated PPLM-Affinity on the PPB-Affinity dataset²⁸, a large and comprehensive benchmark for protein–protein binding affinity prediction. To avoid data leakage, we regrouped the five-fold splits according to structural similarity and performed five-fold cross-validation on the revised partitions. To assess the effectiveness of language models for this task, we implemented ESM2-Affinity by concatenating the receptor and ligand sequences and fine-tuning the model using the same architecture and training procedure as PPLM-Affinity. In addition, we retrained the structure-based PPB-Affinity model using its released source code under the same cross-validation setting. Detailed fold-wise results, mean \pm standard deviation values and statistical comparison are summarized in **Supplementary Tables S7–S9**.

The left panel of **Figure 3D** reports the Pearson (PCC) and Spearman (SRCC) correlations between predicted and experimental ΔG values. These two metrics capture complementary aspects of predictive performance: PCC quantifies linear association, whereas SRCC measures rank consistency. PPLM-Affinity achieves mean \pm standard deviation PCC and SRCC values of 0.643 ± 0.058 and 0.636 ± 0.082 , representing improvements of 17.3% and 16.4% over ESM2-Affinity (0.548 ± 0.061 and 0.547 ± 0.053 ; Cohen’s $d = 1.274$ and 0.958) and 18.0% and 17.8% over the structure-based PPB-Affinity model (0.545 ± 0.072 and 0.540 ± 0.088 ; Cohen’s $d = 1.326$ and 1.064). **Supplementary Figure S6** depicts the distribution of absolute errors relative to experimental ΔG values. All three models show smooth, unimodal distributions without marked skewness. PPLM-Affinity exhibits the most concentrated error profile, with the lowest mean absolute error (1.68) and the smallest fitted dispersion ($\sigma = 1.44$), compared with higher values for ESM2-Affinity (1.85 and 1.52) and PPB-Affinity (1.85 and 1.63). These trends are consistent with the overall RMSE

comparisons, where PPLM-Affinity again attains the lowest value (2.312 ± 0.297) relative to ESM2-Affinity (2.476 ± 0.376) and PPB-Affinity (2.463 ± 0.394).

Table S9. Statistical comparison between PPLM-Affinity and baseline models (ESM2-Affinity and PPB-Affinity) across five-fold cross-validation. For each dataset and metric, the table reports the fold-wise mean difference (Δ), relative percentage improvement, and paired effect size (Cohen's d).

Dataset	PCC			SRCC			RMSE		
	Mean Δ	Percent gain (%)	Cohen's d	Mean Δ	Percent gain (%)	Cohen's d	Mean Δ	Percent gain (%)	Cohen's d
vs ESM2-Affinity									
Entire dataset	0.095	17.3	1.274	0.09	16.4	0.958	0.164	6.6	0.719
Antibody-antigen subgroup	0.205	117.1	1.45	0.213	111.5	1.645	0.137	6.7	0.96
TCR-pMHC subgroup	0.216	144.0	0.706	0.175	127.7	1.421	0.465	22.2	0.893
vs PPB-Affinity									
Entire dataset	0.097	18.0	1.326	0.096	17.8	1.064	0.152	6.2	0.62
Antibody-antigen subgroup	0.143	60.3	1.163	0.147	57.2	1.14	0.09	4.5	0.705
TCR-pMHC subgroup	0.103	39.2	0.394	0.084	36.8	0.25	0.209	11.4	0.522

Page 6:

PPLM-Contact outperforms existing methods in inter-protein contact prediction

We evaluated PPLM-Contact on four benchmark datasets: Homodimer300 and Heterodimer99 from DeepInter20, and CASP Homodimer43 and CASP Heterodimer20. Comparisons were made against five state-of-the-art methods: PLMGraph-Inter³⁴, DeepInter²⁰, CDPred¹⁹, GLINTER³⁵, and DeepHomo2.0³². All baseline methods were installed and executed locally to ensure reproducibility. Inter-protein contact precision was assessed at multiple cutoffs (top 1, 10, 50, $L/10$, $L/5$, and L), where L is the length of the shorter protein chain in the complex. Summary statistics and paired comparisons are provided in **Supplementary Table S10–13**, with per-protein results in **Supplementary Data 1–2**.

Figure 4B presents results on heterodimer test sets. On Heterodimer99, PPLM-Contact achieves 48.9% precision using experimental monomers and 45.1% using AlphaFold2-predicted monomers. These correspond to improvements of 37.0%–182.7% over competing methods, all statistically significant. On CASP Heterodimer20, PPLM-Contact attains 45.6% precision with experimental monomer structures and 40.5% with AlphaFold2 monomers, again outperforming all baseline approaches. Comprehensive statistical analyses, including paired Cohen's d effect sizes and Benjamini–Hochberg false-discovery-rate (FDR) correction across the four test sets, are provided in **Supplementary Text S4**.

Page 7:

Figure 5C–D summarize the top L precision of all general contact-prediction methods (using AlphaFold2-predicted monomer structures) and the complex structure-based methods (AlphaFold2.3, AlphaFold3, DMFold, and PPLM-Contact2) on the 434 homodimer and 119 heterodimer proteins. As expected, the integration of complex-structure features allows PPLM-Contact2 to markedly improve upon PPLM-Contact, increasing top L precision from 65.0% to 85.1% on homodimers and from 44.3% to 88.0% on heterodimers. Compared with structure-modeling approaches, PPLM-Contact2 achieves improvements of 4.8% and 7.6% over AlphaFold2.3, 8.7% and 5.6% over AlphaFold3, and 5.6% and 8.1% over DMFold on homodimers and heterodimers. All improvements are statistically significant (**Supplementary Table S15**).

Table S12. Comprehensive statistical comparison between PPLM-Contact and baseline models (PLMGraph-Inter, DeepInter, CDPred, DeepHomo2.0, and GLINTER) across four test datasets (Homodimer300, Heterodimer99, CASP_Homo43, and CASP_Hetero20) using distance maps derived from AlphaFold2-predicted monomer structures. For each dataset and baseline, improvements in per-target top L contact precision were evaluated using mean difference (Δ), relative percent gain, paired effect size (Cohen’s d), Wilcoxon signed-rank p -values, and Benjamini–Hochberg FDR-adjusted q -values. Statistically significant results ($q < 0.05$) are indicated.

Dataset	Mean Δ	Percent gain (%)	Effect size (Cohen’s d)	Wilcoxon p -value	FDR q -value	Significant ($q < 0.05$)
vs PLMGraph-Inter						
Homodimer300	22.52	51.1	0.88	3.92E-33	1.57E-32	✓
Heterodimer99	20.07	80.2	0.67	2.39E-08	4.77E-08	✓
CASP_Homodimer43	15.29	39.5	0.51	1.53E-03	2.04E-03	✓
CASP_Heterodimer20	17.20	73.9	0.53	4.95E-02	4.95E-02	✓
vs DeepInter						
Homodimer300	6.27	10.4	0.35	3.96E-13	1.59E-12	✓
Heterodimer99	12.18	37.0	0.43	6.10E-05	1.22E-04	✓
CASP_Homodimer43	6.17	12.9	0.27	1.30E-02	1.30E-02	✓
CASP_Heterodimer20	10.78	36.3	0.68	4.29E-03	5.71E-03	✓
vs CDPred						
Homodimer300	10.54	18.8	0.48	3.43E-21	1.37E-20	✓
Heterodimer99	17.64	64.2	0.55	1.69E-06	3.38E-06	✓
CASP_Homodimer43	17.09	46.4	0.66	1.51E-04	2.02E-04	✓
CASP_Heterodimer20	20.81	105.9	0.65	2.00E-02	2.00E-02	✓
vs DeepHomo2.0						
Homodimer300	22.43	50.8	0.89	1.68E-37	3.37E-37	✓
CASP_Homodimer43	22.63	72.3	0.84	1.02E-05	1.02E-05	✓
vs GLINTER						
Homodimer300	35.85	116.7	1.25	1.30E-42	5.19E-42	✓
Heterodimer99	28.28	167.9	0.88	1.29E-11	2.58E-11	✓
CASP_Homodimer43	31.48	140.0	1.09	5.33E-07	7.11E-07	✓
CASP_Heterodimer20	33.52	483.3	0.94	2.26E-03	2.26E-03	✓

Table S15. Comprehensive statistical comparison between PPLM-Contact2 and complex structure-based models (AlphaFold2.3, AlphaFold3, and DMFold). For each dataset and baseline, improvements in per-target top L contact precision were evaluated using mean difference (Δ), relative percent gain, paired effect size (Cohen’s d), Wilcoxon signed-rank p -values, and Benjamini–Hochberg FDR-adjusted q -values. Statistically significant results ($q < 0.05$) are indicated.

Dataset	Mean Δ	Percent gain (%)	Effect size (Cohen’s d)	Wilcoxon p -value	FDR q -value	Significant ($q < 0.05$)
vs AlphaFold2.3						
Homodimers	3.93	0.05	0.26	2.42E-18	4.84E-18	✓
Heterodimers	6.20	0.08	0.36	2.01E-10	2.01E-10	✓
vs AlphaFold3						
Homodimers	6.77	0.09	0.34	1.62E-23	3.24E-23	✓
Heterodimers	4.65	0.06	0.25	4.60E-05	4.60E-05	✓
vs DMFold						
Homodimers	4.53	0.06	0.28	1.34E-23	2.68E-23	✓
Heterodimers	6.57	0.08	0.40	5.51E-11	5.51E-11	✓

Among general contact-prediction methods, PPLM-Contact achieves the highest average precision, reaching 0.824 for homodimers and 0.695 for heterodimers. These values represent improvements of 5.0% and 12.1% over DeepInter, 7.6% and 7.1% over PLMGraph-Inter, 5.3% and 14.2% over CDPred, 19.3% over DeepHomo2.0 (homomers), and 28.7% and 39.7% over GLINTER. PPLM-Contact also attains the highest median precision (0.916 and 0.752), indicating consistently strong predictions across targets. All improvements are statistically significant (**Supplementary Table S16**). Integrating complex-structure features further boosts performance: PPLM-Contact2 markedly improves upon PPLM-Contact, increasing average and median precision to 0.897 and 0.962 for homodimers and 0.904 and 0.945 for heterodimers. Compared with structure-based methods, PPLM-Contact2 achieves the highest average and median precision across both homodimer and heterodimer test sets. Comprehensive statistical analyses confirm that these improvements are significant (**Supplementary Table S18**).

Table S16. Precision of interface residue identification by PPLM-Contact, DeepInter, CDPred, DeepHomo2.0 and GLINTER on the 343 homodimer and 119 heterodimer test proteins, using AlphaFold2-predicted monomer structures. The improvements of PPLM-Contact over other baseline methods were evaluated using mean difference (Δ), relative percent gain, paired effect size (Cohen's d), Wilcoxon signed-rank p -values, and Benjamini-Hochberg FDR-adjusted q -values. Statistically significant results ($q < 0.05$) are indicated.

Dataset	Average	Median	Mean Δ	Percent gain (%)	Effect size (Cohen's d)	Wilcoxon p -value	FDR q -value	Significant ($q < 0.05$)
Homodimer								
PPLM-Contact	0.824	0.916	-	-	-	-	-	-
DeepInter	0.784	0.892	0.039	5.0	0.253	7.82E-16	1.56E-15	✓
PLMGraph-Inter	0.765	0.847	0.058	7.6	0.408	1.75E-21	3.50E-21	✓
CDPred	0.782	0.869	0.041	5.3	0.173	3.29E-07	3.29E-07	✓
DeepHomo2.0	0.691	0.779	0.133	19.3	0.622	3.51E-34	3.51E-34	✓
GLINTER	0.640	0.679	0.184	28.7	0.990	8.31E-43	1.66E-42	✓
Heterodimer								
PPLM-Contact	0.695	0.752	-	-	-	-	-	-
DeepInter	0.620	0.649	0.075	12.1	0.370	2.04E-04	2.04E-04	✓
PLMGraph-Inter	0.649	0.674	0.046	7.1	0.288	2.12E-04	2.12E-04	✓
CDPred	0.608	0.679	0.087	14.2	0.409	1.63E-05	1.63E-05	✓
GLINTER	0.497	0.486	0.198	39.8	1.080	7.45E-18	7.45E-18	✓

4. The Reviewer commented:

4. To train PPLM, sequence pairs from PDB were clustered, and clusters redundant with those from the PDB were removed, yielding 629,045 clusters. But for the binding affinity prediction task, the method used five-fold cross-validation and grouped samples sharing the same PDB ID into the same fold to prevent data leakage (as mentioned in line 583). While this is a reasonable precaution to avoid exact duplicate entries between training and test sets, it does not adequately prevent homology-based leakage. Protein complexes with high sequence or structural similarity may be associated with different PDB IDs and end up in separate folds. This can lead to overestimated model performance, as the model may still encounter similar complexes in both training and testing. This issue must be examined, and the PDB IDs in each fold should be provided in supplementary materials.

Response: We thank the Reviewer for raising this important concern. To prevent potential homology-based leakage in the binding affinity prediction task, we reconstructed the entire five-fold cross-validation scheme using structure-level clustering rather than relying solely on PDB identifiers. Specifically, all protein complexes in the affinity dataset were clustered using US-align with a complex-level TM-score cutoff of 0.8, ensuring that complexes sharing high

quaternary structural similarity were grouped together. Complexes with different numbers of chains were clustered separately (e.g., dimers with dimers, trimers with trimers) to maintain meaningful structural comparability. During fold assignment, all complexes belonging to the same structural cluster were placed into the same fold, fully preventing closely related structures from being split across training and test sets. The resulting folds contain 606, 606, 605, 605, and 605 complexes, respectively, and the complete lists of PDB IDs for each fold have been provided in **Supplementary Data 7**. We subsequently retrained PPLM-Affinity and both baseline methods (ESM2-Affinity and PPB-Affinity) from scratch using this revised five-fold scheme, in which four folds were used for training and the remaining fold served as the independent validation set in each round. The detailed results were reported in **Supplementary Table S7-9**. The results demonstrate that the advantages of PPLM-Affinity are not attributable to fold leakage.

We have updated the following paragraph to clarify the insights (Page 9):

Page 9:

For protein–protein binding affinity prediction, we adopted the PPB-Affinity dataset²⁸, a large curated resource compiled from multiple public affinity databases. The dataset provides experimentally determined binding affinities, crystal structures, mutation **profiles**, and annotations of receptor and ligand chains, comprising 12,062 interaction samples from 3,032 distinct PDB entries, **including subgroups such as antibody–antigen and TCR–pMHC complexes. To prevent potential homology-based leakage, all PDB entries were clustered with US-align⁵⁰ using a complex-level TM-score cutoff of 0.8. Complexes with different numbers of chains were clustered separately to maintain meaningful structural comparability. All complexes within each structural cluster were assigned to the same fold in the five-fold cross-validation scheme. The resulting five folds contained 606, 606, 605, 605, and 605 PDB entries, respectively, and the list of PDB IDs for each fold is provided in **Supplementary Data 7**.**

5. The Reviewer commented:

Minor points:

5. - several misspellings in the text (line 58) and figure 1d.

Response: We thank the Reviewer for pointing this out. We have carefully proofread the manuscript and corrected these and other spelling errors and typographical mistakes throughout.

Response to Reviewer #3's comments

We sincerely thank the Reviewer for the insightful comments, which have substantially improved the quality of our manuscript. A major concern involved potential data leakage in the affinity and PPI tasks; we addressed this by reconstructing the affinity cross-validation using structure-level clustering and by performing detailed similarity-stratified analyses for PPI. The Reviewer also raised important points regarding the relationship between PPI and affinity predictions, AlphaFold comparisons, feature ablations, and pairwise clustering, all of which have been clarified or expanded in the revision. Finally, we updated the dataset descriptions and improved the software documentation and installation workflow. Point-by-point responses are detailed below, and all revisions in the manuscript are highlighted in yellow.

1. The Reviewer commented:

Overview

This paper introduces a new protein language model trained specifically on protein-protein interaction data. The authors cleverly modify existing architectures and show impressive performance gains in a variety of PPI-related tasks, from binary PPI prediction, to protein-ligand binding affinity prediction, to contact map prediction.

*1. My main concern is that I don't exactly understand how data leakage is guarded against in a variety of tasks. In the PPLM-Affinity task, the authors say they avoid data leakage by grouping samples that share the same PDB ID into the same fold of cross-validation. But this is insufficient, since partial crystal structures of the same protein will be given different PDB IDs. The authors need to repeat their experiments instead with the requirement of some structure similarity cutoff to avoid including the same structure across different cross-validation folds. As another example, in the PPI-prediction task, they follow D-SCRIPT's paradigm of training on human PPI, and then testing on PPI in five other species. However, it seems their MSA is species agnostic and constructed from UniRef30 using HHblits. It is hard to tell if this results in a serious data leakage problem or not. The performance they are getting is similar to deep learning methods for this problem that are trained and then tested on the same species, which makes me worried. In order to investigate further, the authors should also test a simple non-deep predictor on this dataset that relies only on the MSA: looking at the closest human homolog to any protein included in the MSA, they should predict "interact" if the human proteins participating in the same MSA interact, and don't interact otherwise. How does this simple predictor relying on the MSA do on the PPI prediction task compared to their method? The authors could also benchmark their PPI-predictor on a dataset carefully designed to avoid this problem, namely the dataset from Bernett, Judith, David B. Blumenthal, and Markus List. "Cracking the black box of deep sequence-based protein-protein interaction prediction." *Briefings in Bioinformatics* 25, no. 2 (2024), but they should still test the same suggested non-deep predictor suggested above on this dataset.*

Response: We thank the Reviewer for raising this important concern about potential data leakage for both PPI and binding affinity prediction tasks.

To prevent potential homology-based leakage in the binding affinity prediction task, we reconstructed the entire five-fold cross-validation scheme using structure-level clustering rather than relying solely on PDB identifiers. Specifically, all protein complexes in the affinity dataset were clustered using US-align with a complex-level TM-score cutoff of 0.8, ensuring that complexes sharing high quaternary structural similarity were grouped together. Complexes with different numbers of chains were clustered separately (e.g., dimers with dimers, trimers with trimers) to maintain meaningful structural comparability. During fold assignment, all complexes belonging to the same structural cluster were placed into the same fold, fully preventing closely related structures from being split across training and test sets. The resulting folds contain 606, 606, 605, 605, and 605 complexes, respectively, and the complete lists of

PDB IDs for each fold have been provided in **Supplementary Data 7**. We subsequently retrained PPLM-Affinity and both baseline methods (ESM2-Affinity and PPB-Affinity) from scratch using this revised five-fold scheme, in which four folds were used for training and the remaining fold served as the independent validation set in each round. The detailed results were reported in **Supplementary Table S7-9**. The results demonstrate that the advantages of PPLM-Affinity are not attributable to fold leakage.

We have updated the following paragraph and tables to clarify the insights (Page 9):

Page 9:

For protein–protein binding affinity prediction, we adopted the PPB-Affinity dataset²⁸, a large curated resource compiled from multiple public affinity databases. The dataset provides experimentally determined binding affinities, crystal structures, mutation profiles, and annotations of receptor and ligand chains, comprising 12,062 interaction samples from 3,032 distinct PDB entries, including subgroups such as antibody–antigen and TCR–pMHC complexes. To prevent potential homology-based leakage, all PDB entries were clustered with US-align⁵⁰ using a complex-level TM-score cutoff of 0.8. Complexes with different numbers of chains were clustered separately to maintain meaningful structural comparability. All complexes within each structural cluster were assigned to the same fold in the five-fold cross-validation scheme. The resulting five folds contained 606, 606, 605, 605, and 605 PDB entries, respectively, and the list of PDB IDs for each fold is provided in **Supplementary Data 7**.

Table S8. Mean performance of PPLM-Affinity, ESM2-Affinity, and PPB-Affinity on the full affinity dataset, the antibody–antigen subset, and the TCR–pMHC subset across five-fold cross-validation. Values represent mean \pm standard deviation of PCC, SRCC, and RMSD computed over the five folds.

Dataset	Method	PCC (mean \pm s.d.)	SRCC (mean \pm s.d.)	RMSD (mean \pm s.d.)
Entire dataset	PPLM-Affinity	0.643 \pm 0.058	0.636 \pm 0.082	2.312 \pm 0.297
	ESM2-Affinity	0.548 \pm 0.061	0.547 \pm 0.053	2.476 \pm 0.376
	PPB-Affinity	0.545 \pm 0.072	0.540 \pm 0.088	2.463 \pm 0.394
Antibody–antigen subgroup	PPLM-Affinity	0.380 \pm 0.135	0.404 \pm 0.105	1.906 \pm 0.232
	ESM2-Affinity	0.175 \pm 0.041	0.191 \pm 0.071	2.044 \pm 0.283
	PPB-Affinity	0.237 \pm 0.146	0.257 \pm 0.148	1.997 \pm 0.191
TCR–pMHC subgroup	PPLM-Affinity	0.366 \pm 0.150	0.312 \pm 0.102	1.633 \pm 0.266
	ESM2-Affinity	0.260 \pm 0.181	0.137 \pm 0.175	2.098 \pm 0.583
	PPB-Affinity	0.263 \pm 0.283	0.228 \pm 0.252	1.842 \pm 0.276

Table S9. Statistical comparison between PPLM-Affinity and baseline models (ESM2-Affinity and PPB-Affinity) across five-fold cross-validation. For each dataset and metric, the table reports the fold-wise mean difference (Δ), relative percentage improvement, and paired effect size (Cohen’s d).

Dataset	PCC			SRCC			RMSD		
	Mean Δ	Percent gain (%)	Cohen’s d	Mean Δ	Percent gain (%)	Cohen’s d	Mean Δ	Percent gain (%)	Cohen’s d
vs ESM2-Affinity									
Entire dataset	0.095	17.3	1.274	0.09	16.4	0.958	0.164	6.6	0.719
Antibody–antigen subgroup	0.205	117.1	1.45	0.213	111.5	1.645	0.137	6.7	0.96
TCR–pMHC subgroup	0.216	144.0	0.706	0.175	127.7	1.421	0.465	22.2	0.893
vs PPB-Affinity									
Entire dataset	0.097	17.8	1.326	0.096	17.8	1.064	0.152	6.2	0.62
Antibody–antigen subgroup	0.143	60.3	1.163	0.147	57.2	1.14	0.09	4.5	0.705
TCR–pMHC subgroup	0.103	39.2	0.394	0.084	36.8	0.25	0.209	11.4	0.522

We apologize for the misleading caused by our previous description of the PPI task. We would like to clarify that the PPLM-PPI model does not use multiple sequence alignments (MSA) or any evolutionary information as input. Instead, it relies solely on PPLM-derived sequence embeddings and attention matrices for the input sequence pair. Therefore, the MSA-based leakage and the suggested “simple MSA predictor” are not applicable to our model. Moreover, our training/testing setup (training on human PPI, testing on other five other species) explicitly minimizes homologous overlap, further ensuring fair evaluation and demonstrating cross-species generalization. Nevertheless, we still conducted a similarity-stratified analysis across all test species based on each protein’s maximal sequence identity to the human training set (**Supplementary Text S3**), and the results indicate that PPLM-PPI does not rely on homology-based shortcuts or inadvertent cross-species leakage.

We appreciate the Reviewer’s recommendation to benchmark on the dataset of Bennett et al. (Briefings in Bioinformatics, 2024). This dataset adopts a per-species training/testing strategy that differs fundamentally from our cross-species generalization setting, and, as our method is sequence-based rather than MSA-dependent, we have not benchmarked on this dataset. Nevertheless, we greatly appreciate the contribution of this work and have added a citation to it. in the revised manuscript.

We have updated the following paragraphs to clarify the details (Pages 11 and 6, and SI):

Page 11:

Interaction prediction using PPLM

Building upon PPLM, we developed PPLM-PPI for protein–protein interaction prediction by leveraging the protein pair representations generated by PPLM (**Figure 1B**). Given a pair of input protein sequences, PPLM is first used to extract several features: the embeddings of each sequence, the intra-protein attention matrices for both sequences, and the inter-protein attention matrix between sequences. For each feature, max pooling and mean pooling are applied independently along the sequence dimension, forming two parallel representation branches. Each pooled representation is projected into a unified dimensional space through a dedicated linear layer and subsequently passed into its own multilayer perceptron (MLP) followed by a sigmoid activation to produce an interaction probability. Each MLP consists of five linear layers, where the first four layers are followed by layer normalization and a ReLU activation. The final interaction score is obtained by averaging the outputs of the two branches. The dimensionality parameters used at each transformation step are summarized in **Tables S20–21**.

Page 6:

Finally, to assess inductive generalization and evaluate whether shortcut mechanisms might arise in our paired-input setting^{42,43}, we performed a sequence-similarity–stratified evaluation. Test interactions were grouped by the maximum single-sequence identity of either protein to any human protein in the training set, and all methods were compared within each identity interval. As detailed in **Supplementary Text S3**, PPLM-PPI maintains strong performance across all homology ranges and delivers the largest gains over existing methods, demonstrating that its predictive accuracy is not driven by shortcut learning and that the model generalizes robustly to unseen proteins under a strict cross-species inductive setting.

Supplementary Text S3. Inductive generalization analysis under sequence-similarity stratification for PPI prediction.

To evaluate whether the performance of PPLM-PPI could be affected by shortcut learning and to directly assess its inductive generalization ability, we conducted a sequence-similarity–stratified analysis across all five non-human species test sets. For every protein in each test organism, we computed its maximum sequence identity to any human protein in the training set (alignment coverage ≥ 0.8). Test interaction pairs were then grouped into seven identity bins: (0, 0.5], (0.5, 0.6], (0.6, 0.7], (0.7, 0.8], (0.8, 0.9], (0.9, 1.0), and [1.0], according to the higher identity of its two constituent proteins, which reflects the maximal similarity between any protein in the pair and the

training set. For each bin, we evaluated the AUPRC of PPLM-PPI and all baseline models and recorded the negative-to-positive sample ratio to quantify class imbalance (**Supplementary TableS6 and Supplementary Figure S5**).

Across species, the lowest-identity (0, 0.5] bin typically contains the strongest class imbalance, with negative–positive ratios ranging from 8.5:1 (*E. coli*) to 86.4:1 (mouse). Higher-identity bins are progressively less imbalanced, except for the small [1.0] bin in *D. melanogaster*, which is highly skewed due to limited sample size. This pattern indicates that raw AUPRC in extremely low-identity bins reflects the combined effect of homology and class imbalance rather than shortcut learning.

Despite large differences in class imbalance and sequence identity, PPLM-PPI maintains robust performance across all bins. In all five species, PPLM-PPI achieves high AUPRC for identity > 0.5 and does not exhibit degradation when identity decreases. In *D. melanogaster* and *C. elegans*, PPLM-PPI even attains higher AUPRC values in the (0.6, 0.9] bins than in the (0.9, 1.0] bin, showing that performance does not strictly track sequence similarity to the human training set. In the *E. coli* dataset, where all test proteins exhibit sequence identities below 0.8 relative to the human training set, PPLM-PPI also continues to deliver strong predictive accuracy.

Crucially, PPLM-PPI consistently outperforms all baseline methods in nearly every identity bin, including the most remote (0, 0.5] range. In this bin, PPLM-PPI improves AUPRC over TUN-A by 26.0%, 12.7%, 13.9%, 33.4%, and 13.3% on the *M. musculus*, *D. melanogaster*, *C. elegans*, *S. cerevisiae*, and *E. coli*. datasets, respectively. Similar or larger improvements are observed relative to ESM-based PPI, D-SCRIPT, and Topsy-Turvy. These results demonstrate that the advantage of PPLM-PPI is greatest in the most challenging remote-homology regime, where shortcut learning—if present—would be least effective.

Taken together, this similarity-stratified evaluation shows that (i) the dependence of AUPRC on identity is largely driven by underlying class imbalance and biological conservation and is shared by all models; (ii) PPLM-PPI maintains strong and stable predictive accuracy across sequence-identity ranges; and (iii) the method does not rely on protein-identity shortcuts, but instead generalizes robustly to unseen proteins in a strict cross-species inductive setting.

Figure S5:

Figure S5. AUPRC of PPLM-PPI and baseline methods across sequence-identity bins for five species test sets. (A) *M. musculus*. (B) *D. melanogaster*. (C) *C. elegans*. (D) *S. cerevisiae*. (E) *E. coli*.

2. The Reviewer commented:

Major comments

2. --*Very impressive performance gains in every task, and clever modifications to existing network architectures, PPI vs Affinity model variants*

--*The PPI task model and the Affinity task model are quite different, but when restricted to the Affinity dataset, how well do PPI model predicted probabilities correlate with binding affinity?*

Response: We thank the Reviewer for this insightful question. In the affinity dataset from PPB-Affinity, binding strength is quantified by changes in Gibbs free energy upon binding (ΔG), where more negative ΔG values indicate stronger affinity. In contrast, PPLM-PPI is trained as a binary classifier, whose output reflects the probability that two proteins form an interaction, rather than a quantitative estimate of binding strength.

From a biological and physical standpoint, these two quantities capture related but distinct endpoints. Whether two proteins interact in a cellular context depends not only on their thermodynamic affinity, but also on factors such as expression levels, localization, competition with other partners and regulatory mechanisms. Moreover, once an interaction is sufficiently favorable to occur (i.e., above a context-dependent threshold), additional changes in ΔG do not necessarily alter the qualitative outcome of “interacting vs. non-interacting”. Thus, one should not generally expect a simple monotonic function linking “probability of interaction” to absolute ΔG values across diverse systems. The affinity dataset further aggregates data from many different protein–protein systems and experimental conditions, each with its own intrinsic affinity scale and measurement protocol.

We have indeed explored quantitative correlation analyses between PPLM-PPI probabilities and ΔG under different settings. In all cases, we observed only weak and variable correlations, which is consistent with the conceptual distinction outlined above and indicates that PPLM-PPI scores are not a robust quantitative proxy for ΔG .

Given these considerations, we do not regard a single global “PPI probability vs. ΔG correlation” as a meaningful or interpretable benchmark for either task, and we have therefore not included such a correlation analysis in the main text of the manuscript. Instead, we explicitly design and evaluate a separate PPLM-Affinity model for quantitative affinity prediction, while using PPLM-PPI for its intended objective of binary interaction prediction.

Technically, PPLM-PPI and PPLM-Affinity are trained for different objectives. PPLM-PPI treats PPLM as a frozen feature extractor and trains a multilayer perceptron with a sigmoid output and binary cross-entropy loss to distinguish interacting from non-interacting pairs. By contrast, PPLM-Affinity fine-tunes the final transformer block of PPLM on the affinity dataset and adds a dedicated regression head: the pooled sequence representations are passed through fully connected layers to predict continuous binding affinity (ΔG) values, with mean squared error between predicted and experimental affinities as the training objective. By adapting PPLM’s representations under direct supervision from quantitative affinity measurements and optimizing a regression loss rather than a classification loss, PPLM-Affinity is explicitly tailored to better capture a monotonic relationship between its predictions and binding strength.

Nevertheless, we very appreciate the Reviewer for raising this important question, which is likely shared by many readers. To prevent potential ambiguity, we have added the following paragraph to the manuscript to clarify the distinction between protein–protein interaction prediction and binding affinity prediction (Page 5):

Page 5:

PPLM-Affinity achieves superior performance in protein binding affinity prediction

Although protein–protein interaction and binding affinity are related concepts, they answer different questions: PPLM-PPI estimates the likelihood that two proteins interact, whereas affinity prediction quantifies the continuous strength of that interaction. To directly model quantitative binding strength, we developed PPLM-Affinity by fine-tuning the final transformer block of PPLM to output affinity values. We evaluated PPLM-Affinity on the PPB-Affinity dataset²⁸, a large and comprehensive benchmark for protein–protein binding affinity prediction. To avoid data leakage, we regrouped the five-fold splits according to structural similarity and performed five-fold cross-validation on the revised partitions. To assess the effectiveness of language models for this task, we implemented ESM2-Affinity by concatenating the receptor and ligand sequences and fine-tuning the model using the same architecture and training procedure as PPLM-Affinity. In addition, we retrained the structure-based PPB-Affinity model using its released source code under the same cross-validation setting. Detailed fold-wise results, mean \pm standard deviation values and statistical comparison are summarized in **Supplementary Tables S7–S9**.

3. **The Reviewer commented:**

3. *–Initial Contact model should be compared to AlphaFold contact predictions. AlphaFold results are reported later when comparing to Contact2, but are relevant in assessing the quality of the original Contact model as well. By eye comparing across figures, AlphaFold outperforms the original Contact model - this should be discussed*

Response: We thank the Reviewer for this helpful suggestion. In the revised manuscript, we have added the top L contact precision and interface-residue identification accuracy of all general contact-prediction methods together with complex-structure-based methods (including AlphaFold) in the same panels (**Figure 5C–D** and **Figure 6A–B**), enabling direct comparison with the original PPLM-Contact model.

As the Reviewer noted, AlphaFold outperforms general contact-prediction methods. This is expected because AlphaFold leverages deep 3D structure–prediction pipelines, extensive geometric reasoning, and large-scale template and MSA signals, giving it substantially more structural context than sequence- or MSA-based contact predictors. To take advantage of these strengths, we had introduced PPLM-Contact2, which incorporates inter-protein distance maps from protein complex structure models. As shown in the revised figures, PPLM-Contact2 surpasses both the original PPLM-Contact and structure-based methods such as AlphaFold across all benchmark datasets.

We have updated the following paragraphs and figures to summarize the results (Pages 7 and 8):

Page 7:

Enhanced inter-protein contact prediction using predicted complex structures

Figure 5C–D summarize the top L precision of all general contact-prediction methods (using AlphaFold2-predicted monomer structures) and the complex structure-based methods (AlphaFold2.3, AlphaFold3, DMFold, and PPLM-Contact2) on the 434 homodimer and 119 heterodimer proteins. As expected, the integration of complex-structure features allows PPLM-Contact2 to markedly improve upon PPLM-Contact, increasing top L precision from 65.0% to 85.1% on homodimers and from 44.3% to 88.0% on heterodimers. Overall, all complex structure-based methods outperform general contact-prediction methods, consistent with the fact that AlphaFold leverages deep 3D structure-prediction pipelines, extensive geometric reasoning, and large-scale template and MSA signals, thereby providing much richer structural context than purely sequence- or MSA-based contact predictors. Compared with structure-modeling approaches, PPLM-Contact2 achieves improvements of 4.8% and 7.6% over AlphaFold2.3, 8.7% and 5.6% over AlphaFold3, and 5.6% and 8.1% over DMFold on homodimers and heterodimers. All improvements are statistically significant (**Supplementary Table S15**).

Figure 5C-D:

Figure 2. Perplexity of PPLM and ESM2 on protein sequence pairs. (C–D) top L contact precision of general contact-prediction methods (GLINTER, DeepHomo2.0, CDPred, PLMGraph-Inter, DeepInter, and PPLM-Contact) and complex structure-based approaches (AlphaFold2.3, AlphaFold3, DMFold, and PPLM-Contact2) on 343 homodimers and 119 heterodimers, respectively.

Page 8:

PPLM-Contact enhances interface residue recognition

Among general contact-prediction methods, PPLM-Contact achieves the highest average precision, reaching 0.824 for homodimers and 0.695 for heterodimers. These values represent improvements of 5.0% and 12.1% over DeepInter, 7.6% and 7.1% over PLMGraph-Inter, 5.3% and 14.2% over CDPred, 19.3% over DeepHomo2.0 (homomers), and 28.7% and 39.7% over GLINTER. PPLM-Contact also attains the highest median precision (0.916 and 0.752), indicating consistently strong predictions across targets. All improvements are statistically significant (**Supplementary Table S16**). Integrating complex-structure features further boosts performance: PPLM-Contact2 markedly improves upon PPLM-Contact, increasing average and median precision to 0.897 and 0.962 for homodimers and 0.904 and 0.945 for heterodimers. Compared with structure-based methods, PPLM-Contact2 achieves the highest average and median precision across both homodimer and heterodimer test sets. Comprehensive statistical analyses confirm that these improvements are significant (**Supplementary Table S18**).

Figure 6A-B:

Figure 6. Comparative analysis of protein–protein interface residue identification. (A–B) Precision of interface-residue identification by general contact-prediction methods (GLINTER, DeepHomo2.0, PLMGraph-Inter, CDPred, DeepInter, and PPLM-Contact) and complex structure-based approaches (AlphaFold2.3, AlphaFold3, DMFold, and PPLM-Contact2) on 343 homodimers and 119 heterodimers, respectively.

4. The Reviewer commented:

4. --Ablation study shows that the PPLM embeddings give the smallest boost in performance out of all the input features, might be worth noting/discussing that

Response: We thank the Reviewer for highlighting this interesting point. When designing PPLM-Contact, we first analyzed existing contact-prediction pipelines and found that most input information falls into two major categories: (i) structure- or geometry-derived distance features, and (ii) MSA-based co-evolution features. These two categories already provide rich and high-resolution intra-protein information, but they lack explicit inter-protein interaction knowledge, which is directly relevant to the task.

To fill this gap while avoiding redundancy with existing inputs, we intentionally incorporated only the inter-protein attention matrices from PPLM, which encode interaction-aware pairwise semantics learned from protein–protein sequence pairs. We did not include other PPLM-derived features such as sequence embeddings or intra-chain attention, since these signals overlap with information already captured by the MSA and structural features.

As the Reviewer observed, the inter-protein attention matrices contribute the smallest absolute performance gain among the three feature categories. This is expected given that we deliberately excluded PPLM features that may be redundant with co-evolution or structure inputs. Nevertheless, it is worth emphasizing that even this single PPLM feature still provides a consistent and meaningful improvement, increasing the top L contact precision from 73.3% to 77.2% for homodimers and 45.3% to 49.7% for heterodimers, whereas the MSA-derived inputs comprise six separate features (**Supplementary Table S22**).

We have updated the following paragraphs to clarify these insights (Pages 7 and 8):

Page 6:

Enhanced inter-protein contact prediction using predicted complex structures

Figure 4D–F summarize the mean and standard deviation of top L precision across three random seeds for PPLM-Contact and all ablation variants. These results show that removing any major network component or feature of the model reduces precision, with the largest performance drops observed when excluding the triangle-multiplication module, PPLM inter-protein attention, MSA features, or monomer distance maps. Detailed ablation analyses and statistical comparisons are provided in **Supplementary Text S5**. The effect of MSA depth is further examined in **Supplementary Text S6**.

Supplementary Text S5. Ablation analysis of PPLM-Contact.

To evaluate the contribution of individual architectural modules and feature types, we trained six ablated variants of PPLM-Contact, each excluding one key component while keeping all other settings unchanged: (i) w/o cross-attn, which removes the cross-attention module from the architecture; (ii) w/o self-attn, which removes the self-attention module; (iii) w/o tri-multi, which excludes the triangle-multiplication module; (iv) w/o MSA, which excludes MSA-derived features including PSSM, DCA, and ESM-MSA-1b outputs; (v) w/o PPLM, which removes inter-protein attention features generated by PPLM; and (vi) w/o Mdist, which omits monomer distance maps extracted from either experimental or AlphaFold-predicted monomer structures.

Figure 4D–F illustrates the overall performance changes of these variants on the Homodimer300 and Heterodimer99 test sets. For each variant, we report the mean top L inter-protein contact precision and 95 % confidence intervals (CI) across three random seeds. Detailed top 1 to top L results are provided in **Supplementary Table S13**. Compared with the full model ($77.2 \pm 0.58\%$ for homodimers and $49.7 \pm 1.02\%$ for heterodimers), all ablations resulted in consistent and statistically meaningful declines, confirming the robustness of each component’s contribution.

At the network-architecture level, removing either cross-attention or self-attention modules led to modest yet reproducible reductions—by approximately 0.8–1.0 percentage points (abbreviated as pp) for homodimers and 1.7–2.0 pp for heterodimers—indicating their supportive roles in enhancing intra- and inter-chain contextualization. In contrast, excluding the triangle-multiplication module caused a much greater decline, reducing precision by 6.7 pp and 9.4 pp on homodimers and heterodimers, respectively, highlighting its central role in capturing residue-pair geometric relationships.

At the feature level, removing the PPLM inter-protein attention features—the paired-sequence embeddings generated by PPLM—lowered precision from 77.2% to 73.3% for homodimers and

from 49.7% to 45.3% for heterodimers, underscoring PPLM’s key role in modeling cross-chain interactions. Removing MSA-derived features reduced precision by 7.0 pp and 9.3 pp, reflecting the contribution of co-evolutionary information, while omitting M-distance features yielded the largest degradation (23.3 pp and 16.8 pp for homodimers and heterodimers, respectively), emphasizing the indispensable role of intra-chain geometric priors. The inter-protein attention matrices contribute the smallest absolute performance gain among the three feature categories. This is expected, as we intentionally used only this single inter-protein attention feature—rather than also including sequence embeddings or intra-chain attention—to provide complementary interaction-aware information without duplicating intra-protein features that are already richly captured by structure- and MSA-based inputs.

Together, these analyses confirm that every component of PPLM-Contact contributes meaningfully to predictive accuracy, with particularly strong and statistically supported effects from the triangle-multiplication, PPLM inter-protein attention, MSA, and monomer distance modules.

Figure 4D-E:

Figure 4. Performance of PPLM-Contact and the comparison methods on the inter-protein contact prediction. (D–E) Mean top L contact precision and 95% confidence intervals (CI) of PPLM-Contact and its ablation variants across three seeds on the Homodimer300 and Heterodimer99 test sets, respectively.

5. The Reviewer commented:

5. --Not clear how the pairwise nature of the data was handled when clustering the dataset with MMseqs2.

Is the 80% coverage minimum applied to each member of the pair separately?

Are two pairs clustered together only if both members of one pair are highly homologous to a member of the other pair or just one?

Which clustering method (greedy set cover or connected component) was used in MMseqs clustering?

Response: We thank the Reviewer for the insightful questions regarding the handling of pairwise data during MMseqs2-based clustering. In the revised manuscript, we have added a detailed clarification of the clustering criteria and procedures (**Supplementary Text S7**).

First, the minimum coverage requirement was applied to each member of the pair separately. For two sequence pairs $A = (A_1, A_2)$ and $B = (B_1, B_2)$, we considered the two pairs redundant only when both sequence alignments, either (A_1 with B_1 and A_2 with B_2) or (A_1 with B_2 and A_2 with B_1), satisfy at least 50% sequence identity and at least 80% alignment coverage.

Second, two pairs were assigned to the same cluster only when both sequences of one pair had homologs to the two sequences of the other pair, with the matching evaluated separately for each chain and allowing either chain correspondence (A_1-B_1/A_2-B_2 or A_1-B_2/A_2-B_1) to meet the identity and coverage thresholds.

Third, clustering was not performed using the built-in greedy set cover or connected component algorithms in MMseqs2. Instead, MMseqs2 was used solely to generate per-sequence homology tables under the 50% identity and 80% coverage cutoffs. Pairwise clustering was then carried out with a custom clustering procedure.

We have updated the following paragraphs to clarify the clustering process (Page 9 and SI):

Page 9:

Datasets

To train PPLM, we constructed a composite protein dataset of protein interaction sequence pair by integrating protein complexes released before January 1, 2024 from the Protein Data Bank (PDB)⁴⁷ and interaction sequences from the STRING⁴⁸ database. After filtering for physical interfaces and removing redundancy through pair-level clustering using MMseqs2⁴⁹ together with a custom clustering procedure, the dataset comprised 25,245 heteromeric clusters and 23,082 homomeric clusters from PDB, as well as 629,045 clusters from STRING, covering over 3.3 million protein sequence pairs. For model validation, we randomly selected 5,000 singleton clusters and used the remaining clusters for training. After removing validation pairs that were redundant to any pairs in the training clusters, the final validation set consisted of 1,204 heteromeric, 1,003 homomeric, and 2,471 STRING-derived pairs (**Supplementary Text S7**). During the training of PPLM, sequence pairs were sampled from the PDB and STRING clusters at a 1:2 ratio.

Supplementary Text S7. Construction of sequence-pair datasets.

We constructed a composite dataset of protein–protein interaction sequence pairs by integrating protein complex structures from the Protein Data Bank (PDB) and high-confidence interaction pairs from the STRING database. From the PDB, we extracted sequence pairs from complexes released before January 1, 2024, retaining only those in which at least one pair of residues from different chains exhibited a C_{β} – C_{β} (C_{α} for Glycine) distance within 8Å. This yielded 273,295 heteromeric and 86,915 homomeric sequence pairs. To further expand the diversity and coverage of interaction types, we incorporated 2,995,604 high-confidence sequence pairs from STRING.

To remove redundancy at the protein-pair level, we applied a custom MMseqs2-assisted clustering procedure. For two sequence pairs $A = (A_1, A_2)$ and $B = (B_1, B_2)$, redundancy was defined only when both sequences of one pair had homologs to the two sequences of the other pair. Specifically, two pairs were considered redundant when either (A_1 with B_1 and A_2 with B_2) or (A_1 with B_2 and A_2 with B_1) satisfied at least 50% sequence identity and at least 80% alignment coverage. The identity and coverage thresholds were applied separately to each chain in the pair.

MMseqs2 was first used to identify homologs for every sequence in the dataset under the 50% identity and 80% coverage cutoffs (alignment mode 3, coverage mode 0), resulting in a homolog table for each sequence. All sequence pairs were then sorted in descending order of combined chain length. The first pair was taken as the representative of the first cluster. For each subsequent pair, we examined whether its two sequences appeared, respectively, in the homolog tables of the two sequences in any existing representative pair. If this criterion was satisfied, the pair was assigned to that cluster; otherwise, a new cluster was created with the query pair as its representative. This procedure was repeated until all pairs had been assigned.

This clustering strategy was applied separately to heteromeric PDB pairs, homomeric PDB pairs, and STRING-derived pairs, due to differences in the masking strategies used during model training. This resulted in 25,245 heteromeric clusters, 23,082 homomeric clusters, and 629,045 STRING clusters.

We then randomly selected 1,306 heteromeric, 1,194 homomeric, and 2,500 STRING-derived clusters that each contained only a single sequence pair for model validation. The remaining clusters were used for training. To ensure strict non-redundancy, we further removed all validation pairs sharing at least 50% sequence identity and at least 80% coverage with any training pair, regardless of heteromeric or homomeric status and the source of pairs. This filtering eliminated 332 pairs and yielded a final validation set of 1,204 heteromeric pairs, 1,003 homomeric pairs, and 2,471 STRING-derived pairs, for a total of 4,678 validation sequence pairs.

6. The Reviewer commented:

Minor Comments

6. --Line 95 talks about independent benchmark datasets in such a way that it sounds like all these datasets were created by the authors: in fact, based on the more detailed description further on in the paper, they choose several benchmarks constructed in previous work for their tests: it only strengthens the paper to use an existing benchmark. But they should make clear here that the PPI dataset they train and test on is nearly the same as the D-SCRIPT dataset (with some small housekeeping adjustments), and the binding affinity dataset they use is the PPB-Affinity dataset.

Response: We thank the Reviewer for this helpful suggestion. In the revised manuscript, we have removed this misleading paragraph and provided a clear and explicit description of all datasets in the Datasets section.

We have updated the following paragraphs to clarify the clustering process (Page 9):

Page 9:

Datasets

To train PPLM, we constructed a composite protein dataset of protein interaction sequence pair by integrating protein complexes released before January 1, 2024 from the Protein Data Bank (PDB)⁴⁷ and interaction sequences from the STRING⁴⁸ database. After filtering for physical interfaces and removing redundancy through pair-level clustering using MMseqs2⁴⁹ together with a custom clustering procedure, the dataset comprised 25,245 heteromeric clusters and 23,082 homomeric clusters from PDB, as well as 629,045 clusters from STRING, covering over 3.3 million protein sequence pairs. For model validation, we randomly selected 5,000 singleton clusters and used the remaining clusters for training. After removing validation pairs that were redundant to any pairs in the training clusters, the final validation set consisted of 1,204 heteromeric, 1,003 homomeric, and 2,471 STRING-derived pairs (**Supplementary Text S7**). During the training of PPLM, sequence pairs were sampled from the PDB and STRING clusters at a 1:2 ratio.

To train and evaluate PPLM-PPI for PPI prediction, we adopted benchmark datasets from D-SCRIPT¹⁵ spanning six species: *H. sapiens*, *M. musculus*, *D. melanogaster*, *C. elegans*, *S. cerevisiae*, and *E. coli*. Given the relative rarity of true PPIs, each dataset maintains a 10:1 ratio of negative to positive pairs. To ensure data integrity, we identified and removed duplicate, erroneous, and invalid samples arising from the random negative sampling process (**Supplementary Text S8**). The PPLM-PPI model was trained and validated on the *H. sapiens* dataset and tested on the remaining five species datasets. Detailed dataset statistics are listed in **Table S12**.

For protein–protein binding affinity prediction, we adopted the PPB-Affinity dataset²⁸, a large curated resource compiled from multiple public affinity databases. The dataset provides experimentally determined binding affinities, crystal structures, mutation profiles, and annotations of receptor and ligand chains, comprising 12,062 interaction samples from 3,032 distinct PDB entries, including subgroups such as antibody–antigen and TCR–pMHC complexes. To prevent potential homology-based leakage, all PDB entries were clustered with US-align⁵⁰ using a complex-level TM-score cutoff of 0.8. Complexes with different numbers of chains were clustered separately to maintain meaningful structural comparability. All complexes within each structural cluster were assigned to the same fold in the five-fold cross-validation scheme. The resulting five folds contained 606, 606, 605, 605, and 605 PDB entries, respectively, and the list of PDB IDs for each fold is provided in **Supplementary Data 7**.

For inter-protein contact prediction, we adopted the same datasets used in DeepInter²⁰ to ensure fair comparison with existing methods. These datasets consist of non-redundant homodimers and heterodimers curated from the PDB. The training set includes 3,504 homodimers and 1,881 heterodimers, while the validation set comprises 296 homodimers and 96 heterodimers. The test set contain 300 homodimers (Homodimer300) and 99 heterodimers (Heterodimer99). We also constructed two independent test sets comprising 43 homodimer targets (CASP_Homodimer43) and 20 heterodimer targets (CASP_Heterodimer20) collected from CASP13 to CASP16 to further assess model robustness and real-world applicability (**Supplementary Data 1 and 2**). Two sets of PPLM-Contact models were trained separately for homodimers and heterodimers: the homodimer model was trained exclusively on homodimers, whereas the heterodimer model was trained on the full combined training set.

7. The Reviewer commented:

7. --Some scatterplots have too many points (figures 3E,F S2B-E)- should convert to 2D histograms or color each point based on the density of other points around it

Response: We thank the Reviewer for this helpful suggestion. In the revised manuscript, we have replaced the scatterplots in Figures 3E-F and corresponding Supplementary Figures with hexagonal density maps. This representation avoids overplotting in high-density regions and provides a clearer visualization. The color scale now reflects the local point density (count), which allows the linear trend and correlation structure to be more easily interpreted.

We have updated the following figures in the manuscript (Figure 3E-F):

Figure 3E-F:

Figure 3. Performance of PPLM-PPI and PPLM-Affinity on protein-protein interaction and binding affinity prediction. (E-F) Head-to-head comparison between experimental and predicted binding affinity for PPLM-Affinity and ESM2-Affinity, respectively. Plots show hexagonal density maps to improve visibility at high point density. Color intensity reflects the local log-transformed point density.

8. The Reviewer commented:

8. Software installation and testing notes

--References to `config.py` should always use `pplm_contact/config.py` to avoid confusion

--Should specify that Uniclust30 needs to be unzipped

--Parameter downloads from their lab server seem to be throttled to ~250KB/s - this is impractically slow

--Would be nice to have a script that downloads all model parameters instead of having to do each separately

--Had to uninstall and re-install MKL (version 2023.1.0-h213fc3f_46344) to get the conda environment to work

--Image at top of github page is cropped

--Issue with `libperl.so` file not found despite it's presence in the conda environment folder - prevented PPLM-Contact and PPLM-Contact2 from running properly

PPLM-PPI and PPLM-Affinity run properly on test cases

Response: We sincerely thank the Reviewer for the time and effort to thoroughly test the software and report these installation issues. We have updated the GitHub repository to improve installation instructions, added a unified parameter-download script, and resolved dependency problems. To address the slow download speed, we now host all model parameters on public cloud storage (e.g., Google Drive).

Regarding the MKL and `libperl.so` issues, although we were unable to reproduce these errors in freshly created conda environments on our test machines, we have provided a tested

environment.yml file and added a troubleshooting section to the README describing how to resolve MKL-related import errors and libperl.so loading problems. We hope these revisions will make installation more robust and further improve usability and reproducibility, and we are very grateful for the Reviewer's careful feedback.

Response to Reviewer #4

We thank Reviewer #4 for co-reviewing our manuscript as part of the Nature Communications early-career reviewer initiative and sincerely appreciate their contribution to the evaluation of this work.

1. The Reviewer commented:

Response: We thank the Reviewer for co-reviewing our manuscript. Our detailed point-by-point responses are provided in the corresponding sections above.